# Nuclear phosphoinositide signaling promotes YAP/TAZ-TEAD transcriptional activity in breast cancer

Oisun Jung[1,2], Min-jeong Baek[1,2,3], Colin Wooldrik [1,2,3], Keith R Johnson[1,2,4], Kurt W Fisher [2,5], Jinchao Lou [6], Tanei J Ricks[7], Tianmu Wen[8], Michael D Best[6], Vincent L Cryns [8], Richard A Anderson [8] & Suyong Choi [1,2]✉

## Abstract

The Hippo pathway effectors Yes-associated protein 1 (YAP) and its homolog TAZ are transcriptional coactivators that control gene expression by binding to TEA domain (TEAD) family transcription factors. The YAP/TAZ–TEAD complex is a key regulator of cancer-specific transcriptional programs, which promote tumor progression in diverse types of cancer, including breast cancer. Despite intensive efforts, the YAP/TAZ–TEAD complex in cancer has remained largely undruggable due to an incomplete mechanistic understanding. Here, we report that nuclear phosphoinositides function as cofactors that mediate the binding of YAP/TAZ to TEADs. The enzymatic products of phosphoinositide kinases PIP-KIα and IPMK, including phosphatidylinositol 4,5-bisphosphate (PI(4,5)$P_2$) and phosphatidylinositol 3,4,5-trisphosphate (P(I3,4,5)$P_3$), bridge the binding of YAP/TAZ to TEAD. Inhibiting these kinases or the association of YAP/TAZ with PI(4,5)$P_2$ and PI(3,4,5)$P_3$ attenuates YAP/TAZ interaction with the TEADs, the expression of YAP/TAZ target genes, and breast cancer cell motility. Although we could not conclusively exclude the possibility that other enzymatic products of IPMK such as inositol phosphates play a role in the mechanism, our results point to a previously unrecognized role of nuclear phosphoinositide signaling in control of YAP/TAZ activity and implicate this pathway as a potential therapeutic target in YAP/TAZ-driven breast cancer.

**Keywords** YAP; Phosphoinositide; Hippo Pathway; PIP5K1A; IPMK
**Subject Categories** Cancer; Post-translational Modifications & Proteolysis; Signal Transduction

## Introduction

Phosphoinositides are lipid messengers that control many aspects of human physiology, and their roles in proliferation, survival, and motility are well-established (Di Paolo and De Camilli, 2006; Posor et al, 2022). Amongst the 7 phosphoinositides, P(I4,5)$P_2$ is the most abundant species and a key intermediate in phosphoinositide metabolism and signaling. PI(4,5)$P_2$ is further phosphorylated by phosphoinositide 3-kinases (PI3Ks) to generate PI(3,4,5)$P_3$. PI(4,5)$P_2$ and P(I3,4,5)$P_3$ bind to protein targets known as effectors and regulate cellular functions by modulating the activities, subcellular localizations, and protein-protein interactions of the effectors (Choi et al, 2015, 2018). Canonically, phosphoinositides are considered to be confined at membranes; however, the discovery of a nuclear pool of phosphoinositides (Boronenkov et al, 1998) has expanded our knowledge of phosphoinositide signaling (Barlow et al, 2010). We and others previously showed that a substantial fraction of various phosphoinositide species including PI(4,5)$P_2$ and PI(3,4,5)$P_3$ are found in the nucleoplasm in regions distinct from the nuclear membrane, and their levels are changed by diverse stimuli (Choi et al, 2019b; Sobol et al, 2018). Although many cytoplasmic PI(4,5)$P_2$ and PI(3,4,5)$P_3$ effectors have been identified, the nature and functions of PI(4,5)$P_2$ and PI(3,4,5)$P_3$ and their effectors in the nucleus are only now starting to emerge (Chen et al, 2022; Mellman et al, 2008; Tatomer et al, 2014).

Consistent with a nuclear pool of PI(4,5)$P_2$ and PI(3,4,5)$P_3$, phosphoinositide-generating kinases are also found in the nucleus. Phosphatidylinositol 4-phosphate 5-kinase type 1α (PIPKIα, encoded by the *PIP5K1A* gene) generates PI(4,5)$P_2$ by phosphorylating the 5-OH of the inositol head group of PI(4)P (Choi et al, 2015). Unlike its close homolog PIPKIγ (encoded by *PIP5K1C* gene), a large fraction of PIPKIα is found in the nucleus (Boronenkov et al, 1998), and we showed that depleting PIPKIα reduces nuclear PI(4,5)$P_2$ by 60% (Chen et al, 2020; Choi et al, 2019b), indicating that PIPKIα is the major enzyme impacting nuclear PI(4,5)$P_2$ signaling. Inositol polyphosphate multikinase (IPMK) has the PI3K activity and localizes in the nucleus (Maag et al, 2011). Depletion of IPMK in mouse embryonic fibroblasts

[1]Eppley Institute for Research in Cancer and Allied Diseases, University of Nebraska Medical Center, Omaha, NE, USA. [2]Fred & Pamela Buffett Cancer Center, University of Nebraska Medical Center, Omaha, NE, USA. [3]Interdisciplinary Graduate Program in Biomedical Sciences, University of Nebraska Medical Center, Omaha, NE, USA. [4]Department of Oral Biology, University of Nebraska Medical Center, Omaha, NE, USA. [5]Department of Pathology and Microbiology, University of Nebraska Medical Center, Omaha, NE, USA. [6]Department of Chemistry, University of Tennessee, 1420 Circle Drive, Knoxville, TN 37996, USA. [7]Department of Chemistry, University of Memphis, 3744 Walker Avenue, Memphis, TN 38152, USA. [8]University of Wisconsin Carbone Cancer Center, University of Wisconsin School of Medicine and Public Health, University of Wisconsin-Madison, Madison, WI, USA. ✉E-mail: schoi@unmc.edu

reduces global PI(3,4,5)P$_3$ levels ~40% (Maag et al, 2011). Overexpressed IPMK is largely found in the nucleus of COS-7 cells and dramatically increases the level of nuclear PI(3,4,5)P$_3$ (Resnick et al, 2005). These data indicate that IPMK is a key enzyme maintaining the nuclear PI(3,4,5)P$_3$ pool. Recently, we have shown that upon DNA damage IPMK catalyzes the conversion of PI(4,5)P$_2$ to PI(3,4,5)P$_3$ bound to p53 and promotes the recruitment and activation of AKT on p53 in the nucleus (Chen et al, 2022), suggesting a key role for this PI3K in nuclear phosphoinositide signaling.

The Hippo pathway is a master regulator of organ size and tissue homeostasis (Boopathy and Hong, 2019; Harvey et al, 2013; Ma et al, 2019). The Hippo kinase cascade (TAOK→MST1/2→LATS1/2) phosphorylates and maintains YAP/TAZ in the cytoplasm. Unphosphorylated YAP/TAZ, in contrast, translocate to the nucleus and control the transcription of oncogenic genes (including CTGF and CYR61) by binding to a set of transcription factors such as TEAD, SMAD, RUNX, and p53 family proteins (Kim et al, 2018). The TEAD family transcription factors are major DNA-binding partners of YAP/TAZ in many tissues (Kim et al, 2018). Enhanced nuclear localization and constitutive association with the TEADs are frequently found in many types of cancer, including breast, liver, and colon cancer (Zanconato et al, 2016b). While the YAP/TAZ–TEAD pathway is dispensable for normal homeostasis of adult tissues, it critically regulates cancer-specific transcriptional addiction supporting cancer cell proliferation, survival, and motility (Zanconato et al, 2016a, 2016b, 2018). These findings have resulted in intensive efforts to develop therapeutics targeting the YAP/TAZ–TEAD pathway in cancer. Peptide mimetics and small molecule inhibitors that block the binding of YAP/TAZ with the TEADs were shown to suppress YAP/TAZ-dependent transcription and tumor growth in mouse models (Liu-Chittenden et al, 2012; Pobbati and Rubin, 2020). However, the clinical translation of these agents has been limited by their low solubility, nonspecific side effects, and poor pharmacokinetics (Dey et al, 2020). A better understanding of the molecular mechanisms of the binding of YAP/TAZ with the TEADs would provide novel methods to target the pathway in cancer. There are indications that the YAP/TAZ pathway is regulated by phosphatidic acid (PA) and PI transfer proteins (Han et al, 2018; Li et al, 2022), but the mechanisms for this regulation are lacking.

In this study, we discovered unexpected roles of nuclear phosphoinositides in regulating the YAP/TAZ–TEAD pathway. In response to membrane receptor activation, we found YAP/TAZ translocate to the nucleus and interact with PIPKIα and IPMK. In the nucleus, PI(4,5)P$_2$, PI(3,4,5)P$_3$, and possibly other products enzymatically generated by IPMK, function as cofactors facilitating the binding of YAP/TAZ with the TEADs. We identified the phosphoinositide-binding motifs on YAP and showed that mutants which are unable to bind to PI(4,5)P$_2$ and PI(3,4,5)P$_3$ are greatly impaired in their ability to interact with the TEADs. Furthermore, depletion of PIPKIα and IPMK disrupted the binding of YAP/TAZ with the TEADs, leading to lower expression of YAP/TAZ target genes and reduced cell motility in breast cancer cells. Taken together, our results demonstrate that nuclear phosphoinositide signaling is a key regulator of YAP/TAZ-dependent transcription. Also, our discovery demonstrates that PIPKIα and IPMK are novel therapeutic targets to disrupt the YAP/TAZ–TEAD pathway in cancer.

# Results

## PIPKIα and IPMK control the expression of YAP/TAZ target genes independent of mutant p53

We previously showed that nuclear PIPKIα maintains mutant p53 stability by recruiting a family of small heat shock proteins (Choi et al, 2019b). Wild-type (WT) p53 is a transcription factor, while most mutant p53 proteins found in cancer have lost the ability to bind DNA but still regulate transcription by associating with other transcription regulators such as transcription factors and coactivators (Pfister and Prives, 2017). Mutant p53 forms a complex with YAP/TAZ and regulates the expression of YAP/TAZ target genes such as cyclins and cyclin-dependent kinases (Di Agostino et al, 2006, 2016).

To further test if PIPKIα controls YAP/TAZ target gene expression via mutant p53, the protein expression of two well-characterized YAP/TAZ target genes CTGF and CYR61 (encoded by the CCN2 and CCN1 genes, respectively) was measured by immunoblotting. Knockdown of PIPKIα dramatically reduced CTGF and CYR61 levels in the triple-negative breast cancer cell lines MDA-MB-231 and MDA-MB-468 (Figs. 1A and EV1A). Unexpectedly, however, the depletion of mutant p53 in MDA-MB-231 cells had no impact on CTGF and CYR61 protein levels, indicating that PIPKIα regulates CTGF and CYR61 expression via a mutant p53-independent mechanism.

IPMK is a nuclear-localizing PI3K and is reported to function as a transcriptional coactivator (Wang et al, 2017; Xu et al, 2013a, 2013b), but the exact mechanisms of how IPMK regulates transcription remain to be elucidated. Interestingly, knockdown of IPMK also reduced the protein expression of CTGF and CYR61 (Figs. 1A and EV1A). AXL receptor tyrosine kinase is a well-established YAP/TAZ target gene in many cell types (King et al, 2020; Wang et al, 2018; Yamaguchi and Taouk, 2020). However, we observed no difference in AXL protein level in PIPKIα or IPMK-depleted MDA-MB-468 cells (Fig. EV1A). This is consistent with prior reports that YAP depletion or inhibition has no impact on AXL mRNA level in MDA-MB-231 cells (Zanconato et al, 2018) and in lung cancer cell lines at the protein level (Saab et al, 2019). Based on our and the prior observations, it appears that AXL is a cellular context-dependent YAP/TAZ target gene.

The roles of PIPKIα and IPMK in the expression of YAP/TAZ target genes were further tested at the mRNA level (Fig. 1B,C). Depletion of PIPKIα or IPMK significantly reduced the mRNA level of CTGF and CYR61 along with the other YAP/TAZ targets (Moya and Halder, 2019; Zanconato et al, 2018) ANKRD1, survivin, cyclin A2, and KIF23 (encoded by the ANKRD1, BIRC5, CCNA2, KIF23 genes, respectively). Consistently, depletion of PIPKIα or IPMK significantly reduced promoter activity of a YAP/TAZ reporter (Dupont et al, 2011) (Fig. EV1B).

## PIPKIα/IPMK interact with the WW domains of YAP via LPXYY motifs

To test how PIPKIα and IPMK control YAP/TAZ target gene expression, the physical association of the proteins was analyzed by in vitro binding assays with recombinant proteins expressed in E. coli. We found that His$_6$-tagged PIPKIα and IPMK were pulled down with GST-tagged YAP (GST-YAP) but not with GST alone

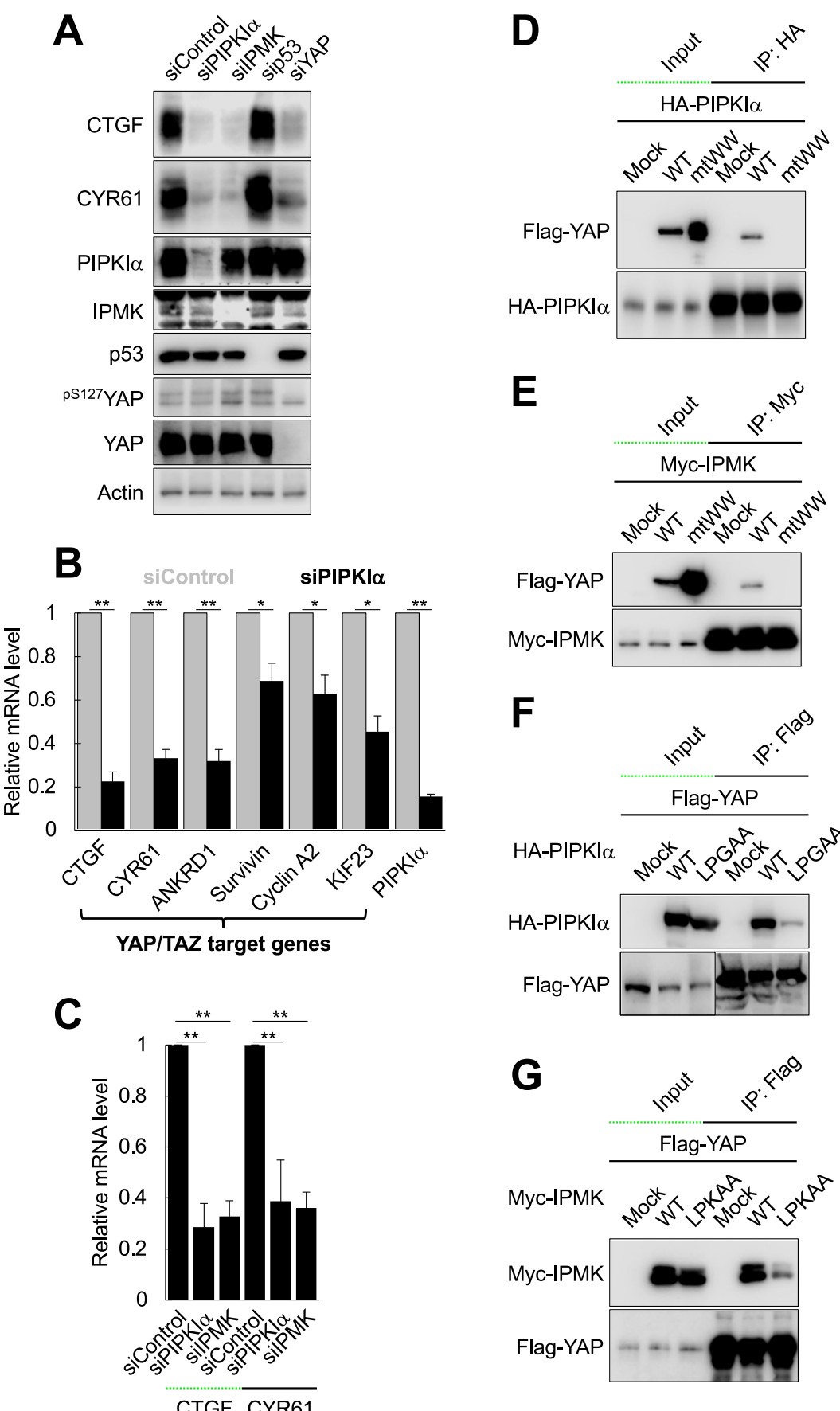

◀   **Figure 1.   PIPKIα and IPMK control the expression of YAP/TAZ target genes by physical association.**

(A) The indicated proteins were knocked down in MDA-MB-231 cells by transfecting siRNAs for 72 h and protein levels were analyzed by immunoblotting. Representative immunoblots of $n = 3$ independent experiments are shown. Similar to YAP knockdown, knockdown of either PIPKIα or IPMK strongly reduced the expression of the YAP targets CTGF and CYR61 at the protein level while p53 knockdown had no effect. (B) MDA-MB-231 cells were transfected with non-targeting siRNA (siControl) vs. siRNA against human PIPKIα. The mRNA levels of the indicated YAP/TAZ target genes were analyzed by RT-qPCR. The graph is shown as mean ± s.d. of $n = 3$ independent experiments. *$P < 0.05$; **$P < 0.01$, and n.s.; not significant in Student's $t$ test. (C) PIPKIα and IPMK were knocked down in MDA-MB-231 cells and the mRNA levels of CTGF and CYR61 were analyzed by RT-qPCR. The graph is shown as mean ± s.d. of $n = 3$ independent experiments. Knockdown of either PIPKIα or IPMK reduced the expression of CTGF and CYR61 at the mRNA level. *$P < 0.05$; **$P < 0.01$, and n.s.; not significant in Student's $t$ test. (D, E) Vector control (Mock) or Flag-tagged WT or mtWW YAP were co-transfected with HA-tagged PIPKIα (D) or Myc-tagged IPMK (E) in HEK293 cells for 48 h. PIPKIα and IPMK were immunoprecipitated with the indicated antibodies and the associated YAP was analyzed by immunoblotting. Representative immunoblot images of $n = 3$ independent experiments are shown. Inactivating the WW domains of YAP prevented it from binding PIPKIα and IPMK. (F, G) WT HA-PIPKIα (F) or WT Myc-IPMK (G) or the LPXAA mutants were co-transfected with Flag-YAP in HEK293 cells for 48 h. YAP was immunoprecipitated with an anti-Flag antibody and the associated PIPKIα and IPMK were analyzed by immunoblotting. Representative immunoblot images of $n = 3$ independent experiments are shown. Mutating the LPXAA motifs in PIPKIα and IPMK strongly reduced their association with YAP. Source data are available online for this figure.

(Fig. EV1C,D), indicating that PIPKIα and IPMK bind directly to YAP.

Next, the domains and motifs that are responsible for the binding were mapped via immunoprecipitation (IP) after ectopic expression in HEK293 cells. YAP contains WW domains (Fig. EV1E) that mediate interactions with its regulators (Iglesias-Bexiga et al, 2015; Varelas, 2014; Vargas et al, 2020). WT YAP and a mutant with disrupted ligand binding pockets in the two WW domains (mtWW) (Kim et al, 2020; Oka et al, 2008) (Fig. EV1E) were expressed and their association with PIPKIα and IPMK was analyzed by IP. Compared to WT, the co-IP of mtWW YAP was dramatically reduced even though the expression of mtWW was much higher, indicating that PIPKIα and IPMK interact with YAP via the WW domains (Fig. 1D,E).

Proline-rich sequences, typically PPXY and LPXY motifs (where P is proline, L is leucine, X is any amino acid, and Y is tyrosine), are well-established ligands for the WW domains of YAP/TAZ (Bruce et al, 2008; Ingham et al, 2005). The 195-LPGYY-199 sequence in PIPKIα and the 110-LPKYY-114 sequence in IPMK were mutated to 195-LPGAA-199 and 110-LPKAA-114, respectively, and analyzed for their interaction with YAP by IP. Compared to WT, the LPXAA mutants showed greatly reduced interactions with YAP (Fig. 1F,G), indicating that YAP binds with PIPKIα and IPMK via the LPXYY motifs. These data collectively indicate that PIPKIα/IPMK and YAP interact via the LPXY motifs and the WW domains (Fig. EV1F).

## Extracellular stimuli enhance the interaction of YAP with PIPKIα and IPMK in the nucleus

Often the interaction of phosphoinositide kinases with their binding partners is regulated by extracellular stimuli (Choi et al, 2013, 2015, 2016). To explore if agonists control the interaction of YAP with PIPKIα and IPMK, MDA-MB-231 cells were stimulated with serum (Plouffe et al, 2018), lysophosphatidic acid (LPA) (Yu et al, 2012), stromal cell-derived factor 1α (SDF-1α) (Yu et al, 2012), and epidermal growth factor (EGF) (Fan et al, 2013). These agents are known to induce the dephosphorylation and nuclear accumulation of YAP/TAZ. The interactions were analyzed by IP. In serum-starved conditions, the interaction of YAP with PIPKIα and IPMK was poorly detected. However, upon stimulation with serum or LPA, the interactions were dramatically increased (Fig. EV1G). Modest enhancement of the interactions was observed

when cells grown with 10% serum were further stimulated with LPA, SDF-1α, or EGF (Fig. EV1H). Similarly, the interaction between ectopically expressed YAP and PIPKIα was slightly increased by LPA (Fig. EV1I,J). In these conditions, the phosphorylation of YAP was decreased and the expression of CTGF was increased (Fig. EV1G–K).

In vivo the interaction of YAP with PIPKIα and IPMK was further analyzed by proximity ligation assay (PLA), which measures an interaction in situ (Fredriksson et al, 2002). In serum-starved conditions, few PLA signals were detected, consistent with the IP results (Fig. EV1G). Upon serum stimulation, which is one of the most potent stimuli for the YAP/TAZ pathway (Plouffe et al, 2018; Yu et al, 2012), the YAP-PIPKIα and the YAP-IPMK PLA signals were significantly increased in the cytoplasm and more robustly in the DAPI-positive nuclei (Fig. 2A,B). To investigate how serum regulates YAP interactions with PIPKIα and IPMK, the distribution of endogenous proteins was visualized by immunofluorescence. A substantial fraction of YAP was located in the nucleus of MDA-MB-231 cells consistent with a prior report (Andrade et al, 2017), and serum stimulation dramatically increased YAP nuclear localization compared to serum-starved conditions, while PIPKIα and IPMK were largely in the nucleus regardless of serum (Fig. 2C,D), pointing out that serum-induced YAP interactions with PIPKIα and IPMK are largely caused by nuclear accumulation of YAP. In support, serum-induced YAP nuclear accumulation was independent of PIPKIα and IPMK (Fig. EV1L), and PIPKIα and IPMK remained associated with TEADs independent of serum stimulation and YAP knockout (KO) (Fig. EV1M). These data collectively indicate that in response to agonist activation of membrane receptors YAP translocates to the nucleus and forms complexes with PIPKIα and IPMK.

## PIPKIα and IPMK control the association of YAP/TAZ with the TEAD family

Upon activation of membrane receptors, including integrins, receptor tyrosine kinases (RTKs), and G-protein-coupled receptors (GPCRs), the Hippo kinase cascade is inactivated, and the unphosphorylated YAP/TAZ translocate into the nucleus and bind to TEAD family transcription factors (Boopathy and Hong, 2019; Ma et al, 2019; Pobbati and Rubin, 2020). The YAP/TAZ–TEAD axis is responsible for the transcriptional expression of a set of tumor-promoting genes, including CTGF and CYR61 (Boopathy

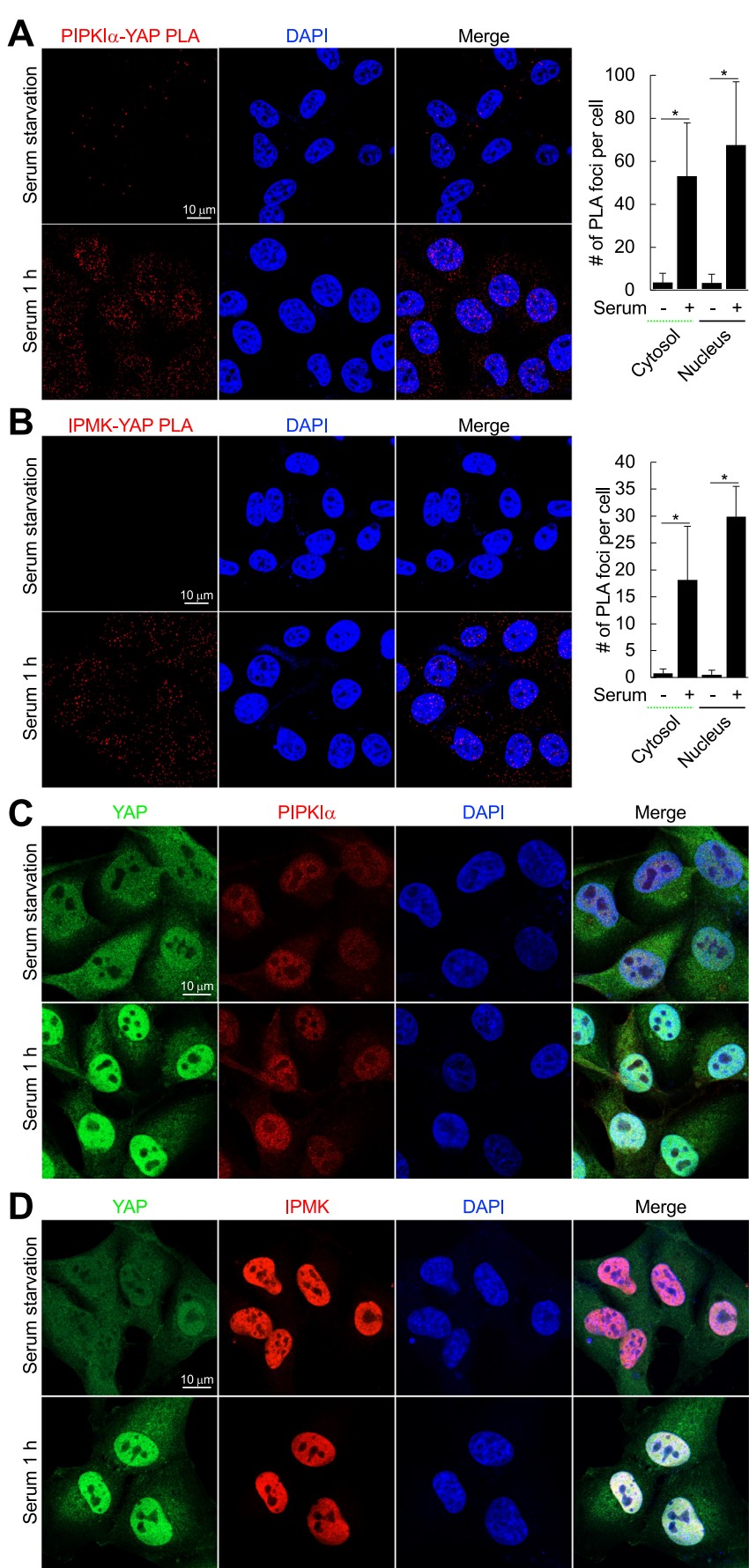

**Figure 2. Serum stimulation facilitates YAP nuclear translocation and the association of YAP with PIPKIα and IMPK.**

(A, B) Serum-starved MDA-MB-231 cells were stimulated with 10% serum for 1 h. Cells were then fixed and PIPKIα association with YAP (A) and IMPK association with YAP (B) were visualized by PLA. The images were obtained by confocal microscopy. The number of PLA puncta was counted from at least ten cells and the graph is shown as mean ± s.d. of $n = 3$ independent experiments. DAPI staining was used to distinguish the nucleus from the cytoplasm. Scale bar, 10 μm. Treating the cells with serum significantly increased the number of PIPKIα-YAP and IMPK-YAP PLA foci. *$P < 0.05$; **$P < 0.01$, and n.s.; not significant in Student's *t* test. (C, D) Untransfected MDA-MB-231 cells were serum-starved and then treated with 10% serum for 1 h. Cells were fixed and the distribution of YAP with PIPKIα (C) or IMPK (D) were visualized by immunofluorescence. Representative confocal images of $n = 3$ independent experiments are shown. Scale bar, 10 μm. Treating the cells with serum dramatically increased YAP nuclear staining. Source data are available online for this figure.

and Hong, 2019; Moya and Halder, 2019). Our results showed that neither YAP phosphorylation nor nuclear accumulation is affected by the depletion of PIPKIα and IMPK (Figs. 1A and EV1K,L). To further test how PIPKIα and IMPK control YAP/TAZ target gene expression, the association of YAP/TAZ with the TEADs was measured by IP. Knockdown of either PIPKIα or IMPK, but not the PIPKIα homolog PIPKIγ, dramatically reduced the co-IP of YAP/TAZ with the TEADs (Figs. 3A and EV2A).

IMPK is a nuclear-localized PI3K that regulates YAP/TAZ signaling. However, in many cell types class IA PI3Ks are responsible for generating the majority of cellular PI(3,4,5)P₃ (Maag et al, 2011; Resnick et al, 2005; Vanhaesebroeck et al, 2010). To test whether other PI3Ks are involved in regulating the YAP/TAZ pathway, the catalytic subunits of two class IA PI3K isoforms (p110α and p110β) were knocked down and the interaction of YAP/TAZ with the TEADs and the expression of CTGF/CYR61 were measured (Fig. 3B). IMPK knockdown dramatically inhibited the interaction of YAP/TAZ with the TEADs and the expression of YAP/TAZ target genes, while the depletion of p110α and p110β had no detectable effect.

The interaction of YAP/TAZ with TEAD1 was further tested in cells via PLA. The YAP-TEAD1 PLA signals were largely detected in the nuclei of parental MDA-MB-231 cells and were significantly diminished in PIPKIα and IMPK KO cells (Fig. 3C,D). To investigate whether the kinase activity of PIPKIα and IMPK is required for modulating the YAP/TAZ–TEAD pathway, the KO cells were reconstituted with either WT or the previously validated kinase-dead (KD) forms of PIPKIα (Choi et al, 2016, 2019b) and IMPK (Xu et al, 2013a). Expression of the WT constructs fully rescued the YAP-TEADs interaction and the expression of CTGF/CYR61, but expression of the KD mutants had no or minor impacts (Fig. 3E,F), indicating that the generation of PI(4,5)P₂ and PI(3,4,5)P₃ (or other enzymatic products of IMPK) is required for regulating the YAP/TAZ–TEAD pathway. Using in vitro kinase assays, we tested if the binding of YAP to PIPKIα and IMPK regulated their kinase activities. We found the addition of recombinant YAP did not significantly affect their kinase activities (Fig. EV2B,C). Taken together, these data demonstrate that specific enzymatic products generated by PIPKIα and IMPK, but not by the other phosphoinositide kinases tested, control the expression of YAP/TAZ target genes by regulating the interaction of YAP/TAZ with TEAD family transcription factors in the nucleus.

## YAP binds to PI(4,5)P₂ and PI(3,4,5)P₃ and the binding is enhanced by extracellular stimuli

Often, phosphoinositide kinase interacting proteins also bind to the phosphoinositide products of the kinases, which modulate the

function of the target proteins (Chen et al, 2020; Choi et al, 2013, 2015, 2016, 2018, 2019b). To determine if YAP binds to phosphoinositides, purified GST-YAP isolated from *E. coli* was incubated with PIP Beads (Echelon Biosciences) and immunoblotting was used to determine if they interacted. This demonstrated robust binding of recombinant YAP with PI(5)P and we observed more binding with PI(4,5)P₂ and PI(3,4,5)P₃. Binding to other phosphoinositide species and phosphatidylinositol (PI) was minimal (Fig. 4A). This indicates the binding of YAP to the inositol head group of phosphoinositides is specific to PI(5)P, PI(4,5)P₂, and PI(3,4,5)P₃. Binding affinities between YAP and phosphoinositides were measured using microscale thermophoresis (MST) (Fig. EV2D). No significant binding of GST-YAP with PI was detected, while the binding constants ($K_d$) of GST-YAP binding to PI(4,5)P₂ and PI(3,4,5)P₃ were determined to be 165.67 ± 13.50 and 8.13 ± 2.63 nM, respectively (Fig. EV2D; Appendix Fig. S1). Considering that the cellular concentrations of PI(4,5)P₂ and PI(3,4,5)P₃ are greater than several μM in stimulated cells (Insall and Weiner, 2001), the binding of YAP to PI(4,5)P₂ and PI(3,4,5)P₃ could occur at physiological concentrations.

The association of phosphoinositides with the YAP-TEADs complex was further assayed in cells via a metabolic labeling approach. Acetylated 2-azidopropylinositol (Ac₃2API) is a *myo*-inositol probe that can be metabolically incorporated into the cellular PI and phosphoinositide species (Fig. EV2E) (Ricks et al, 2019). After the copper-catalyzed click reaction, the azide group contained within metabolic products labeled by probe 2API is readily conjugated with an alkyne-biotin. Thus, the biotinylated PI/phosphoinositides complexed with proteins can be detected by an HRP-tagged streptavidin in western blot (WB). After click reaction endogenous YAP was subjected to IP and a strong streptavidin signal was observed at the size of YAP in SDS-PAGE but not in the non-clicked conditions. A relatively weaker signal was detected at the size of endogenous TEADs (Figs. 4B and EV2F). By comparison, cellular Myc (c-Myc) which is another serum-activated transcription factor failed to show a strong association with biotinylated 2API (Fig. EV2G). These indicate that a *myo*-inositol probe is metabolically incorporated into the YAP-TEADs complex, and the incorporation is strong enough to be resistant to SDS-PAGE. To further validate the specificities of the PI(4,5)P₂ and PI(3,4,5)P₃ antibodies used in this study (Sharma et al, 2008), we compared their immunoblotting signals with that of streptavidin in clicked lysates from cells treated with Ac₃2API. Detecting phosphoinositide-binding proteins by immunoblotting was validated previously (Carrillo et al, 2023). As shown in Fig. EV2H, some but not all the bands recognized by streptavidin overlapped with the bands recognized by the antibodies to PI(4,5)P₂ and PI(3,4,5)P₃, implying that the PI(4,5)P₂ and PI(3,4,5)P₃ antibodies at least in part react with antigens in cells.

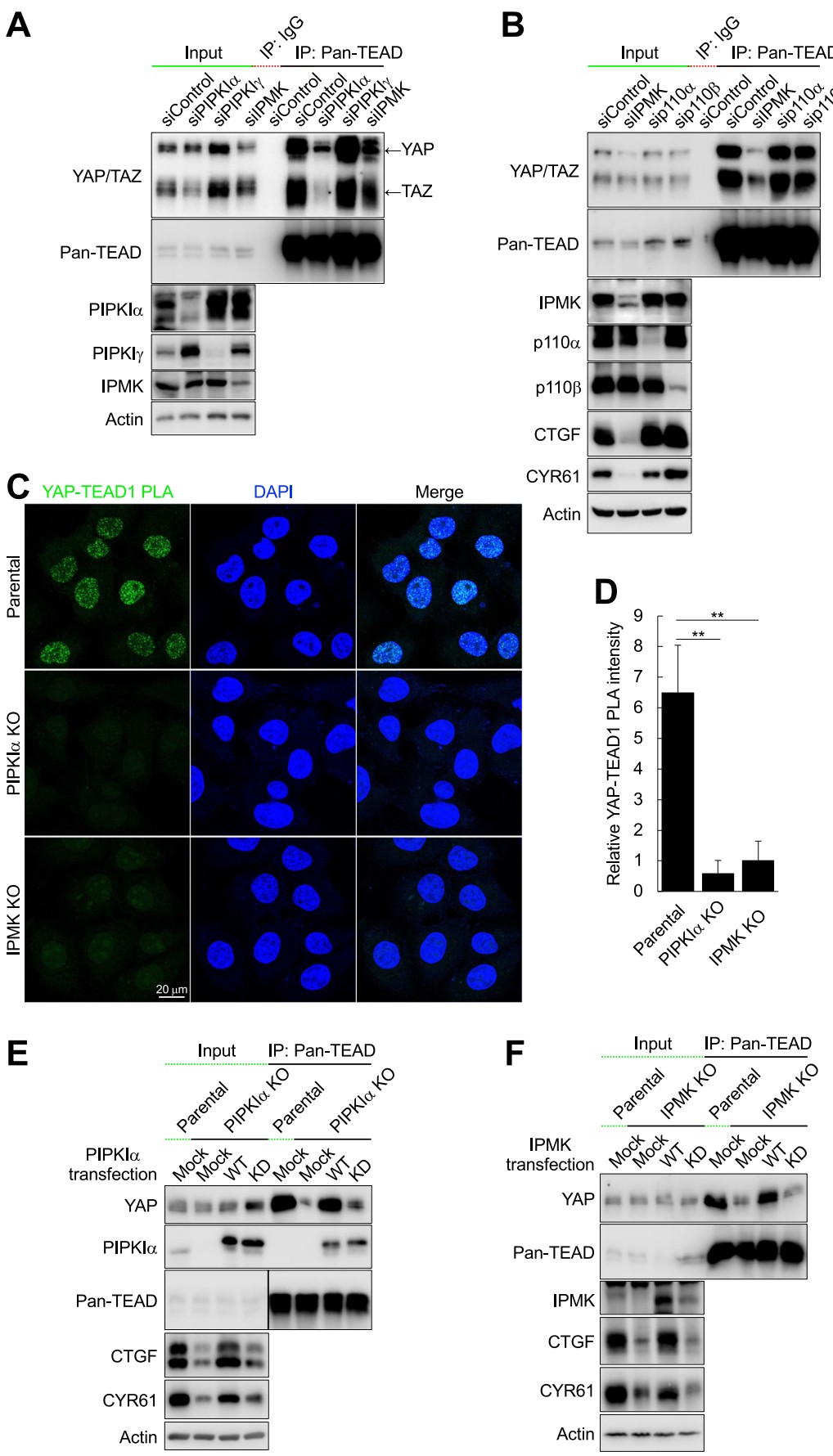

◀ **Figure 3.   The kinase activity of PIPKIα and IPMK is required for mediating the interaction of YAP/TAZ with the TEADs.**

(A, B) PIPKIα, PIPKIγ, IPMK, p110α, and p110β were knocked down in MDA-MB-231 cells by transfecting siRNA for 72 h. Endogenous TEAD proteins were immunoprecipitated with an anti-pan-TEAD antibody and the associated protein complexes were analyzed by immunoblotting. Representative immunoblot images of $n = 3$ independent experiments are shown. Knocking down PIPKIγ, p110α, or p110β did not affect the association of YAP/TAZ with the TEADs while knockdown of either PIPKIα or IPMK did. In addition, knockdown of PIPKIα and IPMK largely reduced the expression of CTGF and CYR61. (C, D) Parental or the indicated KO cells (pooled clones) grown in 10% serum were fixed and the association of YAP with TEAD1 was visualized by PLA. The images were obtained by confocal microscope (C). Scale bar, 20 μm. The intensity of the PLA signal was quantified using ImageJ software from at least ten cells and the graph shows the intensity as mean ± s.d. of $n = 3$ independent experiments (D). *$P < 0.05$; **$P < 0.01$, and n.s.; not significant in Student's $t$ test. (E, F) PIPKIα (E) and IPMK (F) KO MDA-MB-231 cells were transiently transfected with empty vector (Mock), WT, or kinase-dead (KD) PIPKIα and IPMK mutants. Endogenous TEAD proteins were immunoprecipitated and the associated YAP was analyzed by immunoblotting. Representative immunoblot images of $n = 2$ independent experiments are shown. The WT enzymes but not the KD rescued the association of YAP with the TEADs. Source data are available online for this figure.

We previously showed that PLA is a powerful technique to detect cellular interactions of phosphoinositides with proteins (Chen et al, 2022; Choi et al, 2019b). In serum-starved conditions, PI(4,5)P$_2$-YAP PLA was detected, mainly in the cytoplasm. The PLA signal was greatly enhanced by serum stimulation and the increase was largely seen in the nuclei (Fig. 4C). PI(3,4,5)P$_3$-YAP PLA was weakly detected in serum-starved conditions but was significantly elevated upon serum stimulation and this was detected prominently in the nuclei (Fig. 4D). Z-stack images revealed that about 90% of the serum-stimulated PLA signals were detected in the nuclei (Fig. EV2I,J). In addition, stimulation with LPA also significantly increased the PI(4,5)P$_2$-YAP and PI(3,4,5)P$_3$-YAP complexes largely in the nuclei detected by PLA (Fig. EV2K,L). These data show that, upon agonist stimulation, YAP becomes associated with PI(4,5)P$_2$ and PI(3,4,5)P$_3$ in the nuclear compartment, which is consistent with the agonist-induced nuclear association of PIPKIα and IPMK with YAP (Fig. 2A,B).

To determine if the PI(4,5)P$_2$ and PI(3,4,5)P$_3$ that are associated with YAP are generated by PIPKIα and IPMK, PIPKIα and IPMK were depleted with siRNA, and the YAP-phosphoinositide association was measured by PLA in serum-stimulated conditions. Knockdown of PIPKIα eliminated the PI(4,5)P$_2$-YAP PLA signal and, unexpectedly, knockdown of IPMK also significantly reduced the PLA signal (Fig. 4E,F). Knockdown of either PIPKIα or IPMK strongly reduced the PI(3,4,5)P$_3$-YAP PLA signals (Fig. 4G,H). Because of differences in the signals seen in Fig. EV2H, we could not exclude the possibility of cross-reactivity of the PI(4,5)P$_2$ and PI(3,4,5)P$_3$ antibodies with other molecules, such as phosphorylated proteins in the complex. Thus, we further tested the association of phosphoinositides with YAP via metabolic labeling. Knockdown of PIPKIα or IPMK greatly reduced the association of YAP with products resulting from 2API in the metabolic labeling assay (Fig. 4I). Collectively, our data indicate that, upon agonist stimulation, YAP associates with the PI(4,5)P$_2$ and PI(3,4,5)P$_3$ (and perhaps other products) that are generated in the nucleus by the activities of PIPKIα and IPMK.

## The binding of phosphoinositides to YAP facilitates its interaction with the TEADs

We next investigated the functional outcome of YAP binding to phosphoinositide species. YAP/TAZ bind to the TEADs via the TEAD-binding domain (TBD), and the TEADs bind to YAP/TAZ via the YAP/TAZ-binding domain (Y/TBD) (Fig. EV3A) (Boopathy and Hong, 2019; Dey et al, 2020). YAP/TAZ do not contain canonical phosphoinositide-binding modules (e.g., the pleckstrin

homology, phox homology, and C2 domains (Itoh and Takenawa, 2002)), but a stretch of basic amino acids known as a polybasic motif (PBM) often mediates the interactions of phosphoinositides with proteins (Choi et al, 2013, 2015, 2016, 2019a; Papayannopoulos et al, 2005; Tan et al, 2015b). Interestingly, we found a potential PBM in the TBD of YAP/TAZ, raising the possibility that the binding of phosphoinositides may regulate the binding of YAP/TAZ to the TEADs. We hypothesize that the negatively charged inositol head group of the phosphoinositides electrostatically interacts with the positively charged amino acids in the PBM, and the hydrophobic acyl chains of the phosphoinositides interact with the reported hydrophobic TBD-Y/TBD interface (Li et al, 2010; Tian et al, 2010) (Fig. 5A).

To test this hypothesis, in vitro binding assays were performed with recombinant YAP/TAZ and TEAD1 in the presence of various synthetic and natural lipids (Figs. 5B–F and EV3B–E). PI(4,5)P$_2$ from the porcine brain (natural) significantly enhanced the binding of TEAD1 to YAP in vitro, while synthetic diC8 lipids had no impact (Fig. 5B). Both natural PI(4)P and PI(4,5)P$_2$ enhanced the interaction of TAZ with TEAD1 (Fig. EV3D). The impact of natural source PI(5)P and PI(3,4,5)P$_3$ could not be tested as they are not commercially available. To circumvent this issue, we instead used synthetic phosphoinositides consisting of stearic and arachidonic acids (denoted as 18:0/20:4). Natural source phosphoinositides have variable acyl chains largely consisting of arachidonic (20:4), stearic (18:0), and oleic acids (18:1) (Gu et al, 2013; Traynor-Kaplan et al, 2017). 18:0/20:4 PI(5)P, PI(4,5)P$_2$, and PI(3,4,5)P$_3$ significantly enhanced the binding of TEAD1 to YAP, and these synthetic lipids were comparable to natural source PI(4,5)P$_2$ (Fig. 5C). To further test the specificity of YAP-TEAD1 binding mediated by phosphoinositides, similar in vitro binding assays were performed in the presence of an increasing amount of 18:0/20:4 PI(4,5)P$_2$ and PI(3,4,5)P$_3$ (Fig. 5D,E). The TEAD1 binding to YAP was increased and then saturated with a 0.1 μM or greater PI(4,5)P$_2$ and PI(3,4,5)P$_3$ concentrations, demonstrating that PI(4,5)P$_2$ and PI(3,4,5)P$_3$ mediate a specific YAP-TEAD1 binding rather than a concentration-dependent nonspecific binding. These data collectively indicate that the inositol head group and the acyl chain composition of phosphoinositides are critical for mediating the binding of YAP/TAZ to TEAD1 in vitro.

IPMK has both inositol phosphate and phosphoinositide kinase activity, and inositol phosphates generated by IPMK are reported to mediate protein-protein interactions (Hanakahi et al, 2000; Lee et al, 2021; Scott and Kleiger, 2020). To test which IPMK enzymatic products regulate the binding of YAP with TEAD1, their interaction was assayed in the presence of all known IPMK

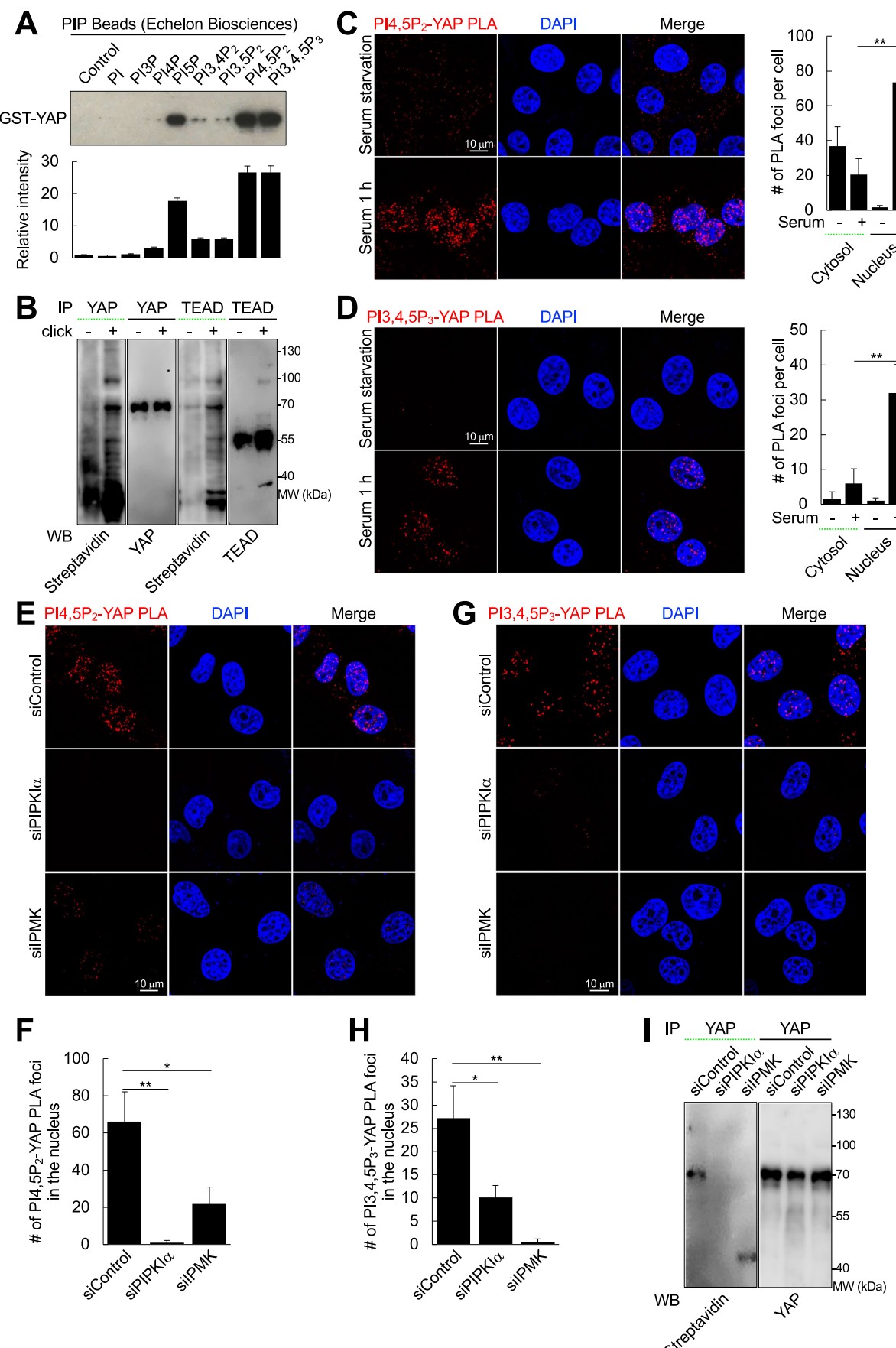

◀ **Figure 4.  YAP interacts with PI(4,5)P$_2$ and PI(3,4,5)P$_3$ in the nucleus in response to serum stimulation.**

(A) In all, 0.5 µM GST-YAP was incubated with 10 volume % of the indicated PIP Beads (Echelon Biosciences). After washing, the amount of YAP associated with phosphoinositide-coated beads was analyzed by immunoblotting. The intensity of the immunoblots was quantified using ImageJ and the graph shows the mean ± s.d. of $n = 3$ independent experiments. (B) Starved MDA-MB-231 cells were fed with 50 µM Ac$_3$2API for 24 h in the presence of 10% dialyzed serum. Cells were lysed and 2API was conjugated to biotin–alkyne after click reaction. Endogenous YAP and pan-TEAD were immunoprecipitated and the associated complexes were analyzed by immunoblotting with anti-YAP and pan-TEAD antibodies. Biotinylated 2API was resolved by streptavidin. Representative immunoblot images of $n = 3$ independent experiments are shown. Biotinylated 2API signal was detected at the size of endogenous YAP and pan-TEAD. (C, D) Serum-starved MDA-MB-231 cells were stimulated with 10% serum for 1 h. Cells were fixed and the association of YAP with PI(4,5)P$_2$ (C) or PI(3,4,5)P$_3$ (D) was visualized by PLA. The images were obtained by confocal microscopy. The number of PLA puncta was counted from at least 10 cells and the graph is shown as mean ± s.d. of $n = 3$ independent experiments. DAPI staining was used to distinguish the nucleus from the cytoplasm. Treating the cells with serum markedly enhanced the PLA signals in the nuclei. Scale bar, 10 µm. *$P < 0.05$; **$P < 0.01$, and n.s.; not significant in Student's $t$ test. (E–H) MDA-MB-231 cells were transfected with siRNAs targeting either PIPKIα or IPMK for 72 h. Serum-starved cells were stimulated with 10% serum and the association of YAP with PI(4,5)P$_2$ (E) or PI(3,4,5)P$_3$ (G) was visualized by PLA. The images were obtained by confocal microscopy. The number of PLA puncta was counted from at least ten cells and the graph shows the mean ± s.d. of $n = 3$ independent experiments (F, H). DAPI staining was used to distinguish the nucleus from the cytoplasm. Knocking down either kinase reduced the number of nuclear PLA puncta. Scale bar, 10 µm. *$P < 0.05$; **$P < 0.01$, and n.s.; not significant in Student's $t$ test. (I) MDA-MB-231 cells were transfected with siRNAs targeting either PIPKIα or IPMK for 48 h. Cells were starved for 24 h, then fed with Ac$_3$2API for 15 h. Cells were lysed and 2API was conjugated to biotin–alkyne after click reaction. Endogenous YAP was immunoprecipitated and the associated complexes were analyzed by immunoblotting. Representative immunoblot images of $n = 2$ independent experiments are shown. The association of 2API with YAP was reduced by PIPKIα or IPMK knockdown. Source data are available online for this figure.

products (Fig. 5F). IPMK generates inositol 1,3,4,5-tetrakisphosphate (Ins(1,3,4,5)P$_4$), Ins(1,4,5,6)P$_4$, Ins(1,3,4,5,6)P$_5$, and PI(3,4,5)P$_3$ (Lee et al, 2021). All three inositol phosphates were tested at 0.1 µM concentration, and none had an observable impact on the binding. In contrast, PI(4,5)P$_2$ and PI(3,4,5)P$_3$ at 0.1 µM greatly induced the binding of YAP and TEAD1 in vitro, directly pointing out the specificity of phosphoinositides mediating the interaction.

We next examined if YAP, the phosphoinositides, and TEAD1 can form a ternary complex in cells via immunostaining. Serum-stimulated PI(4,5)P$_2$-YAP and PI(3,4,5)P$_3$-YAP PLA foci nicely colocalized with TEAD1 in the nucleus (Fig. 5G,H), suggesting that the nuclear ternary complex might be a functional unit required for the full activation of the YAP/TAZ–TEAD pathway (see below).

## The binding of phosphoinositides to YAP is required for its binding with the TEADs and the expression of YAP/TAZ target genes

We identified a potential phosphoinositide-binding region in the TBD of YAP (the polybasic motif). We tested the functional importance of the PBM via a mutagenesis approach. Positively charged Arg87, Lys90, and Lys97 were mutated to structurally similar but neutral glutamines to generate the 1Q, 2Q, and 3Q mutants (Fig. 6A). These mutants were tested for their ability to bind PI(4,5)P$_2$ and PI(3,4,5)P$_3$ using PIP Beads. Compared to WT YAP, the 1Q and 2Q showed substantially reduced binding to PI(4,5)P$_2$ and PI(3,4,5)P$_3$, and the binding was completely abrogated with the 3Q mutant (Fig. 6B), indicating that YAP binds with phosphoinositides via Arg87, Lys90, and Lys97. Similar multivalent and cooperative PBM-phosphoinositide bindings were reported in N-WASP (Papayannopoulos et al, 2005), LAPTM4B (Tan et al, 2015a), and IQGAP1 (Choi et al, 2013), and now shown in YAP. We further utilized the phosphoinositide-binding-defective 2Q and 3Q mutants in an in vitro assay to investigate how phosphoinositide binding controls the YAP–TEAD interaction. Compared to PI, the addition of PI(4,5)P$_2$ and PI(3,4,5)P$_3$ markedly increased the binding of WT YAP to TEAD1. However, these lipids did not promote the binding of the 2Q and 3Q mutants to TEAD1 (Fig. 6C).

The impact of phosphoinositide binding to YAP on its interaction with the TEADs was tested in cells via IP after overexpression in HEK293 cells. Exogenous WT YAP co-IPed with TEAD4 as expected (Chen et al, 2010), while the 2Q and 3Q mutants did not significantly interact with TEAD4 (Fig. 6D). These changes in the interactions were not due to altered nuclear localization (Fig. EV3F) or the failure of the 2Q and 3Q mutants to bind PIPKIα and IPMK (Fig. EV3G,H).

Besides the TEAD family, YAP also interacts with other transcription factors including p73 (Levy et al, 2007). Interestingly, the 2Q and 3Q mutants still associated with p73 (Fig. 6E), suggesting that phosphoinositide binding specifically controls the YAP–TEAD interaction. These data collectively demonstrate that the binding of phosphoinositides to YAP is critical for mediating its association with TEAD both in vitro and in vivo. The overexpression of WT YAP in HEK293 cells is reported to increase CTGF expression (Zhao et al, 2008). Consistently, we observed that ectopic expression of WT YAP elevates the protein level of CTGF, while the 2Q and 3Q mutants had no impact (Fig. 6F), indicating that the binding of YAP to phosphoinositides also controls the expression of YAP/TAZ target genes.

## The PIPKIα/IPMK→PI(4,5)P$_2$/PI(3,4,5)P$_3$→YAP/TAZ→TEAD pathway regulates breast cancer cell motility

The YAP/TAZ–TEAD pathway is known to regulate essentially every step of cancer progression, including survival, proliferation, motility, and immune evasion (Dey et al, 2020; Ma et al, 2019; Zanconato et al, 2016b). To investigate how the phosphoinositide-driven YAP/TAZ–TEAD pathway regulates cellular functions, we quantified changes in cell proliferation and motility using PIPKIα and IPMK KO MDA-MB-231 cells. Depletion of PIPKIα and IPMK did not alter proliferation (Fig. EV4A), whereas serum-induced chemotaxis and laminin, vitronectin, and collagen-induced haptotaxis were significantly reduced by PIPKIα and IPMK KO (Fig. EV4B; Appendix Fig. S2A).

To further test the impact of phosphoinositide binding to YAP in breast cancer cell motility, endogenous YAP was depleted by siRNAs targeting the 3'UTR and then rescued with WT, 2Q, or 3Q

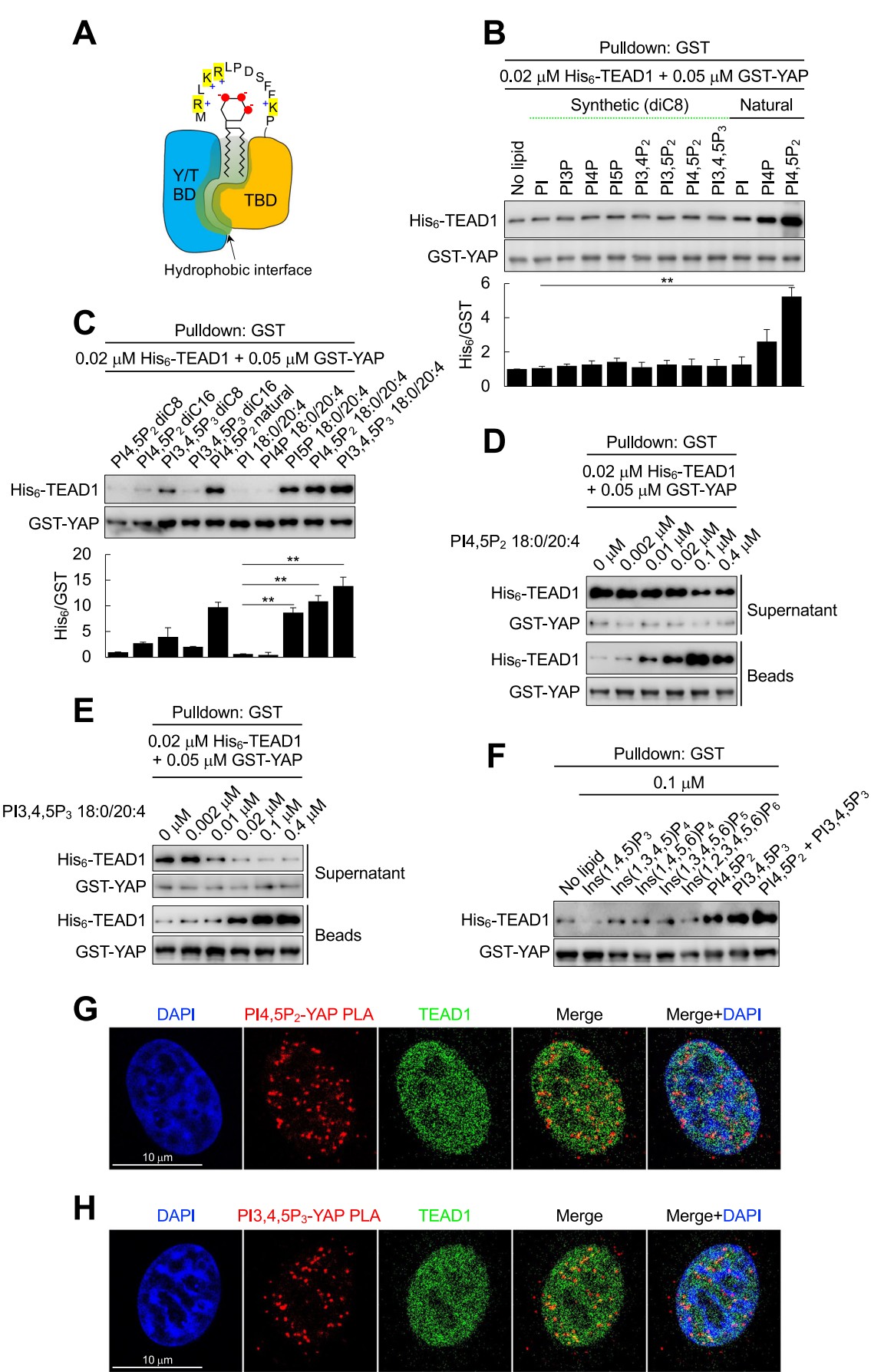

Figure 5. PI(4,5)P$_2$ and PI(3,4,5)P$_3$ facilitate the interaction of YAP with the TEADs.

(A) A schematic representation of phosphoinositide-facilitated binding of YAP/TAZ with TEADs is shown. The negatively charged inositol head group of the phosphoinositides binds with a polybasic motif (PBM) in the TEAD-binding domain (TBD) of YAP/TAZ. The acyl chains of the phosphoinositides interact with the hydrophobic interface formed between the YAP/TAZ-binding domain (Y/TBD) of TEADs and the TBD of YAP/TAZ. (B, C) Overall, 0.02 μM His$_6$-TEAD1 was incubated with 0.05 μM GST-YAP in the absence or presence of the indicated lipids (0.4 μM). The structures of the synthetic phosphoinositides are shown in Fig. EV3E. The natural phosphoinositides were from the bovine brain. GST-YAP was pulled down with glutathione beads and the associated TEAD1 was analyzed by immunoblotting. The graph shows the mean ± s.d. of $n = 3$ independent experiments. Natural PI(4,5)P$_2$ and the synthetic 18:0/20:4 PI(4,5)P$_2$ and PI(3,4,5)P$_3$ enhanced the binding of TEAD1 to YAP in vitro. *$P < 0.05$; **$P < 0.01$, and n.s.; not significant in Student's $t$ test. (D, E) In total, 0.02 μM His$_6$-TEAD1 was incubated with 0.05 μM GST-YAP in the absence or presence of the indicated concentrations of 18:0/24:4 PI(4,5)P$_2$ (D) or PI(3,4,5)P$_3$ (E). GST-YAP was pulled down with glutathione beads and the associated TEAD1 was analyzed by immunoblotting. Representative immunoblot images of $n = 3$ independent experiments are shown. (F) In all, 0.02 μM His$_6$-TEAD1 was incubated with 0.05 μM GST-YAP in the absence or presence of the indicated inositol phosphates and phosphoinositides (0.1 μM). GST-YAP was pulled down with glutathione beads and the associated TEAD1 was analyzed by immunoblotting. Representative immunoblot images of $n = 3$ independent experiments are shown. (G, H) Starved MDA-MB-231 cells were stimulated with 10% serum for 1 h. Cells were fixed and the association of YAP with PI(4,5)P$_2$ (G) and PI(3,4,5)P$_3$ (H) was visualized by PLA. Colocalization of endogenous TEAD1 with the PLA signals was analyzed by immunostaining. The images were obtained by confocal microscopy. Representative images of $n = 3$ independent experiments are shown. Scale bar, 10 μm. Source data are available online for this figure.

YAP by ectopic expression of the ORF-expressing constructs and cell motility was measured. Knockdown of YAP dramatically reduced CTGF expression. In addition, serum, laminin, vitronectin-directed cell migration was significantly attenuated. Re-expression of WT YAP fully rescued CTGF expression and cell migration, while the 2Q and 3Q mutants had no significant impact (Fig. 7A; Appendix Fig. S2B). In contrast, neither knockdown of YAP nor rescue with WT YAP, 2Q, and 3Q affected apoptosis or proliferation (Fig. 7B,C). These data support that the phosphoinositide-directed YAP/TAZ–TEAD pathway specifically regulates breast cancer cell motility without compromising proliferation or promoting apoptosis.

It has been documented that the YAP/TAZ–TEAD pathway is critical for cancer cell motility and metastasis in many types of cancers including breast cancer (Lamar et al, 2012; Warren et al, 2018; Zanconato et al, 2016b), however, the specific YAP/TAZ targets involved in cancer cell motility are not well-established. We next tested if CTGF and CYR61 are the key YAP/TAZ targets modulating breast cancer cell motility as the expression of CTGF and CYR61 faithfully correlated with YAP/TAZ function compared to the other targets. Knockdown of CTGF and CYR61 significantly reduced breast cancer cell motility (Fig. EV4C; Appendix Fig. S2C). Collectively, the data demonstrate that the PIPKIα/IPMK→PI(4,5)P$_2$/PI(3,4,5)P$_3$→YAP/TAZ→TEAD pathway regulates breast cancer cell motility by controlling the expression of CTGF and CYR61.

PIPKIα overexpression is reported in breast and other cancer types (Sarwar et al, 2019; Waugh, 2014), while the implications of IPMK in cancer are largely unknown. To further study the clinical implication of PIPKIα and IPMK in breast cancer, we first analyzed the expression of PIPKIα and IPMK mRNA using the databases of normal breast (GTEx) (Consortium, 2013) and breast cancer (TCGA) (Cancer Genome Atlas, 2012) tissues. Compared to normal tissue, the levels of PIPKIα and IPMK mRNA were significantly elevated in breast tumor tissues (Fig. EV4D). In the TCGA breast cancer carcinoma samples ($n = 106$), we found that the expression of PIPKIα and IPMK mRNA is statistically significantly correlated with the expression of CTGF (but not CYR61), although the correlation was moderate (Fig. EV4E). Next, the protein levels of PIPKIα and IPMK were analyzed by tissue microarray using commercially available tissues (US Biomax). We used 6 normal breast and 98 breast tumor (breast cancer stage 2A, 2B, 3A, and 3B/3C) biopsy tissues and stained the tissues with IgG control, anti-PIPKIα, and anti-IPMK antibodies. In each tissue, the

breast tumor was histologically located by H&E staining, and the staining of PIPKIα and IPMK in the tumor area was scored by a 0, 1, 2, and 3 scale (Appendix Fig. S3A,S3B). We found that ~76% (75/98) and ~71% (69/97) of breast tumor tissues show moderate or strong staining with antibodies to PIPKIα and IPMK, respectively. In contrast, PIPKIα and IPMK staining in the six normal breast tissues were negative or weak (Fig. 7D). Breast cancer can be classified into subtypes based on the expression of estrogen receptor (ER), progesterone receptor (PR), and human epidermal growth factor receptor 2 (HER2) (Onitilo et al, 2009). We found that the protein expression levels of PIPKIα and IPMK did not correlate with ER-positive, PR-positive, HER2-positve, or triple-negative breast cancer. In summary, PIPKIα and IPMK expression is elevated in breast cancer at both the mRNA and protein levels.

## Discussion

In this study, we identified phosphoinositide-driven signaling mechanisms by which the binding of YAP/TAZ to the TEADs is spatiotemporally regulated by nuclear phosphoinositides generated by PIPKIα and IPMK. Upon agonist activation of membrane receptors including integrins, RTKs, and GPCRs, PIPKIα, and IPMK inducibly associate with YAP/TAZ via LPXY motifs on PIPKIα and IPMK and the WW domains on YAP. YAP/TAZ translocate into the nucleus and form a complex with PIPKIα and IPMK. Then, PIPKIα and IPMK generate PI(4,5)P$_2$ and PI(3,4,5)P$_3$, and the generated PI(4,5)P$_2$ and PI(3,4,5)P$_3$ are transferred to and bind to YAP/TAZ. The binding of PI(4,5)P$_2$ and PI(3,4,5)P$_3$ to YAP/TAZ facilitates their association with the TEADs, leading to the transcription of the YAP/TAZ target genes, including CTGF and CYR61. Activation of this nuclear phosphoinositide-mediated YAP/TAZ–TEAD pathway controls breast cancer cell motility (Fig. 7E).

The existence and functions of phosphoinositides in non-membranous nuclear compartments have been enigmatic until recently. We and others have shown that substantial amounts of phosphoinositide species, including PI(4)P, PI(5)P, PI(4,5)P$_2$, and PI(3,4,5)P$_3$ are present in the nucleus, and their nuclear content is regulated by stimuli and stress (Boronenkov et al, 1998; Chen et al, 2020, 2022; Choi et al, 2019b; Faberova et al, 2020; Poli et al, 2019). The majority of nuclear phosphoinositides have been shown to locate in the nuclear speckles where genes are actively transcribed (Choi et al, 2019b; Faberova et al, 2020; Ha, 2020; Sobol et al, 2018),

## A

YAP domain structure: 1 — 50 — TBD — PBM (red) — 100 — 171 — WW — WW — 263 — TAD — 504

YAP WT  86-M**R**L**RK**LPDSFF**K**P-98
YAP 1Q  86-M**R**L**RK**LPDSFF**Q**P-98
YAP 2Q  86-M**Q**L**RQ**LPDSFF**K**P-98
YAP 3Q  86-M**Q**L**RQ**LPDSFF**Q**P-98

## B

GST-YAP (WT, 1Q, 2Q, 3Q) — Coomassie — -100, -70, MW (kDa)

PI4,5P$_2$ — PI3,4,5P$_3$

GST-YAP (WT, 3Q, 2Q, 1Q | WT, 3Q, 2Q, 1Q)

GST-YAP — Supernatant
GST-YAP — Beads

## C

Pulldown: GST
0.02 μM His$_6$-TEAD1 + 0.05 μM GST-YAP
18:0/20:4    PI    PI5P    PI4,5P$_2$    PI3,4,5P$_3$
GST-YAP  WT 2Q 3Q  WT 2Q 3Q  WT 2Q 3Q  WT 2Q 3Q

His$_6$-TEAD1
GST-YAP

Graph: His$_6$/GST (y-axis 0–50); * and ** significance marks

## D

Myc-TEAD4

Input — IP: Myc
Flag-YAP: Mock, WT, 2Q, 3Q | Mock, WT, 2Q, 3Q

Flag-YAP
Myc-TEAD4

## E

HA-p73

Input — IP: HA
Flag-YAP: Mock, WT, 2Q, 3Q | Mock, WT, 2Q, 3Q

Flag-YAP
HA-p73

## F

HEK293

Flag-YAP transfection: Mock, WT, 2Q, 3Q

CTGF
Flag-YAP
Actin

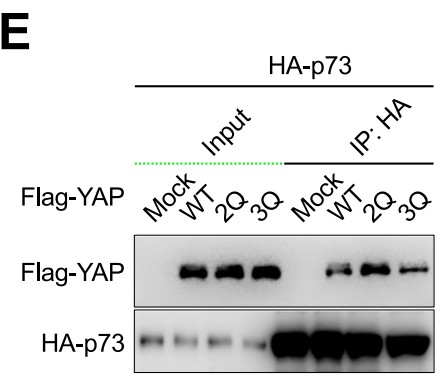
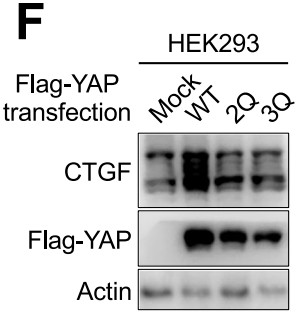

but the exact mechanisms how nuclear phosphoinositides regulate gene expression remain largely unexplored. Here we report that the nuclear-localized phosphoinositides PI(4,5)P$_2$ and PI(3,4,5)P$_3$ generated by PIPKIα and IPMK in the proximity of YAP/TAZ function as critical cofactors mediating YAP/TAZ binding to the TEADs, leading to the expression of YAP/TAZ targets (Figs. 1–5).

The cytoplasmic-nuclear shuttling and the binding of the TEADs to YAP/TAZ are tightly regulated by phosphorylation. In response to intrinsic and extrinsic signals such as cell–cell contact, stress signals, and cell polarity, the Hippo kinases LATS1/2 phosphorylate YAP at several serine/threonine residues including S127 and S381, resulting in its cytoplasmic sequestration and

**Figure 6. The binding of phosphoinositides to YAP controls its association with the TEADs and the expression of the YAP target gene CTGF.**

(A) The schematic diagram shows the location of the PBM in YAP. The amino acid sequences of the WT PBM and the phosphoinositide-binding-defective mutants 1Q, 2Q, and 3Q are shown. (B) In all, 0.5 μM WT GST-YAP and the 1Q, 2Q, and 3Q mutants were incubated with 10 volume % of the indicated PIP Beads (Echelon Biosciences). After washing, the amount of YAP bound to the phosphoinositide beads and remained unbound in supernatant were analyzed by immunoblotting. Representative immunoblot images of $n = 3$ independent experiments are shown (bottom). A representative Coomassie-stained image of the recombinant proteins used is shown (top). The binding of $PI(4,5)P_2$ and $PI(3,4,5)P_3$ to 1Q and 2Q mutants was partially reduced, while the 3Q binding was completed abrogated compared to WT YAP. (C) Overall, 0.02 μM His$_6$-TEAD1 was incubated with 0.05 μM GST-YAP in the absence or presence of the indicated synthetic lipids (0.4 μM). YAP was pulled down with glutathione beads and the associated TEAD1 was analyzed by immunoblotting. The graph shows mean±s.d. of $n = 3$ independent experiments. In the presence of $PI(4,5)P_2$ and $PI(3,4,5)P_3$, TEAD1 associated with WT YAP but not the mutants. *$P < 0.05$; **$P < 0.01$, and n.s.; not significant in Student's $t$ test. (D) Flag-tagged WT YAP and the 2Q and 3Q mutants were co-transfected with Myc-TEAD4 in HEK293 cells. Exogenous TEAD4 was immunoprecipitated and the associated YAP proteins were analyzed by immunoblotting with an anti-Flag antibody. Representative immunoblot images of $n = 3$ independent experiments are shown. Only WT YAP co-immunoprecipitated with TEAD4. (E) Flag-tagged WT YAP and the 2Q and 3Q mutants were co-transfected with HA-p73 in HEK293 cells. Exogenous p73 was immunoprecipitated and the associated YAP proteins were analyzed by immunoblotting with an anti-Flag antibody. Representative immunoblot images of $n = 3$ independent experiments are shown. All the forms of YAP associated with p73. (F) Flag-tagged WT YAP and the 2Q and 3Q mutants were transfected in HEK293 cells. Expression of endogenous CTGF was measured by immunoblotting. Representative immunoblot images of $n = 3$ independent experiments are shown. The 2Q and 3Q mutants are impaired in their ability to promote the expression of CTGF. Source data are available online for this figure.

degradation by the proteasomal pathway (Boopathy and Hong, 2019; Dey et al, 2020; Zhao et al, 2010). Upon cellular energy deprivation, AMP-activated protein kinase (AMPK) phosphorylates YAP at S61 and S94 residues, and the AMPK-mediated phosphorylation of YAP inhibits its binding with the TEADs (Mo et al, 2015; Wang et al, 2015). Importantly, one of the AMPK phosphorylation sites on YAP (S94) is in the PBM that binds phosphoinositides (Fig. 6A). We speculate that during energy scarcity (such as serum starvation) phosphorylation of YAP/TAZ by AMPK (at S94 residue for YAP) prevents phosphoinositides from binding by electrostatic hindrance, leading to improper binding of YAP/TAZ to the TEADs. In contrast, in energy surplus (such as serum stimulation) or agonist stimulation, AMPK activity will be suppressed and phosphoinositides become docked on the unphosphorylated PBMs to facilitate the binding of YAP/TAZ with the TEADs. Consistently, we observed that the association of YAP/TAZ with PIPKIα, IMPK, $PI(4,5)P_2$, and $PI(3,4,5)P_3$ are enhanced by serum and agonist stimulation (Figs. 2–4, EV1, and EV2).

In this study, we report another layer of regulation which is required for the full activation of the YAP/TAZ–TEAD pathway in cancer. Importantly, the mechanisms we discovered are specifically regulated by a few specific phosphoinositides and their generating enzymes rather than a global impact by the perturbation of phosphoinositides. PIPKIα is found in the nucleus and regulates the nuclear functions of p53 and AKT (Chen et al, 2022). Nuclear $PI(4,5)P_2$ generated by PIPKIα is tightly associated with p53 and resistant to SDS-PAGE and further phosphorylated by IMPK to activate AKT upon DNA damage stress (Chen et al, 2022; Choi et al, 2019b). Consistently, we showed that a *myo*-inositol probe that can be incorporated into phosphoinositides becomes tightly (resistant to SDS-PAGE) associated with the YAP/TAZ-TEADs complex (Figs. 4 and EV2). We further envision that phosphatases which dephosphorylate $PI(4,5)P_2$ and $PI(3,4,5)P_3$ may turn off the phosphoinositide-mediated activation of YAP/TAZ. In support of this hypothesis, the inactivation of the $PI(3,4,5)P_3$-specific phosphatase PTEN is reported to enhance YAP nuclear localization and target gene expression in gastric cancer (Xu et al, 2018). Another signaling lipid, phosphatidic acid (PA), has recently been shown to regulate the YAP/TAZ pathway. PA directly binds to the Hippo components LATS1/2 and NF2 in the cytoplasm and negatively regulates the Hippo kinase cascade leading to YAP/TAZ activation (Han et al, 2018). Interestingly, PA activates the kinase activity of

PIPKIα (Jenkins et al, 1994). Further, PI transfer proteins, PI4KA, and PIPKIγ activate the YAP/TAZ pathway by controlling the levels of PI, PI(4)P, and $PI(4,5)P_2$, respectively, in the plasma membrane (Li et al, 2022). The regulation of the YAP/TAZ pathway by these cytoplasmic and plasma membrane-oriented lipid signaling pathways is distinct from our nucleus-oriented mechanism.

Although not a focus here, we observed that PI(4)P and PI(5)P are also bound to YAP or mediate YAP/TAZ–TEAD interactions in vitro (Figs. 4A, 5C, 6C and EV3D), suggesting that PI(4)P and PI(5)P might be involved in YAP/TAZ target gene expression. Further studies are needed to investigate the roles of nuclear PI(4)P and PI(5)P and the enzymes that generate these phosphoinositides in cancer. PI(5)P is the least abundant phosphoinositide species only accounting for ~0.5% of total phosphoinositides (Hasegawa et al, 2017). By comparison, $PI(4,5)P_2$ is the most abundant phosphoinositide species in most cells (Di Paolo and De Camilli, 2006), and $PI(3,4,5)P_3$ levels are sharply increased by agonist stimulation (Traynor-Kaplan et al, 1989). Cellular PI(5)P level is largely regulated via synthesis by PIKfyve (encoded by *PIKFYVE* gene) and turnover by type II phosphatidylinositol phosphate kinases (encoded by *PIP4K2A*, *PIP4K2B*, and *PIP4K2C* genes) (Emerling et al, 2013; Hasegawa et al, 2017). Considering the very low cellular concentration of PI(5)P, YAP could associate with PI(5)P metabolizing enzymes if PI(5)P is a key regulator of the YAP/TAZ–TEAD pathway. This mechanism ensures the generation of phosphoinositide signals are efficiently utilized by phosphoinositide effectors (Choi et al, 2015; Tan et al, 2015b). To investigate potential roles of PI(5)P in the YAP/TAZ–TEAD pathway, we tried to find the LPXY or PPXY motifs on PI(5)P metabolizing enzymes. As shown in Fig. EV1F, the LPXY or PPXY motifs were reported to mediate YAP/TAZ interactions in many YAP/TAZ-binding proteins (Ma et al, 2019; Moroishi et al, 2015). We found that PIKfyve, PIP4K2A, PIP4K2B, and PIP4K2C do not contain LPXY or PPXY motifs. These results suggest that the cellular contribution of PI(5)P in the regulation the YAP–TEAD signaling might be limited (compared to $PI(4,5)P_2$ and $PI(3,4,5)P_3$) although it potentially could play a role in vitro.

IMPK has both inositol phosphate and phosphoinositide kinase activities. The kinase-dead mutant (Fu et al, 2018; Maag et al, 2011) used in Fig. 3F is defective of both kinase activities, thus the result does not answer which IMPK product is responsible for the

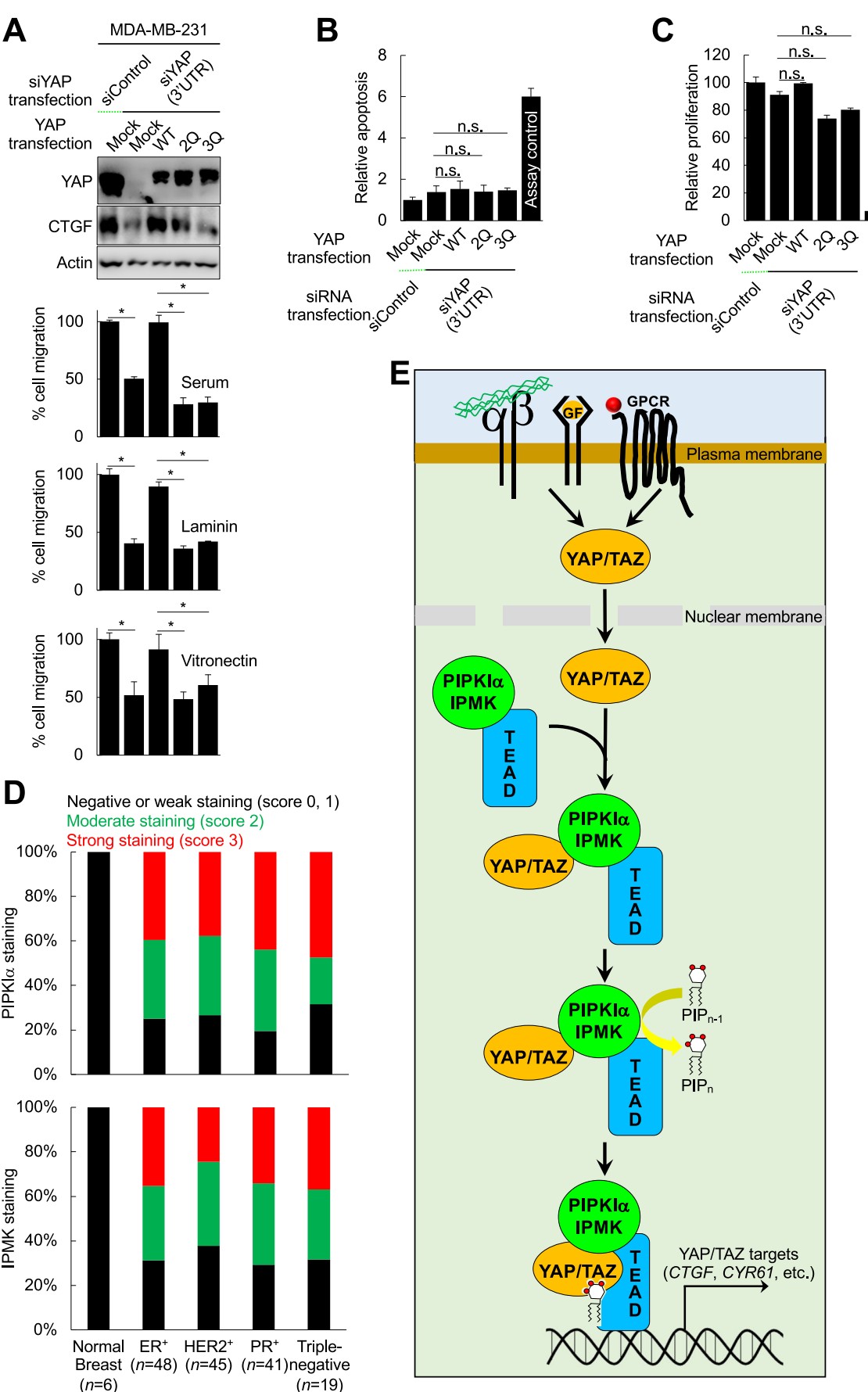

◄

**Figure 7.  The binding of phosphoinositides to YAP controls breast cancer cell motility.**

(A–C) Endogenous YAP was depleted in MDA-MB-231 by siRNAs. After 24 h WT YAP and the 2Q and 3Q mutants were then ectopically expressed for an additional 48 h. The expression of CTGF was determined by immunoblotting. Only WT YAP fully rescued the expression of CTGF. Serum, laminin, and vitronectin-induced cell migration was measured using Transwell inserts (A). Only WT YAP fully restored the migration rates. The relative apoptosis of the cells shown in (C) was measured using the Caspase-Glo assay (Promega). As an assay control, 1 μM staurosporine was used (B). None of the constructs promoted apoptosis. The relative proliferation of the cells was measured using the Cell-Titer Glo assay. As an assay control, 1 μM staurosporine was used (C). None of the constructs significantly altered the proliferation rate nor apoptosis. The graphs show the mean±s.d. of $n = 3$ independent experiments. *$P < 0.05$; **$P < 0.01$, and n.s.; not significant in Student's $t$ test. (D) Normal and breast cancer biopsy tissues were stained with anti-PIPKIα and anti-IPMK antibodies. The intensity of staining was histologically scored using a scale of 0, 1, 2, and 3 (Appendix Fig. S3A,S3B). The breast cancers were divided into subtypes ER[+], HER2[+], PR[+], and triple-negative. The graph shows the distribution of the scores. The expression of both PIPKIα and IPMK was higher in all the subtypes of breast cancer. (E) A schematic representation of how PIPKIα and IPMK regulate the YAP/TAZ–TEAD pathway in breast cancer. In response to stimulation via integrins (αβ), growth factor receptors (GF), and G-protein coupled receptors (GPCR), YAP/TAZ translocate into the nucleus. In the nucleus, PIPKIα and IPMK form a complex with YAP/TAZ. In the complex, phosphoinositides generated by PIPKIα and IPMK bind to and in turn facilitate the association of YAP/TAZ with the TEADs. The nuclear phosphoinositides stimulate the expression of YAP/TAZ target genes, promoting breast cancer cell motility. Source data are available online for this figure.

regulation of the YAP/TAZ–TEAD pathway. To this end, we performed in vitro binding assays between YAP and TEAD1 in the presence of all known IPMK enzymatic products. As shown in Fig. 5F, $Ins(1,4,5)P_3$, $Ins(1,3,4,5)P_4$, $Ins(1,4,5,6)P_4$, $Ins(1,3,4,5,6)P_5$, and $Ins(1,2,3,4,5,6)P_6$ at 0.1 μM concentration had no detectable impact on facilitating TEAD1 binding to YAP in vitro, while $PI(4,5)P_2$ and $PI(3,4,5)P_3$ dramatically increased the binding. Since the cellular concentrations of the inositol phosphates are in the range of 10–50 μM (Qiu et al, 2020), in future studies we plan to further explore the potential contribution of inositol phosphates in mediating the YAP–TEAD interaction in cells.

The regulation of the YAP/TAZ–TEAD pathway by nuclear phosphoinositides has critical implications for cancer. The YAP/TAZ–TEAD pathway has fundamental roles in the initiation and progression of breast and other cancer types, leading to extensive efforts to target the pathway in cancer. Unfortunately, current agents that disrupt YAP/TAZ signaling, including small molecule inhibitors and peptide mimetics, have been disappointing in clinical trials likely due to their low solubility, nonspecific side effects, and poor pharmacokinetics (Dey et al, 2020). Our current findings underscore that blockade of the nuclear phosphoinositide pathway may be a novel strategy to suppress the YAP/TAZ–TEAD pathway in cancer. Indeed, depletion of PIPKIα and IPMK attenuated the binding of YAP/TAZ with the TEADs and the expression of target genes in breast cancer cell lines (Figs. 1, 2, 3, 7 and EV1 and EV2). In addition, PIPKIα and IPMK expression was significantly elevated and correlated with CTGF expression in breast cancer clinical samples, making PIPKIα and IPMK promising drug targets in YAP/TAZ-driven breast and other cancers.

# Methods

## Cell lines and constructs

MDA-MB-231, MDA-MB-468, and HEK293 cells were purchased from the ATCC (American Type Culture Collection) and maintained in DMEM supplemented with 10% fetal bovine serum (FBS). All of the cell lines were routinely tested for mycoplasma contamination and mycoplasma-negative cells were used in the study. None of the cell lines used in this study is listed in the database of commonly misidentified cell lines maintained by the ICLAC (International Cell Line Authentication Committee).

The PIPKIα (HA-tagged WT and KD mutant and His$_6$-tagged WT) (Chen et al, 2022; Choi et al, 2016, 2019b), IPMK (Myc-tagged WT and KD mutant and His$_6$-tagged WT) (Maag et al, 2011; Wang and Shears, 2017), YAP (Flag and GST-tagged WT and Flag-tagged mtWW) (Kim et al, 2020), TAZ (GST-tagged) (Kim et al, 2020), TEAD1 (His$_6$-tagged) (Kim et al, 2020), p73 (HA-tagged) (Jost et al, 1997), and TEAD4 (Myc-tagged) (Li et al, 2010) constructs used for this work have been described previously. The HA-LPGAA mutant PIPKIα, Myc-LPKAA mutant IPMK, and Flag and GST-2Q and 3Q mutant YAP constructs were generated by site-directed mutagenesis and validated by DNA sequencing in University of Nebraska Medical Center Genomics Core. Constructs were transfected into mammalian cells by the lipid-based delivery system from Invitrogen (Lipofectamine 3000) according to the manufacturer's instructions. Typically, 2–5 μg DNA and 6–10 μl lipids were used for transfecting cells in six-well plates. In all transient expression experiments, a green fluorescent protein construct was transfected in parallel to monitor the transfection efficiency. Cells that had at least 80% transfection efficiency were used for further analysis.

## CRISPR-Cas9-mediated KO cell line generation

To stably KO the genes encoding PIPKIα and IPMK in MDA-MB-231, the CRISPR-Cas9 genome-editing method was used (Ran et al, 2013). Guide RNA sequences (5′-GAAAAACCGACCGAA-GACGA-3′ and 5′-CTATGGACTGTACTGTGTGC-3′ for the gene encoding PIPKIα and 5′-CATGTACGGGAAGGACAAAG-3′ and 5′- GACCAGATGCCATAATATTT-3′ for the gene encoding IPMK) were cloned into the PX459V2.4 vector (Ran et al, 2013). Constructs were transfected for 36 h and then transiently selected with 1 μg/ml puromycin. After incubation for 48 h, puromycin was removed and single cells were seeded in 96-well tissue culture dishes. Cells were expanded and positive colonies were selected by immunoblotting with specific antibodies against PIPKIα and IPMK. As negative controls, empty vector-transfected cells or parental non-transfected cells were used. To generate YAP KO cells, guide RNAs against human YAP1 gene (CRISPR772157_SGM and CRISPR504070_SGM, Thermo Fisher Scientific) were co-transfected with GFP-tagged Cas9 plasmid vector (sc-418922, Santa Cruz Biotechnology). GFP-positive cells were sorted into 96-well tissue culture dishes. Cells were expanded and positive colonies were selected by immunoblotting with specific antibodies against YAP.

## Antibodies and reagents

Monoclonal and polyclonal antibodies against YAP (clone D8H1X), $^{pS127}$YAP (clone D9W2I), CTGF (clone D8Z8U), CYR61 (clone D4H5D), p53 (clone 7F5), the HA tag (clone 6E2), the Myc tag (clone 9B11), the Flag tag (clone D6W5B), AXL (C89E7), pan-TEAD (clone D3F7L), YAP/TAZ (clone E9M8G), p110α (clone C73F8), p110β (clone C33D4), PIPKIα (9693), and PIPKIγ (3296) were purchased from Cell Signaling Technology. Other commercially available antibodies against actin (clone C4, MP Biomedicals), the GST tag (clone RPN1236V, GE Healthcare), and the His$_6$ tag (clone HIS.H8, Sigma-Aldrich) were purchased from the indicated companies. Commercially available polyclonal antibodies against human IPMK (PA5-21629, Thermo Fisher Scientific; HPA037837, Sigma-Aldrich) were purchased and used for this study. Polyclonal and monoclonal antibodies against PIPKIα, PIPKIγ, and IPMK were produced as described previously (Chen et al, 2022; Choi et al, 2013, 2016, 2019b). A homemade IPMK antibody was gifted from Dr. Seyun Kim's laboratory (KAIST, South Korea). For conventional immunostaining and PLA analysis of phosphoinositides, anti-PI(3)P (Z-P003), anti-PI(4)P (Z-P004), anti-PI(3,4)P$_2$ (Z-P034), anti-PI(4,5)P$_2$ (P-Z045) and anti-PI(3,4,5)P$_3$ (Z-P345) antibodies were purchased from Echelon Biosciences. For immunostaining analyses and PLA, antibodies were diluted in a 1/100 ratio. Pooled short interfering RNAs (siRNAs) against human PIPKIα, PIPKIγ, IPMK, YAP, TAZ, p53, CTGF, and CYR61 were purchased from Dharmacon (ON-TARGETplus). For the knockdown/rescue experiments, siRNAs targeting sites in the 3′ untranslated regions of PIPKIα (5′-UGACUCCUGGAAGAAUA-CUCCUGUA-3′), IPMK (5'-CCAAGAGAGCUGGAAUUCUAU AAUA-3'), and YAP (5'-GCUUAUAAGGAUGAGACAUU-3') were purchased from Thermo Fisher Scientific. Non-targeting siRNA (Dharmacon) was used as a control. siRNAs were delivered to cells by RNAiMAX reagent (Thermo Fisher Scientific) and knockdown efficiency was determined by immunoblotting. Knockdown efficiency >85% was required to observe phenotypic changes in the study. The PIPKIα inhibitor ISA-2011B and AT9283 were purchased from Sellekchem and dissolved in DMSO to a 20 mM working solution. Synthetic and natural phosphoinositides were purchased from Avanti Polar Lipids or Echelon Biosciences and used as detailed in Fig. EV3E. Natural lipids PI (bovine liver, 840042), PI(4)P (porcine brain, 840045), and PI(4,5)P$_2$ (porcine brain, 840046) and synthetic lipids 18:0/20:4 PI (850144), 18:0/20:4 PI(4)P (850158), 18:0/20:4 PI(5)P (850190), 18:0/20:4 PI(4,5)P$_2$ (850165), and 18:0/20:4 PI(3,4,5)P3 (850166) were purchased from Avanti Polar Lipids. Synthetic lipids diC8 PI (P-0008), diC8 PI(3)P (P-3008), diC8 PI(4)P (P-4008), diC8 PI(5)P (P-5008), diC8 PI(3,4)P$_2$ (P-3408), diC8 PI(3,5)P$_2$ (P-3508), diC8 PI(4,5)P$_2$ (P-4508), diC8 PI(3,4,5)P$_3$ (P-3908), diC16 PI(4,5)P$_2$ (P-4516), and diC16 PI(3,4,5)P$_3$ (P-3916) were purchased from Echelon Bioscience.

## Immunoprecipitation and immunoblotting

Cells were lysed in a buffer containing 1% Brij58, 150 mM NaCl, 20 mM HEPES, pH 7.4, 2 mM MgCl$_2$, 2 mM CaCl$_2$, and protease and phosphatase inhibitor cocktails (Sigma-Aldrich). The protein concentration of lysates was measured by the BCA method (Thermo Fisher Scientific) and equal amounts of protein were used for further analysis. For immunoblotting analyses, antibodies were diluted in a 1/1000 ratio except for p53 (clone 7F5, 1/5000 dilution) and actin (clone C4, 1/500,000 dilution). For immunoprecipitation, 1–2.5 mg protein was incubated with 2 μg antibodies at 4 °C for overnight and then incubated with a 50% slurry of Protein G Sepharose (GE Life Sciences) for another 2 h. After washing five times with lysis buffer, the protein complexes were eluted with SDS sample buffer. For immunoblotting, 5–20 μg proteins were loaded. For immunoblotting of immunoprecipitated complexes, horseradish peroxidase (HRP)-conjugated primary antibodies were used to avoid nonspecific detection of immunoglobulin in immunoprecipitation samples. HRP-conjugated anti-Flag tag (86861), HA tag (14031), p53 (32532), Myc tag (2040), c-Myc (18583), YAP/TAZ (28287), and YAP (15028) antibodies were purchased from Cell Signaling Technology. HRP-conjugated anti-pan-TEAD, PIPKIα, IPMK, and PI(4,5)P$_2$ (P-Z045 antibodies were generated by a HRP Conjugation Kit (ab102890, Abcam) according to the manufacturer's instructions. Immunoblots were developed by the Odyssey Imaging System (LI-COR Biosciences) and the intensity of protein bands was quantified using ImageJ. Unsaturated exposures of immunoblot images were used for quantification with the appropriate loading controls (e.g., actin) as standards. Statistical analysis of the data was performed with Microsoft Excel, using data from at least three independent experiments.

## Metabolic labeling and click chemistry

Ac$_3$2API was reported previously (Ricks et al, 2019). MDA-MB-231 cells were starved with an inositol-free OPTI-MEM (Thermo Fisher Scientific) for 24 h. Starved cells were treated with 50 μM Ac$_3$2API in the presence of 10% dialyzed serum (Thermo Fisher Scientific) for 24 h. Cells were harvested with the 1% Brij58-containing lysis buffer and diluted lysates (less than 0.5 mg/ml total protein) were further utilized for click chemistry reaction. The conjugation of azide-tagged molecules resulting from probe treatment to biotin was carried out via click chemistry reaction as described previously (Schiapparelli et al, 2014) with some modifications. Briefly, an alkyne-biotin (Sigma-Aldrich) was added to cell lysates with a 20 μM final concentration. THPTA (Sigma-Aldrich, 10 mM final concentration) was added to lysates and briefly vortexed to mix. CuSO$_4$ (Sigma-Aldrich, 2 mM final concentration) was added to lysates and briefly vortexed to mix. Sodium ascorbate (Sigma-Aldrich, 20 mM final concentration) was added to lysates and briefly vortexed to mix. Lysates were incubated in dark for 30 min at room temperature to allow click chemistry reaction. The click'ed lysates were further utilized for IP with anti-YAP, pan-TEAD, c-Myc, and p53 antibodies and resolved by SDS-PAGE. As a negative control, no alkyne-biotin was added in lysates.

## In vitro binding and kinase assay

Recombinant proteins were expressed in the BL21 *Escherichia coli* strain. GST-tagged proteins were then purified with glutathione Sepharose 4B (GE Life Sciences) and His$_6$-tagged proteins were purified with His-Bind Resin (Novagen). GST-tagged proteins were incubated with glutathione beads before being used in the binding assays. 50 nM to 10 μM His$_6$-tagged proteins were incubated with the indicated concentrations of GST-tagged proteins prebound to glutathione beads. After incubation for 1 h at 25 °C, unbound

proteins were washed out and the protein complexes were analyzed by immunoblotting. In vitro binding assays with the phospholipids (Figs. 5 and 6) were performed in tris-buffered saline (TBS) with 0.005% Triton X-100. Note, the concentration of Triton X-100 and phospholipids used in these studies are below their critical micelle concentrations and thus the phospholipids are solubilized in the buffer without forming micelles. Detailed assay conditions are described in Fig. EV3C. In vitro kinase assays were performed using the ADP-Glo Kinase Assay kit (V9101, Promega) according to the manufacturer's instructions. Briefly, 0.05 μM His$_6$-tagged PIPKIα or IPMK was incubated with 0.2 μM diC8 PI(4)P (P-4008) or PI(4,5)P$_2$ (P-4508, Echelon Biosciences), respectively, in the absence or presence of various concentrations of GST-YAP (0.001, 0.01, 0.1, 1.0, and 10.0 μM). After a 15 min incubation at 30 °C, the ADP-Glo Reagent was added to terminate the kinase reaction and deplete the remaining ATP, and then the Kinase Detection Reagent was added to convert ADP to ATP and to measure the newly synthesized ATP using the luciferase/luciferin reaction.

## Immunofluorescence microscopy and PLA

For immunofluorescence microscopy, glass coverslips were coated with 10 μg /ml collagen, 10 μg/ml fibronectin, 0.2% gelatin, or 10% serum before seeding cells. Cells were grown on coverslips placed inside six-well plates until experimental manipulation. Cells were rapidly fixed by adding an equal volume of 8% paraformaldehyde and 0.5% glutaraldehyde to the tissue culture medium for 15 min at room temperature. After a 30-min wash with PBS containing 50 mM NH$_4$Cl, cells were permeabilized and blocked with a solution of buffer A (20 mM PIPES, pH 6.8, 135 mM NaCl, and 5 mM KCl) containing 0.5% saponin and 5% FBS for 45 min at room temperature. Primary antibodies (2–4 μg/ml) were incubated in a solution of buffer A containing 0.1% saponin and 5% of FBS for 12 h at 4 °C. After a 30-min wash with buffer A, fluorophore-conjugated secondary antibodies were incubated in a solution of buffer A containing 0.1% saponin and 5% of FBS for 1 h at room temperature. Then the cells were washed with buffer A for 45 min at room temperature before post-fixation with 2% paraformaldehyde and 0.125% glutaraldehyde for 10 min at room temperature. Coverslips were washed five times with PBS containing 50 mM NH$_4$Cl and once with distilled water. Epifluorescence microscopy was performed using a ×20 or ×40 plan-fluor objective on a EVOS M7000 (Thermo Fisher Scientific).

For PLA, after fixing, the cells were processed (DUO92101, Sigma-Aldrich) according to the manufacturer's instruction. Post PLA, the slides were further processed for immunofluorescence staining using an Alexa Fluor 488-conjugated anti-TEAD1 antibody (sc-393976 AF488, Santa Cruz Biotechnology). The slides were mounted with Duolink In Situ Mounting Medium with DAPI (DUO82040, Sigma-Aldrich). The images were collected with a Zeiss LSM 800 with Airyscan confocal microscope in the University of Nebraska Medical Center Advanced Microscopy Core Facility and analyzed by ImageJ. The number of PLA puncta or foci in the nuclei, using the DAPI signal to define the region of interest, was used for the quantification. Ten cells were quantified for each group. The quantitative graph was generated by Microsoft Excel.

## Cell growth and survival assay

A total of $5 \times 10^3$ cells/well were plated in 96-well plates in complete medium for 24 h. Cell number and caspase activation/apoptosis were measured using CellTtiter-GLO (G9243) and Caspase-GLO 3/7 (G8092, Promega) according to the manufacturer's instructions.

## Migration assay

The bottom surface of Transwell filters (8-μm pores) was coated with 10% serum, 10 μg/ml laminin, 10 μg/ml vitronectin, or 25 μg/ml collagen IV. Cells were placed in the upper chamber and the cultures were incubated for 16 h at 37 °C. Cells on the bottom of the filter were fixed with 4% paraformaldehyde diluted in PBS and stained with 0.1% Crystal Violets and DAPI. Five random fields were imaged, and the number of stained cells was counted. Statistical analysis was performed using data from at least three independent experiments.

## Measurement of protein-lipid binding via microscale thermophoresis (MST)

Purified GST and GST-tagged YAP were fluorescently labeled using a Monolith Protein Labeling Kit (MO-L011, Nano Temper) according to the manufacturer's instructions. The fluorescently labeled target proteins were diluted in MST buffer containing 50 mM Tris-HCl, pH 8.0, 50 mM NaCl, 80 mM KCl, and 0.05% Tween-20. For quantification of the binding affinity, sequential dilutions of 18:0/20:4 phosphoinositides or phosphatidylinositol in MST buffer were mixed with the fluorescently labeled proteins in equal volumes. The target–ligand mixtures were loaded into Monolith NT.115 Series capillaries (MO-K022, Nano Temper), and the MST traces were measured by Monolith NT.115 pico. The binding affinity was auto-generated by MO. Control v1.6 software.

## Immunohistochemistry

Tissue microarrays (TMA) of paraffin-embedded tissues were purchased from Biomax (PA482). To visualize the expression of PIPKIα and IPMK, TMA slides (0.5 μm thick) were heated at 65 °C in the hybridization oven until the paraffin melted to help sections adhere to the slides and assist in deparaffinization required for staining. The TMA slides were deparaffinized with Xylene and Ethanol. After deparaffinization and rehydration, antigen unmasking was performed at >95 °C for 30 min in a microwave oven using antigen retrieval buffer (0.05% Tween-20, 10 mM Tris, and 1 mM EDTA pH 6) for epitope retrieval, followed by cooling of the slides to room temperature. The TMA sections were incubated with blocking solution (10% normal goat serum) overnight at 4 °C. After washing in PBS with 0.1% Tween-20 for 5 min twice, immunostaining was performed for 60 min at room temperature with anti-PIPKIα or anti-IPMK antibodies (homemade; dilution 1:400). Normal rabbit IgG was used as a negative control. The breast cancer area in the TMA was identified via H&E staining, and PIPKIα and IPMK staining intensity in the breast cancer area was scored by a pathologist (KWF). Intensity was graded as 0 (no staining), 1 (weak staining), 2 (strong staining), and 3 (very strong staining).

## Statistics and reproducibility

Two-tailed unpaired *t* tests were used for pairwise significance unless otherwise indicated. We note that no power calculations were used. Sample sizes were determined based on previously published experiments where differences were observed (Chen et al, 2020, 2022; Choi et al, 2016). Each experiment was repeated independently at least three times with some exceptions. The sample sizes and the number of repeats are given in each figure legend. We used two to four independent experiments or biologically independent samples for statistical analysis. For knockdown experiments, the knockdown efficiency of each experiment was measured by quantifying immunoblots, and samples with a knockdown efficiency of <85% were excluded. Investigators were blinded to allocation during experiments and outcome assessment.

# Data availability

This study includes no data deposited in external repositories.

# Peer review information

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

## Acknowledgements

The authors would like to thank Eek-hoon Jho (University of Seoul) for the YAP, TAZ, and TEAD constructs. A homemade IPMK antibody was gifted from Dr. Seyon Kim (KAIST, South Korea). The authors are grateful to all members of the SC laboratory, and Dr. Jixin Dong (University of Nebraska Medical Center) for helpful discussions. We are grateful to members of the University of Nebraska Medical Center Advanced Microscopy Core Facility and Genomics Core for their service. This work was supported by NIH grants P20GM121316 (SC), R35GM150504 (SC), CA22287 (KWF), R35GM134955 (RAA), the Fred & Pamela Buffett Cancer Center Pilot Projects Program (SC and OJ), startup funds from the University of Nebraska Medical Center (SC), Department of Defense Breast Cancer Research Program grants W81XWH-17-1-0258 (RAA), W81XWH-17-1-0259 (VLC), W81XWH-21-1-0129 (VLC), HT9425-23-1-0553 (VLC), HT9425-23-1-0554 (RAA), a grant from the Breast Cancer Research Foundation (VLC), and a grant from National Science Foundation NSF-CHE-2310263 (MDB).

## Author contributions

**Oisun Jung**: Conceptualization; Resources; Data curation; Formal analysis; Supervision; Investigation; Visualization; Methodology; Writing—original draft; Writing—review and editing. **Min-jeong Baek**: Data curation; Formal analysis. **Colin Wooldrik**: Data curation; Formal analysis; Investigation. **Keith R Johnson**: Supervision; Investigation; Methodology; Project administration; Writing—review and editing. **Kurt W Fisher**: Data curation; Formal analysis; Funding acquisition; Writing—review and editing. **Jinchao Lou**: Resources. **Tanei J Ricks**: Resources. **Tianmu Wen**: Formal analysis; Investigation; Writing—original draft. **Michael D Best**: Resources; Funding acquisition; Writing—review and editing. **Vincent L Cryns**: Conceptualization; Funding acquisition; Writing—review and editing. **Richard A Anderson**: Conceptualization; Data curation; Funding acquisition; Writing—review and editing. **Suyong Choi**: Conceptualization; Resources; Data curation; Formal analysis; Supervision; Funding acquisition; Investigation; Methodology; Writing—original draft; Writing—review and editing.

## Disclosure and competing interests statement

The authors declare no competing interests.

# Expanded View Figures

**Figure EV1.** PIPKIα and IPMK are required for the expression of YAP/TAZ target genes, and agonist stimulation increases YAP associations with PIPKIα and IPMK in triple-negative breast cancer cells. ▶

(A) PIPKIα and IPMK were transiently knocked down in MDA-MB-468 cells, another triple-negative cell line, by transfecting siRNA for 72 h and the expression of several proteins was analyzed by immunoblotting. Representative immunoblot images of $n = 3$ independent experiments are shown. The expression of CTGF and CYR61 but not another known YAP/TAZ target AXL was reduced by PIPKIα or IPMK knockdown. (B) A YAP/TAZ–TEAD firefly luciferase reporter construct along with a *Renilla* luciferase construct were transfected in MDA-MB-231 cells 24 h after siRNA transfection against PIPKIα and IPMK. After another 48 h incubation, firefly and *Renilla* luciferase activities were measured and the graph is shown as mean ± s.d. of $n = 3$ independent experiments. PIPKIα or IPMK knockdown significantly reduced the activity of the YAP/TAZ–TEAD promoter. $*P < 0.05$; $**P < 0.01$, and n.s.; not significant in Student's $t$ test. (C, D) In all, 0.1 μM GST alone and GST-YAP recombinant proteins were incubated with 0.05 μM His$_6$-tagged recombinant PIPKIα (C) or IPMK (D). YAP proteins were pulled down with glutathione beads and the associated PIPKIα and IPMK were analyzed with immunoblotting. Representative immunoblot images of $n = 3$ independent experiments are shown. These results show YAP can directly interact with PIPKIα and IPMK. (E) Schematic representations of WT vs. a mutant YAP (mtWW) which contains inactivating mutations in the WW domains are shown. TBD TEAD-binding domain, TAD transactivation domain. (F) A schematic representation of the modules involved in the interactions of YAP/TAZ with PIPKIα and IPMK is shown. (G) Serum-starved MDA-MB-231 cells were treated with 10% serum or 5 μM LPA for 1 h. Cells were lysed and endogenous YAP was immunoprecipitated and the associated endogenous PIPKIα and IPMK were analyzed by immunoblotting. Representative immunoblot images of $n = 3$ independent experiments are shown. Treating the cells with serum or LPA increased the association of YAP with PIPKIα and IPMK. (H) MDA-MB-231 cells grown in 10% serum were treated with the indicated agonists for 90 min. Cells were lysed, endogenous YAP was immunoprecipitated, and the associated proteins were analyzed by immunoblotting. Representative immunoblot images of $n = 3$ independent experiments are shown. Treating the cells with any of the three agonists increased the association of YAP with PIPKIα and IPMK. (I) HEK293 cells grown in 10% serum were treated with 5 μM LPA for 2 h. The expression of endogenous YAP, phosphoS127 YAP, and CTGF was analyzed by immunoblotting. Representative immunoblot images of $n = 2$ independent experiments are shown. (J) Flag-YAP was co-transfected with HA-PIPKIα in HEK293 cells for 48 h. Cells grown in 10% serum were treated with 5 μM LPA for 2 h. Exogenous YAP was immunoprecipitated with an anti-Flag antibody and the associated exogenous PIPKIα was detected by immunoblotting with an anti-HA antibody. Representative immunoblot images of $n = 2$ independent experiments are shown. Treating the cells with LPA increased the association of PIPKIα with YAP. (K) Starved-parental or the indicated KO cells (pooled clones) were treated with 10% serum for 1 hr. Cells were lysed and the cell lysates were analyzed by immunoblotting with the indicated antibodies. Representative immunoblot images of $n = 2$ independent experiments are shown. KO of neither PIPKIα nor IPMK altered YAP phosphorylation at S127 residue. (L) Starved-parental or the indicated KO cells (pooled clones) were treated with 10% serum for 1 h. Cells were fixed and endogenous YAP localization was analyzed by immunofluorescence. Representative confocal images of $n = 2$ independent experiments are shown. Scale bar, 20 μm. (M) Starved-parental or YAP KO cells (pooled clones) were treated with 10% serum for 1 h. Endogenous TEAD was immunoprecipitated and the associated proteins were analyzed by immunoblotting with the indicated antibodies (top). Expression of the indicated proteins were analyzed by immunoblotting with the indicated antibodies (bottom). Representative immunoblot images of $n = 2$ independent experiments are shown. Serum treatment did not alter TEAD associations with PIPKIα and IPMK.

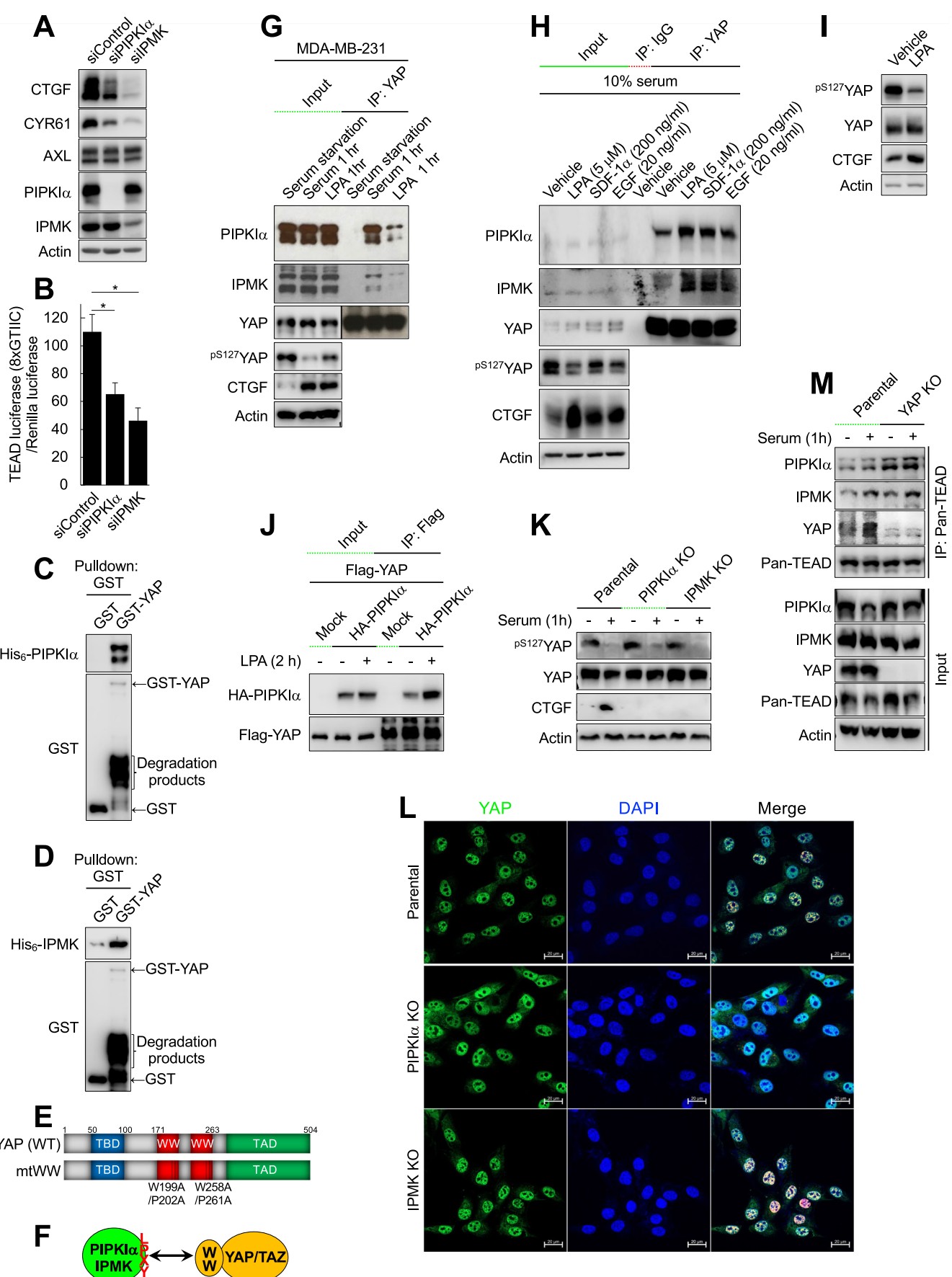

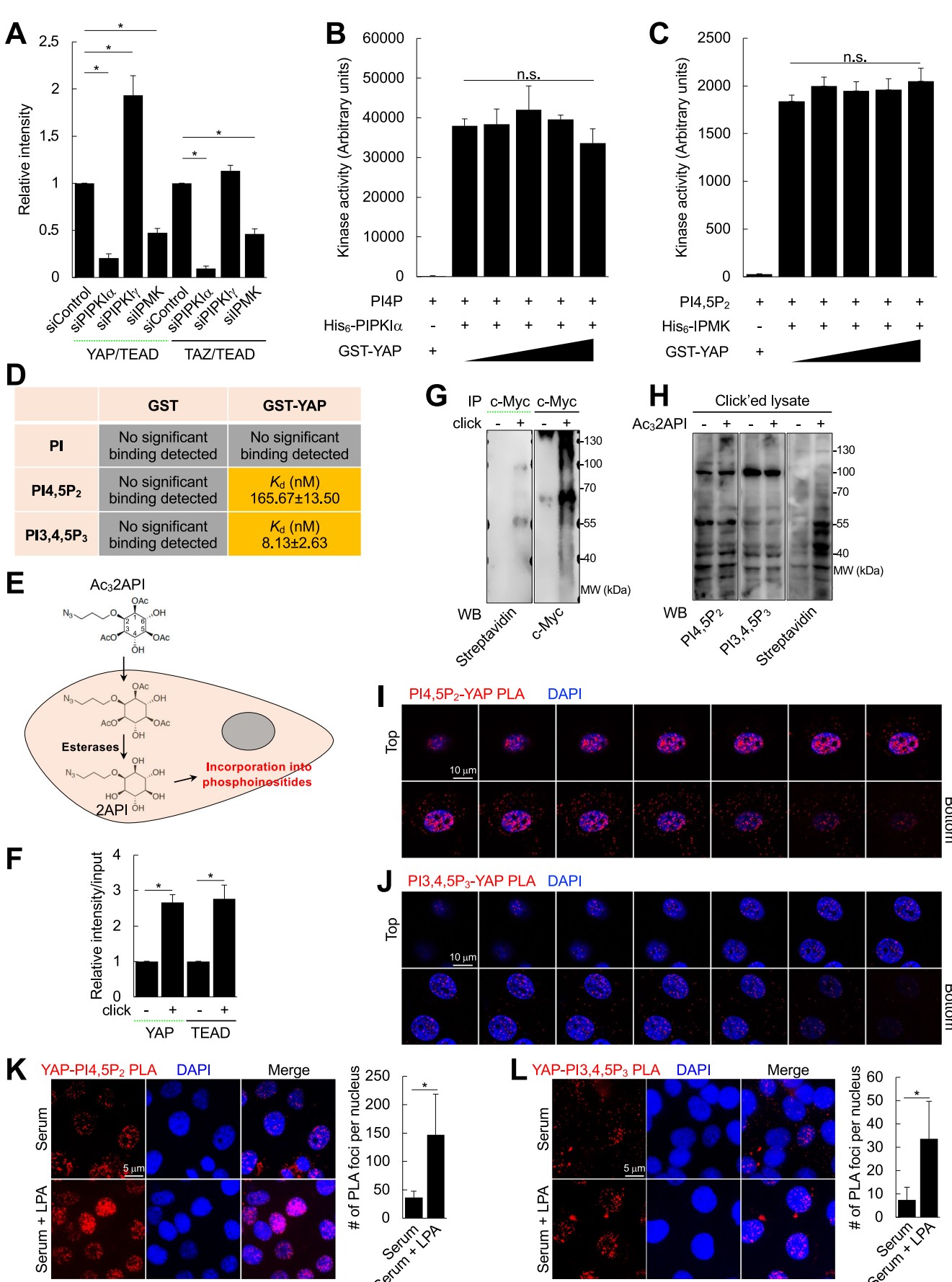

**Figure EV2. YAP does not alter the kinase activity of PIPKIα or IPMK, and serum and LPA stimulate the associations of YAP with PI(4,5)P$_2$ and PI(3,4,5)P$_3$ in the nucleus.**

(A) The intensity of immunoblots of Fig. 3A was quantified using ImageJ and the graph shows the mean ± s.d. of $n = 3$ independent experiments. *$P < 0.05$; **$P < 0.01$, and n.s.; not significant in Student's $t$ test. (B, C) 0.05 μM His$_6$-tagged PIPKIα (B) and IPMK (C) were incubated with 0.2 μM diC8 PI(4)P or diC8 PI(4,5)P$_2$, respectively, in the absence or presence of various concentrations of GST-YAP (0.001, 0.01, 0.1, 1.0, and 10.0 μM). The activities of the kinases were measured using the ADP-Glo assay (Promega). The graphs show the mean±s.d. of $n = 3$ independent experiments. YAP did not alter the activity of either kinase. *$P < 0.05$; **$P < 0.01$, and n.s.; not significant in Student's $t$ test. (D) Summary of GST alone or GST-YAP bindings to PI, PI(4,5)P$_2$, or PI(3,4,5)P$_3$ measured by MST. Raw data are available in Appendix Fig. S1. (E) A schematic representation of the molecular structure Ac$_3$2API and how it can become metabolically incorporated into phosphoinositides after removal of acetyl groups by esterase to produce azido-*myo*-inositol probe 2API. (F) The intensity of the immunoblots of Fig. 4B was quantified using ImageJ and the graph shows the mean ± s.d. of $n = 3$ independent experiments. *$P < 0.05$; **$P < 0.01$, and n.s.; not significant in Student's $t$ test. (G) Starved MDA-MB-231 cells were fed with Ac$_3$2API for 24 h in the presence of 10% dialyzed serum. Cells were lysed and azide-tagged molecules were conjugated to biotin–alkyne through a click reaction. Endogenous c-Myc was immunoprecipitated and the associated complexes were analyzed by immunoblotting. Representative immunoblot images of $n = 2$ independent experiments are shown. (H) The clicked lysates were analyzed by anti-PI(4,5)P$_2$ or PI(3,4,5)P$_3$ antibodies. Biotinylated 2API was resolved by streptavidin. Many immunoblot bands overlapped with streptavidin signals. Representative immunoblot images of $n = 3$ independent experiments are shown. (I, J) Starved MDA-MB-231 cells were stimulated with 10% serum for 1 h. The images are z-stacks of PI(4,5)P$_2$-YAP PLA (E) and PI(3,4,5)P$_3$-YAP PLA (F) taken using a confocal microscope with each frame differing by 0.2 μm. DAPI was used to stain the nucleus. Representative images of $n = 3$ independent experiments are shown. Scale bar, 10 μm. (K, L) MDA-MB-231 cells grown in 10% serum were stimulated with 5 μM LPA for 90 min. Cells were fixed and the association of YAP with PI(4,5)P$_2$ (G) or PI(3,4,5)P$_3$ (H) was visualized by PLA. The images were obtained by widefield epifluorescence microscopy. The number of PLA puncta was counted from at least 10 cells and the graph shows the mean±s.d. of $n = 3$ independent experiments. DAPI staining was used to distinguish the nucleus from the cytoplasm. Treating the cells with LPA significantly increased the number of nuclear puncta. Scale bar, 10 μm. *$P < 0.05$; **$P < 0.01$, and n.s; not significant in Student's $t$ test.

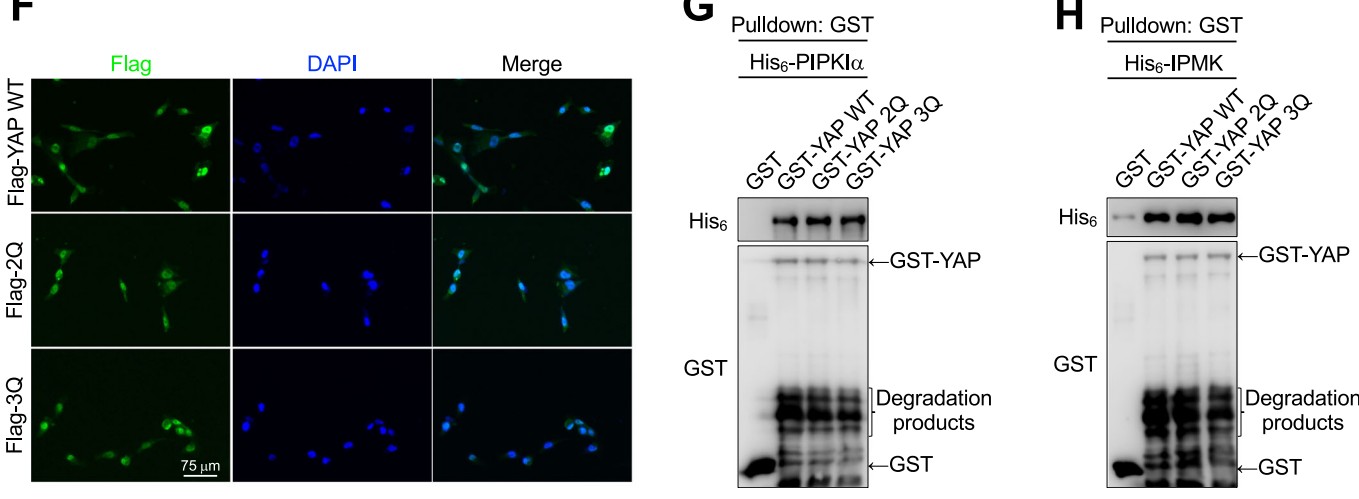

**A**

YAP
PBM
1   50   100   171   263   504
TBD   WW   WW   TAD

TAZ
13   57   171   208   400
TBD   WW   TAD

TEADs
DBD   Y/TBD

**B**

GST-TAZ
GST-YAP
His6-TEAD1

MW (kDa)
-100
-70
-55
-40

Coomassie

**C**

Binding buffer: 0.005% Triton X-100 in TBS

1. Binding of 0.05 μM GST-YAP to glutathione beads (30 min at room temp.)

2. Incubation with 0.4 μM lipids (30 min at room temp.)

3. Wash beads once with binding buffer

4. Incubation of 0.02 μM His6-TEAD1 (30 min at room temp.)

5. Wash beads twice with binding buffer

6. SDS PAGE

**D**

Pulldown: GST
0.02 μM His6-TEAD1
+ 0.05 μM GST-TAZ

Natural lipids

No lipid   PI   PI4P   PI4,5P2

His6-TEAD1

GST-TAZ

His6/GST
8
6
4
2
0
**   **

**E**

PI diC8
PI3P diC8
PI4P diC8
PI5P diC8
PI3,4P2 diC8
PI3,5P2 diC8
PI4,5P2 diC8
PI3,4,5P3 diC8
PI4,5P2 diC16
PI3,4,5P3 diC16
PI 18:0/20:4
PI4P 18:0/20:4
PI5P 18:0/20:4
PI4,5P2 18:0/20:4
PI3,4,5P3 18:0/20:4

**F**

Flag   DAPI   Merge

Flag-YAP WT

Flag-2Q

Flag-3Q

75 μm

**G**

Pulldown: GST
His6-PIPKIα

GST   GST-YAP WT   GST-YAP 2Q   GST-YAP 3Q

His6

GST-YAP

GST

Degradation products

GST

**H**

Pulldown: GST
His6-IPMK

GST   GST-YAP WT   GST-YAP 2Q   GST-YAP 3Q

His6

GST-YAP

GST

Degradation products

GST

◀ **Figure EV3.   The association of TAZ with TEAD1 is facilitated by phosphoinositides, and the binding of phosphoinositides does not alter the nuclear localization of YAP nor its binding to PIPKIα and IPMK.**

(A) The schematic diagrams show the polybasic motifs (PBM) located within the TEAD-binding domains (TBD) of YAP and TAZ that may mediate their binding to phosphoinositides. DBD, DNA-binding domain; TAD, transactivation domain; Y/TBD, YAP/TAZ-binding domain in the TEADs. (B) A Coomassie-stained gel of the three recombinant proteins used in the study is shown. The positions of the full-length proteins are indicated by arrows. (C) A detailed protocol used for the in vitro binding assays with lipids is presented. Note that the lipids and Triton X-100 were used at concentrations below their critical micelle concentrations. (D) 0.02 μM His$_6$-TEAD1 was incubated with 0.05 μM GST-TAZ in the absence or presence of the indicated lipids (0.4 μM). TAZ was pulled down with glutathione beads and the associated TEAD1 was analyzed by immunoblotting. The graph shows the mean ± s.d. of $n = 3$ independent experiments. *$P < 0.05$; **$P < 0.01$, and n.s.; not significant in Student's $t$ test. (E) The chemical structures of the lipids used in the study are shown. (F) Flag-tagged WT YAP and the 2Q and 3Q mutants were transiently expressed in MDA-MB-231 cells. Cells were fixed and exogenous YAP proteins were visualized by immunostaining with an anti-Flag antibody. The images were obtained by widefield epifluorescence microscopy. DAPI staining was used to distinguish the nucleus from the cytoplasm. Representative immunostaining images of $n = 2$ independent experiments are shown. Scale bar, 75 μm. All the forms of YAP localize to the nucleus. (G, H) 0.1 μM GST alone, GST-WT, GST-2Q, and GST-3Q YAP recombinant proteins were incubated with 0.05 μM His$_6$-tagged recombinant PIPKIα (B) or IPMK (C). The YAP proteins were pulled down with glutathione beads and the associated PIPKIα and IPMK were analyzed with immunoblotting. Representative immunoblot images of $n = 3$ independent experiments are shown. Both PIPKIα and IPMK bound equally well to all the forms of YAP.

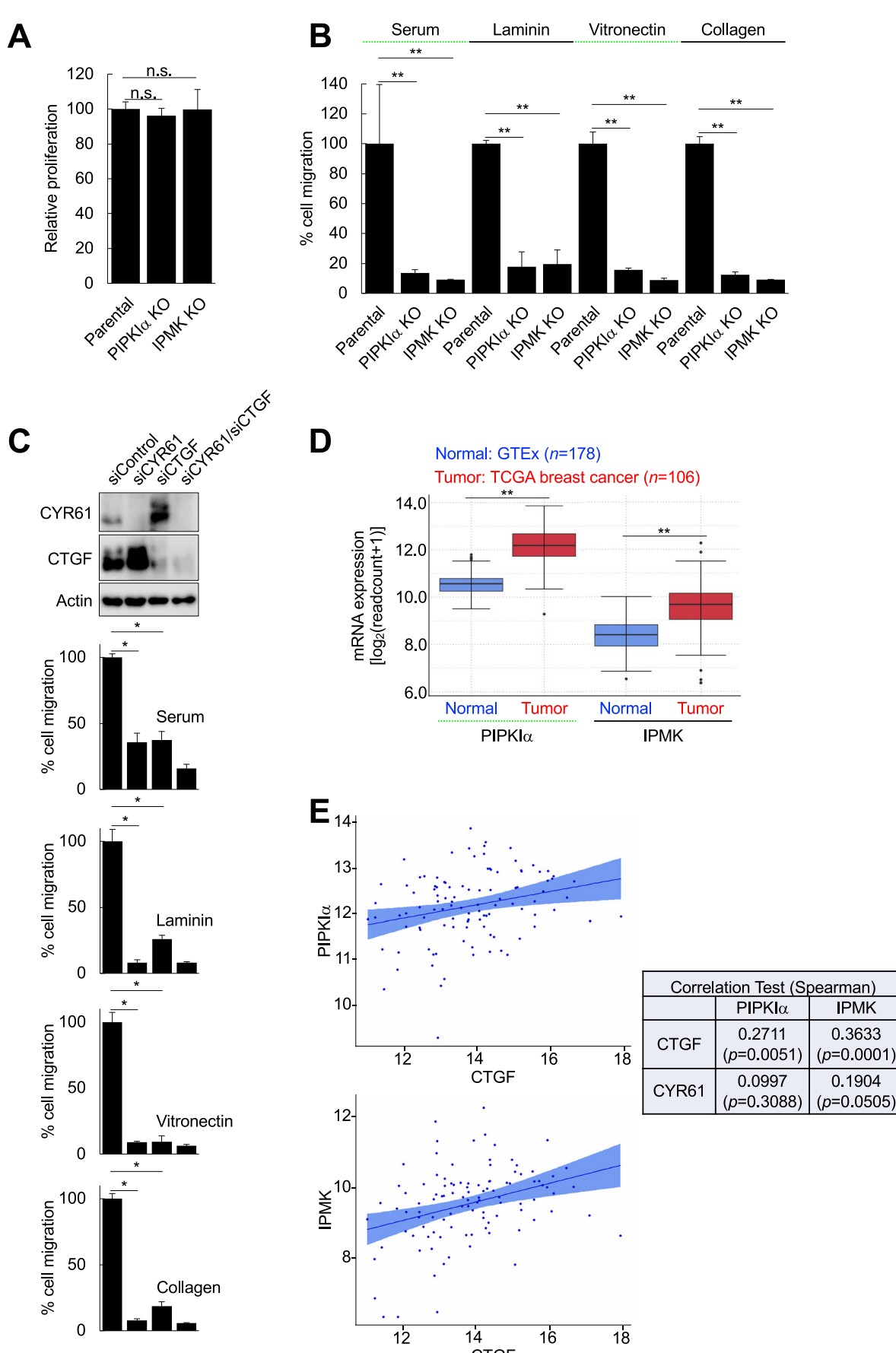

◀ **Figure EV4.  PIPKIα and IPMK are required for breast cancer cell motility and are overexpressed in breast tumors.**

(**A**) The relative proliferation rates of MDA-MB-231 cells and cells with either PIPKIα or IPMK knocked out were measured using the Cell-Titer Glo assay (Promega). The graph shows the mean ± s.d. of $n = 3$ independent experiments. Knocking down either PIPKIα or IPMK did not affect the proliferation rate. *$P < 0.05$; **$P < 0.01$, and n.s.; not significant in Student's *t* test. (**B**) The migration of MDA-MB-231 cells and cells with either PIPKIα or IPMK knocked out towards 10% serum, 10 μg/ml laminin, 10 μg/ml vitronectin, or 25 μg/ml collagen IV was measured using Transwell inserts with 8.0 μm pores (Corning). The cells that migrated through the filter were visualized by DAPI and crystal violet staining and quantified by counting (Appendix Fig. S2A). The graph shows the mean ± s.d. of $n = 3$ independent experiments. In each case the migration of the KO cells was significantly lower than that of the parental cells. *$P < 0.05$; **$P < 0.01$, and n.s; not significant in Student's *t* test. (**C**) CTGF and CYR61 were knocked down singly or together in MDA-MB-231 cells. The extent of knockdown was determined by blotting. Cell migration induced by serum or several extracellular matrix components was measured. The graphs show the mean ± s.d. of $n = 3$ independent experiments. In each case, knocking down CTGF or CYR61 either alone or together impaired the migration rate. *$P < 0.05$; **$P < 0.01$, and n.s.; not significant in Student's *t* test. (**D**) RNA sequencing data of normal breast (GTEx) (Consortium, 2013) and malignant breast tumor (TCGA breast cancer) (Cancer Genome Atlas Research et al, 2013) tissues were used to compare relative mRNA expression levels of PIPKIα and IPMK. The expression of both PIPKIα and IPMK is significantly higher in breast cancer tissue. Center lines of boxplots are the median of each cohort data. Bounds of boxplot is the range from the first quartile to the third quartile. The lower whiskers of boxplots are distance between the first quartile and the first quartile minus 1.5 interquartile. The higher whiskers of boxplots are distance between the third quartile and the third quartile plus 1.5 interquartile. *$P < 0.05$; **$P < 0.01$, and n.s.; not significant in Wilcoxon rank-sum test. (**E**) Using the RNA sequencing data from the TCGA database of malignant breast tumors ($n = 106$), the correlation of PIPKIα and IPMK mRNA expression with that of CTGF and CYR61 was analyzed via Spearman method. The expression of CTGF is significantly correlated with the expression of PIPKIα and IPMK.

