## [Peer Review File · The EMBO Journal]

Nuclear phosphoinositide signaling promotes YAP/TAZ-TEAD transcriptional activity in breast cancer

Suyong Choi, Oisun Jung, Min-jeong Baek, Colin Wooldrik, Keith Johnson, Kurt Fisher, Jinchao Lou, Tanei Ricks, Michael Best, Vincent Cryns, Richard Anderson, and Tianmu Wen

Corresponding author: Suyong Choi (schoi@unmc.edu)

Review Timeline:

Submission Date:	21st Mar 23
Editorial Decision:	24th Apr 23
Revision Received:	9th Oct 23
Editorial Decision:	17th Nov 23
Revision Received:	29th Feb 24
Accepted:	8th Mar 24

Editor: Ieva Gailite

Transaction Report:

Dear Suyong,

Thank you for submitting your manuscript for consideration by the EMBO Journal. We have now received comments from three reviewers, which are included below for your information.

As you will see from the reports, all reviewers find the proposed model of phosphoinositide-mediated enhancement of YAP-TEAD interaction per se interesting. However, they also indicate a number of substantive concerns regarding the specificity of the YAP interaction with PIPK1alpha and IMPK in the nuclear compartment, the relevance of specific phosphoinositides for this interaction, and the regulation of this mode of YAP activation by upstream inputs. In particular, they request application of metabolic labelling (referee #2, points 2 and 3, also echoed by referee #1 in point 11) and mass-spectrometry analysis (referee #3, summary section) to provide further conclusive support to this model.

If you find that you are able to address the main issues raised by the reviewers, I would be happy to consider a revised version of the manuscript. I think it would be helpful to discuss the revision in more detail via email or phone/videoconferencing - please let me know which option you prefer. I should also add that it is The EMBO Journal policy to allow only a single major round of revision and that it is therefore important to resolve the main concerns at this stage. If several of the main points cannot be experimentally addressed, the revised manuscript could be potentially suitable for publication in our sister journal EMBO Reports.

We generally allow three months as standard revision time, which can be extended to six months in the case of major revisions. As a matter of policy, competing manuscripts published during this period will not negatively impact on our assessment of the conceptual advance presented by your study. However, please contact me as soon as possible upon publication of any related work to discuss the appropriate course of action. Should you foresee a problem in meeting this deadline, please let us know in advance to discuss an extension.

When preparing your letter of response to the referees' comments, please bear in mind that this will form part of the Review Process File and will therefore be available online to the community. For more details on our Transparent Editorial Process, please visit our website: <https://www.embopress.org/page/journal/14602075/authorguide#transparentprocess>. Please also see the attached instructions for further guidelines on preparation of the revised manuscript.

Please feel free to contact me if you have any further questions regarding the revision. Thank you for the opportunity to consider your work for publication. I look forward to discussing your revision.

With best regards,

Ieva

At EMBO Press we ask authors to provide source data for the main manuscript figures. Our source data coordinator will contact you to discuss which figure panels we would need source data for and will also provide you with helpful tips on how to upload

and organize the files.

We realize that it is difficult to revise to a specific deadline. In the interest of protecting the conceptual advance provided by the work, we recommend a revision within 3 months (23rd Jul 2023). Please discuss the revision progress ahead of this time with the editor if you require more time to complete the revisions.

Referee #1:

Nuclear phosphoinositide signaling controls the YAP/TAZ-TEAD pathway in breast cancer

Oisun Jung^{1,2}, Min-Jeong Baek^{1,2}, Keith R. Johnson^{1,2,3}, Kurt W. Fisher^{2,4}, Vincent L. Cryns⁵, Richard A. Anderson⁵, and Suyong Choi¹

Summary

The manuscript entitled "Nuclear phosphoinositide signaling controls the YAP/TAZ-TEAD pathway in breast cancer" by Jung et al describes a new role for phospholipids in the regulation of the YAP/TAZ signaling pathway. They provide evidence that specific phosphoinositides function as cofactors facilitating the binding of YAP/TAZ to TEAD in the nucleus. In particular they predict that PIP2 and PIP3 that are generated by PIPK1 α and IPMK are important for YAP activity as mutations in YAP that impair lipid binding prevent YAP/TAZ binding to TEAD. The authors attempt to show the significance of this regulation in breast cancer and speculate that PIPK1 α and IPMK would be new therapeutic targets to disrupt YAP signaling. Overall, the paper has great potential and indeed there are exciting implications for cancer therapeutics as targeting YAP directly has had limited success. In conclusion, the study was found to be of broad significance and some of results were found to support the conclusions. There were several concerns/suggestions that the authors should address in order to improve this study for publication quality and rigor.

Specific Comments:

1. In Figure 1 Panel A YAP phosphorylation is not discussed. Also, there appears to be no differences in P-YAP upon kinase knockdown. If there is this needs to be quantified.
2. Figure 1 Panel H, why does there seems to be no difference in YAP phosphorylation under the different treatments (serum starvation, adding back serum or LPA).
3. While the PLA observation is interesting, I am wondering if an orthogonal method would be beneficial. For instance, blocking the nuclear translocation of PIPK1 α and IPMK by mutating the NLS sequences? Or maybe using a constitutively inactive mutant to IP with Type Is and IPMK?
In this condition YAP should not be in the nucleus, so there should be little binding with the lipid kinases.
4. Also demonstrating that YAP localization is not affected could support a primarily nuclear interaction.

5. Do any of the stimuli tested (serum, LPA) stimulate PIPK1 α or IPMK translocation into the nucleus (has this been shown before?).
6. In Supplementary Figure 1, Panel A: AXL protein levels not discussed.
7. In Supplementary Figure 1, Panel F: I don't understand the PIP2 blot and it is not discussed in text. How does this antibody work? There is a comment in the discussion but should be mentioned in results. As well as P-YAP and CTGF.
8. In In Supplementary Figure 1, Panel G. Is this a positive control for LPA treatment? It is not discussed in text.
9. In Figure 2 a TEAD TEAD-luciferase assay would complement/ support data
10. In Supplemental Figure 2, all these inhibitors have major off target effects and really add nothing to the manuscript and actually pose more complications in data interpretation.

For example, the use of AT9283 makes no sense. "AT9283 is a multi-targeted kinase inhibitor with potent activity against Aurora A/B, JAK2/3, Abl (T315I) and Flt3 (IC50s ranging from 1 to 30 nM). AT9283 inhibits growth and survival of multiple solid tumors in vitro and in vivo." More explanations needed.

Also, even though ISA-2011B does binds with high affinity to PIPK1 α and inhibits its protein expression and cancer growth in multiple models it does have significant off-target effects, including potent binding to multiple other kinases, including the class IA PI3K p110 α .

11. This may be upon the scope of EMBO, but it is concerning that only exogenous supplementation of lipids and PIP beads are used to define lipid binding to YAP, can authors further validate using better more quantitative assays such as ITC, EMSA, NMR or even computational modeling?
12. Figure 3, there is no discussion of why YAP would bind PI5P.
13. In Figure 5, it is not clear if PIP2 and YAP in the cytoplasm from PLA data? Can you explain? And if they do, where would this occur?
14. In Figure 5, as a control, do these YAP mutants still bind PIPK1 α and IMPK? Also, does the localization of YAP change?
15. In the discussion the results are not completely discussed therefore it is difficult to understand some of the author's conclusions. For example, a major issue that needs to be addressed is that in the discussion it states that the YAP- PIPK1 α /IPMK complexes form and then they translocate to nucleus. There is no evidence of this to support this statement. Can this be explained or validated to support the text?

Referee #2:

Jung, et al. provide a compelling connection of PIPK1a and IPMK to breast cancer motility, which is proposed to occur through a new nuclear signaling pathway. The authors show PIPK and IPMK co-IP with YAP/TAZ, where PIPK1a is proposed to locally generate PI(4,5)P2 from PI4P (with the PI4P presumably associated with YAP/TAZ constitutively or the mechanism doesn't make much sense), the PI4,5,P2 is then acted on by IPMK to generate PI(3,4,5)P3. The PI(3,4,5)P3 is proposed to mediate direct inter-protein interactions between YAP and TEAD.

There are some solid aspects of the manuscript, such as some of the PLA data and co-Ips indicating PIPK1a-IPMK and YAP exist in the same cellular complexes, which is dependent on LPXY motif. siRNA of PIPK and IPMK reduce YAP-TEAD interaction, and that siPI3K had no effect if encouraging, but overall many of the conclusions are based on cell biological experiments that cannot be made based on cell biology, but require biochemical experiments and/or metabolic labeling. Thus major revisions are required:

- 1) IPMK generates inositol phosphates that have diverse effects on transcription. Its unclear which IPMK product is effecting the Yap/TAZ pathway, but certainly the authors cannot conclude, as they do, that it is PI(3,4,5)P3 that is the product responsible for the effects. This critique could be addressed by simply changing the language.
- 2) The approaches used in this manuscript rely almost exclusively on antibodies, some of which are notoriously non-specific, such as the PI(4,5)P2 and PI(3,4,5)P3 antibodies. It would be more convincing if 3H-inositol was used to metabolically label the cells, then show specific enrichment of the label with epitope-tagged YAP/TAZ immunoprecipitates in a scint counter, but not other transcription factors that are activated by serum, then show decreases to specific 3H-inositol enrichment upon PIPK1a and IPMK knockdown. HPLC of the immunoprecipitated and deacylated glycerolipid headgroups would prove the radiolabel was

PIPs and not inositol phosphates.

3) Fig 3D-E are troubling. The PIPK1 α and IPMK association with YAP/TAZ suggest the PIPK1 α kinase substrate PI4P must be associated with YAP/TAZ to generate PI(4,5)P₂ and only then can IPMK make the PI(3,4,5)P₃. Thus knockdown of IPMK should not effect PI(4,5)P₂ generation in the PLA, but it does (Figure 3D). Further, knock down of PIPK should eliminate all PI(3,4,5)P₃ by decreasing the substrate, but it does not (Figure 3E). These results need to be back up with metabolic labeling studies that demonstrate the signal being observed is actually a phosphoinositide, and not just antibody cross reactivity of these weak antibody reagents.

4) That short chain diC8 lipids had no effect and all long chain phosphoinositides had an effects suggests the stickiest part of the phospholipid is mediating the effect, and no evidence is provided that the effect is specific, e.g. saturable to a single site on YAP. A specific, saturable and stoichiometric interaction (1:1, 2:1, 2:2:, etc) must be demonstrated with KDs determined, otherwise no convincing evidence is provided that the phosphoinositides are specifically regulating the complex.

Referee #3:

The Hippo pathway is essential for development, organ size regulation, and tissue homeostasis, and its dysregulation contributes to cancer development. The YAP/TAZ transcriptional co-activators bind with TEA domain (TEAD) family transcription factors to promote gene expression, cell growth, and cancer progression. Previous studies have revealed the critical role of plasma membrane phosphatidylinositol-4-phosphate in hippo regulation through phosphatidylinositol transfer proteins α and β (PIP α/β) (Li, F. L., Fu, V., Liu, G., Tang, T., Konradi, A. W., Peng, X., ... & Guan, K. L. (2022). Hippo pathway regulation by phosphatidylinositol transfer protein and phosphoinositides. *Nature Chemical Biology*, 18(10), 1076-1086.). The current study by Oisun Jung et al. proposed a new mechanistic scheme that nuclear phosphoinositides generated by PIPK1 α and IPMK regulate the spatiotemporal binding of YAP/TAZ to TEADs. PIPK1 α and IPMK inducibly associate with YAP/TAZ via LPXY motifs on PIPK1 α and IPMK and the WW domains on YAP upon agonist activation of membrane receptors. The complex is translocated to the nucleus, where PIPK1 α and IPMK produce PI4,5P₂ and PI3,4,5P₃. PI4,5P₂ and PI3,4,5P₃ are transferred to YAP/TAZ and bind to YAP/TAZ. The binding of PI4,5P₂ and PI3,4,5P₃ to YAP/TAZ facilitates their association with TEADs, resulting in the transcription of YAP/TAZ target genes. The studies are interesting; however, the direct evidence of PI4,5P₂ and PI3,4,5P₃ binding to YAP/TAZ is based on in vitro assay with high concentrations of PI4,5P₂ and PI3,4,5P₃ (The results are not very significant though in Figure 4C) or immunofluorescent staining to examine the co-localization, which often results in false positive. It would be critical to examine the existence of the complex of PI4,5P₂, PI3,4,5P₃ and YAP/TAZ in cells by Mass Spectrometry or other techniques. Besides, the manuscript needs to be improved to address several technical and experimental issues before considering published on *The EMBO Journal*.

1, In Figure 1F, the signal of YAP in the input is relatively weak, it would be beneficial to run the blots of input and pull-down separately and have a better image.

2, In Figure 1G, Myc-IPMK has two bands, and both bind to YAP, is the up band the posttranslational modification form of IPMK?

3, In Figure 1H, both PIPK1 α and IPMK showed two bands, are the up bands the posttranslational modification forms or isoforms of PIPK1 α and IPMK?

4, In Figure 1I, the IPMK-YAP PLA signal was also significantly increased in the cytosol. What's the explanation given that the interaction happens in the nucleus?

5, In Figure 2A, PIPK1 α and PIPK1 γ showed compensatory effects, the knockdown of PIPK1 α increased PIPK1 γ and vice versa. It's interesting to see the knockdown of PIPK1 γ increased protein level of PIPK1 α and TEAD-YAP binding, but it's hard to see the binding of TEAD-TAZ given the blot is a little overexposed, it would be interesting to see how modulation of PIPK1 α and PIPK1 γ affects TEAD-YAP as well as TEAD-TAZ binding in more details.

6, The authors showed two bands for PIPK1 α and PIPK1 γ in some blots (Figure 1H, 2A), while in some blots, there is only one band (Figure 2E, 2F), what's the matter in this case?

7, In Figure 1F, the blot for input IPMK is not clear.

8, In Figure 3A, it would benefit the readers if the author could include the blots of unbound fractions in each treatment.

9, In Figure 3B, serum stimulation decreased cytosolic PI4,5P₂-YAP PLA, while increasing its nuclear fraction. While in Figure 3B, serum stimulation increased both cytosolic and nuclear PI3,4,5P₃-YAP PLA, does serum stimulation increase total PI3,4,5P₃, not PI4,5P₂? What's the potential explanation?

10, The authors indicated, "The impact of natural source PI5P and PI3,4,5P₃ could not be tested as they are not commercially available.", however, in Figure 4C, natural PI3,4,5P₃ has been tested though without effects on TAP-TEAD1 binding.

11, In Figure 5B, it looks like Lys97 is the most important site for YAP binding to PI4,5P₂ and PI3,4,5P₃; it would be interesting to test how the single mutation affects the binding of phosphoinositides to YAP.

12, Figure 5B and 5C used TEAD1 to examine how the binding of phosphoinositides to YAP affects TEAD1-YAP binding in vitro. Why do the authors switch to TEAD4-YAP in the cell-based assay?

We would like to thank the referees for their invaluable comments and suggestions. Below we detail the changes to the revised manuscript that address the referees' comments followed by the revisions that we have made. Please note that new figures added or modified are indicated in blue. We noticed that identical images were inadvertently used in Supplementary Fig. 6C (original submission), and the error was corrected in the revised figure (Supplementary Fig. 7C).

Referee #1:

The manuscript entitled "Nuclear phosphoinositide signaling controls the YAP/TAZ-TEAD pathway in breast cancer" by Jung et al describes a new role for phospholipids in the regulation of the YAP/TAZ signaling pathway. They provide evidence that specific phosphoinositides function as cofactors facilitating the binding of YAP/TAZ to TEAD in the nucleus. In particular, they predict that P4,5IP₂ and PI3,4,5P₃ that are generated by PIPKI α and IPMK are important for YAP activity as mutations in YAP that impair lipid binding prevent YAP/TAZ binding to TEAD. The authors attempt to show the significance of this regulation in breast cancer and speculate that PIPKI α and IPMK would be new therapeutic targets to disrupt YAP signaling. Overall, the paper has great potential and indeed there are exciting implications for cancer therapeutics as targeting YAP directly has had limited success. In conclusion, the study was found to be of broad significance and some of results were found to support the conclusions. There were several concerns/suggestions that the authors should address in order to improve this study for publication quality and rigor.

1. In Figure 1 Panel A YAP phosphorylation is not discussed. Also, there appears to be no differences in P-YAP upon kinase knockdown. If there is this needs to be quantified.

In normal culture conditions, knockdown of PIPKI α or IPMK has no observable impact on YAP phosphorylation at S127 residue (Fig. 1A). We further tested this with PIPKI α and IPMK knockout cells. Serum-starved cells were treated with 10% serum for 1 hour and we observed that YAP S127 phosphorylation is dramatically reduced by serum stimulation in MDA-MB-231 cells (Supplementary Fig. 2A, 2E). Importantly, however, there was no observable difference of YAP S127 phosphorylation in PIPKI α and IPMK-depleted cells compared to parental cells (Supplementary Fig. 2E). YAP phosphorylation is well-documented to regulate nucleo-cytoplasmic shuttling of YAP (Ma *et al*, 2019; Moroishi *et al*, 2015; Zanconato *et al*, 2019). In response to agonist activation, the Hippo kinase pathway is inactivated and unphosphorylated YAP translocates to the nucleus. Consistently, upon serum stimulation we observed that YAP accumulates in the DAPI-positive nuclei (Fig. 2C, 2D) and the depletion of PIPKI α nor IPMK has no impact on YAP nuclear accumulation (Supplementary Fig. 2F). Taken together, our phosphoinositide-driven YAP/TAZ-TEAD regulation mechanism is independent of YAP phosphorylation status.

2. Figure 1 Panel H, why does there seem to be no difference in YAP phosphorylation under the different treatments (serum starvation, adding back serum or LPA).

We changed the immunoblot images having better resolution in the YAP phosphorylation. Upon serum and lysophosphatidic acid (LPA) treatment, YAP S127 phosphorylation was noticeably decreased and the expression of a YAP/TAZ target CTGF was concomitantly increased (Supplementary Fig. 2A).

3. While the PLA observation is interesting, I am wondering if an orthogonal method would be beneficial. For instance, blocking the nuclear translocation of PIPKI α and IPMK by mutating the NLS sequences? Or maybe using a constitutively inactive mutant to IP with PIPKI α and IPMK? In this condition YAP should not be in the nucleus, so there should be little binding with the lipid kinases.

In the original submission, we proposed that YAP/TAZ interact with PIPKI α /IPMK initially in the cytoplasm and then translocate into the nucleus upon agonist activation. This model was based on the proximity ligation assay (PLA) data in Fig. 2A and 2B, showing that YAP/TAZ-PIPKI α /IPMK interactions are increased both in the cytoplasm and the nucleus upon serum stimulation. During the revision, we now discovered that PIPKI α and IPMK are largely in the nucleus while YAP sharply accumulates in the

nucleus upon serum stimulation (Fig. 2C, 2D). Via immunoprecipitation, we further showed that PIPKI α and IPMK are constantly associated with TEAD regardless of YAP (Supplementary Fig. 2G). These suggest that TEAD forms a complex with PIPKI α /IPMK in the nucleus, and YAP translocates into the nucleus upon agonist stimulation to form a ternary complex. Based on these observations, we changed our model mechanism as in Fig. 7I.

4. Also demonstrating that YAP localization is not affected could support a primarily nuclear interaction. In MDA-MB-231 cells we mainly used in the present study, a substantial fraction of YAP was found in the nucleus consistently with a prior report (Andrade *et al*, 2017). Upon serum stimulation, YAP phosphorylation was dramatically decreased (Supplementary Fig. 2A, 2E) and YAP nuclear localization was further increased, while most of PIPKI α and IPMK were in the nucleus in the presence or absence of serum (Fig. 2C, 2D and Supplementary Fig. 2F). These indicate that the enhancement of YAP interactions with PIPKI α and IPMK by agonist stimulation is largely due to nuclear translocation of YAP.

5. Do any of the stimuli tested (serum, LPA) stimulate PIPKI α or IPMK translocation into the nucleus (has this been shown before?).

Although immunostaining data in Fig. 2C, 2D demonstrate that the majority of PIPKI α and IPMK are localized in the nucleus, the localization of PIPKI α and IPMK in other subcellular locations has also been reported. PIPKI α is found in the plasma membrane (Gonzales *et al*, 2020), membrane ruffles (Doughman *et al*, 2003), membrane protrusions and invadopodia (Choi *et al*, 2016), intracellular vesicles such as tubulating lysosomes (Rong *et al*, 2012), endoplasmic reticulum, and mitochondria (Wu *et al*, 2020). Consistently, we also observed clear PIPKI α immunostaining signals in the cytoplasm and the plasma membrane (Rebuttal Fig. 1). Although there was no profound change in PIPKI α nuclear localization by serum stimulation (Fig. 2C), we and others previously reported that DNA damage can enhance nuclear PIPKI α accumulation (Chen *et al*, 2022; Choi *et al*, 2019; Wang & Sheetz, 2022).

IPMK was predominantly localized in the nucleus (Rebuttal Fig. 1) and serum stimulation did not alter its nuclear localization (Fig. 2D). Nuclear functions of IPMK are well documented. IPMK regulates key nuclear events such as transcription (Blind *et al*, 2012; Guha *et al*, 2019; Xu *et al*, 2013), DNA damage repair (Wang *et al*, 2017), mRNA export (Wickramasinghe *et al*, 2013), and transcription factor stability (Fu *et al*, 2018). Although largely found in the nucleus, nucleo-cytoplasmic shuttling of IPMK has also been reported. In response to autophagy-inducing stress signals, a subset of IPMK is exported out of the nucleus and inhibits autophagy (Chen *et al*, 2020). Protein kinase CK2-mediated phosphorylation is reported to promote cytoplasmic retention of IPMK (Meyer *et al*, 2012). Wnt3a treatment induces plasma membrane localization of IPMK (Wang & Wang, 2012). In line with these reports, we also observed IPMK localization at the perinuclear area and the plasma membrane (Rebuttal Fig. 1).

6. In Supplementary Figure 1, Panel A: AXL protein levels not discussed.

AXL receptor tyrosine kinase is a well-established YAP/TAZ target gene in many cell types (King *et al*, 2020; Wang *et al*, 2018; Yamaguchi & Taouk, 2020). However, we observed no difference in AXL protein level in PIPKI α or IPMK-depleted MDA-MB-468 breast cancer cells. Our observation is consistent with prior reports that YAP depletion/inhibition has no impact on AXL mRNA level in MDA-MB-231 breast cancer cells (Zanconato *et al*, 2018) and in lung cancer cell lines at the protein level

Rebuttal Fig. 1 Starved MDA-MB-231 cells were stimulated with 10% serum for 2 hours. Cells were fixed with paraformaldehyde and stained for PIPKI α and IPMK. Images were taken with Zeiss LSM 800 confocal laser scanning microscope.

(Saab *et al*, 2019). Based on our and the prior observations, it appears that AXL is a cellular context dependent YAP/TAZ target gene. This is discussed in the revised text.

7. In Supplementary Figure 1, Panel F: I don't understand the PI4,5P₂ blot and it is not discussed in text. How does this antibody work? There is a comment in the discussion but should be mentioned in results. As well as P-YAP and CTGF.

There are a set of monoclonal antibodies have been generated that are specific for PI4,5P₂ and other phosphoinositide species. These monoclonal antibodies bind specifically to the phosphorylated inositol head group and in part to glycerol backbone/acyl chains. These antibodies have been used to specifically detect PI4,5P₂ in cells (Chen *et al.*, 2022; Choi *et al.*, 2019; Wang & Sheetz, 2022). There are diverse PI4,5P₂ effectors that bind to PI4,5P₂ and their functions are regulated by PI4,5P₂ binding. Some of these PI4,5P₂ effectors retain associated with PI4,5P₂ even after SDS-PAGE (Fukami *et al*, 1994; Fukami *et al*, 1988; Iannotti *et al*, 2014; Toska *et al*, 2012; Yildirim *et al*, 2013; Zheng *et al*, 2018), but neither we nor other investigators in the field have yet delineated the mechanisms by which the PI4,5P₂ interaction with PI4,5P₂ effectors is resistant to SDS-PAGE.

We initially concluded that the YAP-PI4,5P₂ interaction is transient as the interaction was not resistant to SDS-PAGE. However, during revision we found that a *myo*-inositol probe which is metabolically incorporated into phosphoinositides in cells is strongly associated with the YAP complex and resistant to SDS-PAGE (Fig. 4B, 4I and Supplementary Fig. 4A-4C). Please see our responses to question #2 of Referee #2. For consistency and clarification, we removed the original PI4,5P₂ blot from Supplementary Fig. 2B. YAP phosphorylation and CTGF expression are discussed in the revised text.

8. In Supplementary Figure 1, Panel G. Is this a positive control for LPA treatment? It is not discussed in text.

This is discussed in the revised text.

9. In Figure 2 a TEAD-luciferase assay would complement/support data.

Using the 8xGTIIC YAP/TAZ luciferase reporter (Dupont *et al*, 2011), the YAP/TAZ-TEAD transcriptional activity was measured, and we found that knockdown of PIPKI α or IPMK significantly reduces reporter activity (Supplementary Fig. 1B).

10. In Supplemental Figure 2, all these inhibitors have major off target effects and really add nothing to the manuscript and actually pose more complications in data interpretation. For example, the use of AT9283 makes no sense. "AT9283 is a multi-targeted kinase inhibitor with potent activity against Aurora A/B, JAK2/3, Abl (T315I) and Flt3 (IC₅₀s ranging from 1 to 30 nM). AT9283 inhibits growth and survival of multiple solid tumors in vitro and in vivo." More explanations needed. Also, even though ISA-2011B does binds with high affinity to PIPKI α and inhibits its protein expression and cancer growth in multiple models it does have significant off-target effects, including potent binding to multiple other kinases, including the class IA PI3K p110 α .

This is an important point. The literature (Adhikari & Counter, 2018; Chen *et al.*, 2022; Choi *et al.*, 2019; Choi *et al.*, 2016; East *et al*, 2020; Sarwar *et al*, 2019; Semenas *et al*, 2014) indicates that PIPKI α is a key drug target in cancer. Unfortunately, however, currently there is no available agent that can effectively target PIPKI α . ISA-2011B is a first reported anti-PIPKI α agent but it has poor potency. We tested the IC₅₀ of ISA-2011B using an *in vitro* kinase assay and it was greater than 300 nM. Selectivity is another potential problem of ISA-2011B as the reviewer pointed out. In the original report (Semenas *et al.*, 2014), ISA-2011B was shown to bind to other kinases including MARK1/4, MET, PIK3CA, and AURKB with similar affinity of PIPKI α . To find better agents against PIPKI α , we performed an *in vitro* kinase assay-based screen using known kinase inhibitor libraries. Out of this screen, we found the multikinase inhibitor AT9283 has a decent IC₅₀ against PIPKI α . We believe that AT9283 would be a lead compound for discovering potent and selective anti-PIPKI α agents. The data presented in the

original Supplementary Fig 2A-2C were supportive of our mechanism but were removed in the revised figure and text to prevent any complication.

11. This may be beyond the scope of EMBO, but it is concerning that only exogenous supplementation of lipids and PIP beads are used to define lipid binding to YAP, can authors further validate using better more quantitative assays such as ITC, EMSA, NMR or even computational modeling?

We used the energy minimization calculation approach to model phosphoinositide binding sites on YAP using reported YAP crystal structures (Kaan *et al*, 2017; Li *et al*, 2010). We found that the phosphoinositide binding region we defined in this study locates at the very end of the reported structures and thus the computational modeling approach is not applicable. To the best of our knowledge, no full-length YAP crystal structure is available currently.

Although we have not measured binding affinity parameters of YAP with phosphoinositides using the suggested quantitative assays, we instead monitored impacts of PI4,5P₂ and PI3,4,5P₃ on mediating YAP binding to TEAD1 with increasing amount of lipids *in vitro* (Fig. 5D, 5E). TEAD1 binding to YAP increased and then saturated with an increasing amount of PI4,5P₂ and PI3,4,5P₃, indicating specific interactions rather than concentration-dependent non-specific interactions.

Beyond the scope of our current study, we will further collaborate with structural biologists in the field to elucidate molecular details how phosphoinositides regulate the YAP/TAZ-TEAD interactions.

12. Figure 3, there is no discussion of why YAP would bind PI5P.

Phosphoinositide interactions with polybasic motifs (PBMs) on phosphoinositide binding proteins are mainly mediated by electrostatic interactions of the negatively charged inositol head group of phosphoinositides with positively charged amino acids in the PBMs. Thus, the PBM-phosphoinositide interactions tend to be multivalent and promiscuous. In other words, one PBM interacts with multiple phosphoinositide species. Examples include IQGAP1 (Choi *et al*, 2013), KRAS4B (Lakshman *et al*, 2019), cofilin (Prakash *et al*, 2023), ion channels (Bernier *et al*, 2012; Hille *et al*, 2015), and p53 (Chen *et al.*, 2022; Choi *et al.*, 2019).

YAP bound to PI5P (Fig. 4A) and PI5P promoted YAP binding to TEAD *in vitro* (Fig. 6C). These suggest that PI5P might also regulate the YAP/TAZ-TEAD pathway like PI4,5P₂ and PI3,4,5P₃ in cells. PI5P is the least abundant phosphoinositide species which accounts for ~0.5% of total phosphoinositides (Hasegawa *et al*, 2017). By comparison, PI4,5P₂ is the most abundant phosphoinositide species in most cells (Di Paolo & De Camilli, 2006), and PI3,4,5P₃ levels are sharply increased by agonist stimulation (Traynor-Kaplan *et al*, 1989). We consistently observed that nuclear PI4,5P₂ and PI3,4,5P₃ levels are increased by serum stimulation (Rebuttal Fig. 2). The cellular PI5P level is largely regulated via synthesis by PIKfyve (encoded by *PIKFYVE* gene) and turnover (via phosphorylation) by type II phosphatidylinositol phosphate kinases (encoded by *PIP4K2A*, *PIP4K2B*, and *PIP4K2C* genes) (Emerling *et al*, 2013; Hasegawa *et al.*, 2017). Considering very low cellular concentration of PI5P, YAP might be physically associated with PI5P

Rebuttal Fig. 2 Serum starved MDA-MB-231 cells were treated with 10% serum for 1 hour. Cells were fixed with paraformaldehyde and stained for PI4,5P₂ (A), PI3,4,5P₃ (B), and SON, a nuclear speckle marker. Images were taken with a Zeiss Elyra PS.1 super-resolution microscope. 3D images were reconstructed with stochastic optical reconstruction microscopy. PI4,5P₂ and PI3,4,5P₃ signals were increased upon serum stimulation both in the cytoplasm and more dramatically in the nucleus.

metabolizing enzymes if PI5P is a key regulator of the YAP-TEAD signaling. This mechanism ensures the generation of phosphoinositide signals are efficiently utilized by phosphoinositide effectors (Choi *et al.*, 2015; Tan *et al.*, 2015b). To investigate potential roles of PI5P in the YAP/TAZ-TEAD pathway, we tried to find LPXY or PPXY motifs on PI5P metabolizing enzymes. LPXY or PPXY motifs have been reported to mediate YAP/TAZ interactions in many YAP/TAZ binding proteins (Supplementary Fig. 1F and ref. (Ma *et al.*, 2019; Moroishi *et al.*, 2015)). We found that PIKfyve, PIP4K2A, PIP4K2B, and PIP4K2C do not contain LPXY or PPXY motifs. These suggest that the cellular contribution of PI5P in the regulation the YAP-TEAD signaling might be limited (compared to PI4,5P₂ and PI3,4,5P₃) although it potentially could play a role. This is discussed in the discussion section of the revised manuscript.

13. In Figure 3, it is not clear if PI4,5P₂ and YAP interact in the cytoplasm from the PLA data? Can you explain? And if they do, where would this occur?

Several different cytoplasmic YAP localizations have been reported. These include the plasma membrane (Rausch & Hansen, 2020), tight junction (Goswami *et al.*, 2021), the actin cytoskeleton (Mason *et al.*, 2019), recycling endosomes (Matsudaira *et al.*, 2017), endoplasmic reticulum (Wu *et al.*, 2015), and autophagic vesicles (Jin *et al.*, 2021). Consistent with these reports, we observed that YAP is localized throughout the cytoplasmic area (Fig. 2C, 2D). Interestingly, PI4,5P₂ and its metabolizing enzymes have also been reported to localize in these same YAP localizing cytoplasmic areas (Chang *et al.*, 2013; Choi *et al.*, 2015; Insall & Weiner, 2001; Tan *et al.*, 2015b; Tan *et al.*, 2016; Thapa & Anderson, 2012). Thus, it is not surprising YAP associates with PI4,5P₂ in the cytoplasmic area.

14. In Figure 5, as a control, do these YAP mutants still bind PIPKI α and IMPK? Also, does the localization of YAP change?

The phosphoinositide-binding defective 2Q and 3Q mutants of YAP indistinguishably localized in MDA-MB-231 cells grown in 10% serum-containing condition compared to wild type YAP (Supplementary Fig. 6A). The binding of 2Q and 3Q mutants to PIPKI α and IPMK was similar to wild type YAP *in vitro* (Supplementary Fig. 6B, 6C). Thus, the attenuation of YAP/TAZ-TEAD signaling by depletion of PIPKI α and IPMK was not due to altered nuclear localization (Supplementary Fig. 6A) or the failure of the 2Q and 3Q mutants to bind PIPKI α and IPMK (Supplementary Fig. 6B, 6C).

15. In the discussion the results are not completely discussed therefore it is difficult to understand some of the author's conclusions. For example, a major issue that needs to be addressed is that in the discussion it states that the YAP-PIPKI α /IPMK complexes form and then they translocate to nucleus. There is no evidence of this to support this statement. Can this be explained or validated to support the text?

As described in our responses to question #3 of Referee #1 above, we now changed our working model based on the new data (Fig. 2C, 2D, 7I and Supplementary Fig. 2G). In summary, YAP/TAZ preforms a complex with PIPKI α /IPMK in the nucleus, and YAP translocates into the nucleus upon agonist stimulation to form a ternary complex.

Referee #2:

Jung, et al. provide a compelling connection of PIPK1 α and IPMK to breast cancer motility, which is proposed to occur through a new nuclear signaling pathway. The authors show PIPK1 α and IPMK co-IP with YAP/TAZ, where PIPK1 α is proposed to locally generate PI4,5P₂ from PI4P (with the PI4P presumably associated with YAP/TAZ constitutively or the mechanism doesn't make much sense), the PI4,5P₂ is then acted on by IPMK to generate PI3,4,5P₃. The PI3,4,5P₃ is proposed to mediate direct inter-protein interactions between YAP and TEAD.

There are some solid aspects of the manuscript, such as some of the PLA data and co-IPs indicating PIPK1 α -IPMK and YAP exist in the same cellular complexes, which is dependent on LPXY motif. siRNA of PIPK1 α and IPMK reduce YAP-TEAD interaction, and that siPI3K had no effect if encouraging, but overall many of the conclusions are based on cell biological experiments that cannot be made based on cell biology, but require biochemical experiments and/or metabolic labeling. Thus, major revisions are required:

1. IPMK generates inositol phosphates that have diverse effects on transcription. It is unclear which IPMK product is affecting the YAP/TAZ pathway, but certainly the authors cannot conclude, as they do, that it is PI3,4,5P₃ that is the product responsible for the effects. This critique could be addressed by simply changing the language.

IPMK has both inositol phosphate and phosphoinositide kinase activities. The kinase dead mutant (Fu *et al.*, 2018; Maag *et al.*, 2011) used in Fig. 3F is defective of both kinase activities, thus the result does not answer which IPMK product is responsible for the regulation of the YAP/TAZ-TEAD pathway. To this end, we performed *in vitro* binding assays between YAP and TEAD1 in the presence of all known IPMK enzymatic products. IPMK is reported to generate inositol 1,3,4,5-tetrakisphosphate (Ins(1,3,4,5)P₄), Ins(1,4,5,6)P₄, Ins(1,3,4,5,6)P₅, and PI3,4,5P₃ (Lee *et al.*, 2021). As shown in Fig. 5F, Ins(1,4,5)P₃, Ins(1,3,4,5)P₄, Ins(1,4,5,6)P₄, Ins(1,3,4,5,6)P₅, and Ins(1,2,3,4,5,6)P₆ had no significant impacts on facilitating TEAD1 binding to YAP *in vitro*, while PI4,5P₂ and PI3,4,5P₃ dramatically increased the binding. These data indicate that amongst IPMK enzymatic products, PI3,4,5P₃ specifically regulates the YAP/TAZ-TEAD pathway. As other PI3-kinases tested (p110 α and p110 β) had no impact (Fig. 3B), the data further demonstrate that the physical interaction of IPMK with YAP is important for the regulation via its PI3-kinase activity in cells.

2. The approaches used in this manuscript rely almost exclusively on antibodies, some of which are notoriously non-specific, such as the PI4,5P₂ and PI3,4,5P₃ antibodies. It would be more convincing if ³H-inositol was used to metabolically label the cells, then show specific enrichment of the label with epitope-tagged YAP/TAZ immunoprecipitates in a scintillation counter, but not other transcription factors that are activated by serum, then show decreases to specific ³H-inositol enrichment upon PIPK1 α and IPMK knockdown. HPLC of the immunoprecipitated and deacylated glycerolipid headgroups would prove the radiolabel was PIPs and not inositol phosphates.

Instead of the suggested ³H-inositol labelling method, we utilized an alternative metabolic labelling method to detect the association of phosphoinositides with the YAP-TEAD complex. The use of an alternative approach is simply due to our lack of expertise in the ³H-inositol method. Dr. Mike Best's group previously reported a clickable *myo*-inositol probe (Acetylated 2-azidopropylinositol (Ac₃2API)) that can be metabolically incorporated into phosphatidylinositol and phosphoinositides in cells as outlined in Supplementary Fig. 4A (Ricks *et al.*, 2019). An azide tag was linked through a propyl ether at the 2-position that can be further linked to an alkyne-biotin after copper-catalyzed click reaction. Thus, the biotinylated cellular phosphatidylinositol and phosphoinositides can be detected by streptavidin without the use of phosphoinositide antibodies.

For this, starved MDA-MB-231 cells were fed with Ac₃2API for 24 hours in the presence of 10% dialyzed serum (inositol free). Cells were harvested and cell lysates were prepared. The 2-azido *myo*-inositol 2API was conjugated with biotin using an alkyne-biotin after the click reaction. YAP or TEAD was immunoprecipitated from these click'ed lysates and resolved by SDS-PAGE. Biotinylated 2API

was detected by streptavidin and YAP and TEAD were detected by immunoblotting (Fig. 4B). We observed a clear streptavidin signal at the size of YAP and a relatively weak signal at the size of TEAD. This indicates that metabolically labelled 2API is resistant to SDS-PAGE and tightly associated with the YAP-TEAD complex. We previously showed that mutant p53 is associated with several phosphoinositide species (Chen *et al.*, 2022; Choi *et al.*, 2019). We could detect the biotinylated 2API signal at the size of p53 (Rebuttal Fig. 3), further validating that the clickable *myo*-inositol probe is working in the metabolic labelling approach. By comparison, cellular Myc (c-Myc) which is another serum-activated transcription factor failed to show a strong association with biotinylated 2API (Supplementary Fig. 4B). Additionally, knockdown of PIPK1 α and IPMK substantially reduced the association of YAP with the biotinylated 2API (Fig. 4I), further implying that PI4,5P₂ and PI3,4,5P₃ generated by PIPK1 α and IPMK are major phosphoinositide species forming strong associations with the YAP complex in cells.

Figure for reviewers removed

We observed the biotinylated 2API probe remains associated with YAP, TEAD, and p53 after SDS-PAGE (Fig. 4B, 4I and Rebuttal Fig. 3). This is consistent with previous observations that some of phosphoinositide binding proteins remain associated with phosphoinositides even after SDS-PAGE (Chen *et al.*, 2022; Choi *et al.*, 2019; Fukami *et al.*, 1994; Fukami *et al.*, 1988; Iannotti *et al.*, 2014; Toska *et al.*, 2012; Yildirim *et al.*, 2013; Zheng *et al.*, 2018). An assumption is that phosphoinositide-effector interactions are covalent (Fukami *et al.*, 1994; Fukami *et al.*, 1988) but neither we nor other investigators in the field have yet delineated the mechanisms by which the phosphoinositide interaction with phosphoinositide effectors is resistant to SDS-PAGE.

Intrinsically the metabolic labelling \rightarrow click approach we used for this study does not distinguish which *myo*-inositol containing moiety (phosphoinositides vs. inositol phosphates) is associated with the YAP-TEAD complex. However, our *in vitro* binding data in Fig. 5F directly point out that PI4,5P₂ and PI3,4,5P₃ but not soluble inositol phosphates mediate TEAD1 binding to YAP. Thus, it is likely that *myo*-inositol which associates with the YAP-TEAD complex in cells would be phosphoinositides.

Although some criticism has been raised about the PI4,5P₂ and PI3,4,5P₃ antibodies used in this study, these reagents have been extensively validated by independent research groups in the field. For example, PI4,5P₂ and PI3,4,5P₃ immunostaining signals were diminished by the addition of excess Ins(1,4,5)P₃ or Ins(1,3,4,5)P₄ (Chen *et al.*, 2022; Hammond *et al.*, 2009; Thapa *et al.*, 2020). Their immunostaining signals overlapped (although not completely) with other PI4,5P₂ and PI3,4,5P₃ biosensors (Chierico *et al.*, 2014; Choi *et al.*, 2013; Yip *et al.*, 2008), which dynamically changed in cells following extracellular stimuli or stress (Bura & Jurak Begonja, 2021; Fukami *et al.*, 1988; Kim *et al.*, 2022; Wang *et al.*, 2017; Wang & Sheetz, 2022). Finally, the levels of staining were changed by inhibition of phosphoinositide kinases and phosphatases (Semenas *et al.*, 2014; Thapa *et al.*, 2020). To further validate the PI4,5P₂ and PI3,4,5P₃ antibodies, we compared their immunoblotting signals with the biotinylated 2API signal using click'ed lysates. Immunoblotting detection of the phosphoinositide binding proteins was validated previously (Carrillo *et al.*, 2023). As shown in Supplementary Fig. 4C, many of bands recognized by streptavidin overlapped with the bands recognized by the antibodies to PI4,5P₂ and PI3,4,5P₃, supporting the functionality of the PI4,5P₂ and PI3,4,5P₃ antibodies. Taken together, PI4,5P₂ and PI3,4,5P₃ antibodies (despite the reputation) are useful tools for detecting phosphoinositides in cells and this is supported by other non-antibody related techniques.

3. Fig 3D-E are troubling. The PIPK1 α and IPMK association with YAP/TAZ suggest the PIPK1 α kinase

substrate PI4P must be associated with YAP/TAZ to generate PI4,5P₂ and only then can IPMK make the PI3,4,5P₃. Thus, knockdown of IPMK should not affect PI4,5P₂ generation in the PLA, but it does (Figure 3D). Further, knock down of PIPK1 α should eliminate all PI3,4,5P₃ by decreasing the substrate, but it does not (Figure 3E). These results need to be back up with metabolic labeling studies that demonstrate the signal being observed is actually a phosphoinositide, and not just antibody cross reactivity of these weak antibody reagents.

This is an important point but, unfortunately, we do not have clear explanation for our observation. As the reviewer pointed out, PI4P is seemingly associated with the YAP/TAZ-TEAD complex. Natural source PI4P slightly but statistically significantly increased TEAD1 binding to YAP (Fig. 5B). Additionally, natural source PI4P dramatically (as much as natural source PI4,5P₂) increased TEAD1 binding to TAZ (Supplementary Fig. 5D). The PLA data in Fig. 4E-4H clearly showed a decrease in the YAP-PI4,5P₂ and YAP-PI3,4,5P₃ associations upon knockdown of PIPK1 α . But knockdown of IPMK unexpectedly showed a decrease in the YAP-PI4,5P₂ PLA signal. This indicates that the generation of YAP-phosphoinositides complexes may not be a solely linear (PI4P \rightarrow PI4,5P₂ \rightarrow PI3,4,5P₃) process. Rather, PI4,5P₂ and PI3,4,5P₃ generated by PIPK1 α and IPMK appear to synergically regulate the YAP/TAZ-TEAD complex. Our *in vitro* binding assays with different concentrations of phosphoinositides showed TEAD1 binding to YAP was saturated with PI4,5P₂ and PI3,4,5P₃ concentrations greater than 0.1 μ M (Fig. 5D, 5E). When both PI4,5P₂ and PI3,4,5P₃ were present at a 0.1 μ M concentration, the YAP-TEAD1 binding was further increased (although subtly) compared to PI4,5P₂ and PI3,4,5P₃ alone (Fig. 5F). This raises a possibility that PIPK1 α , IPMK, YAP, and TEAD form a ternary protein complex which is maintained by both PI4,5P₂ and PI3,4,5P₃ simultaneously in cells. In this model, knockdown of one kinase (leading to the decreased association of one phosphoinositide species with the complex) can result in attenuating the association of the other kinase and the other phosphoinositide species with the complex as shown in Fig. 4E-4H. In line with this possibility, knockdown of IPMK reduced PIPK1 α co-immunoprecipitation with YAP (Rebuttal Fig. 4). We plan to further investigate this hypothesis in cells via more sophisticated approaches by combining the proximity ligation assays with the clickable *myo*-inositol probe, but this is beyond the scope of the current study.

Again, as detailed in our answer to the question #2 of Referee #2, we showed that a *myo*-inositol probe can be metabolically incorporated into the YAP-TEAD complex and this incorporation was dramatically reduced by knockdown of either PIPK1 α or IPMK (Fig. 4B, 4I and Supplementary Fig. 4A-4C).

4. That short chain diC8 lipids had no effect and all long chain phosphoinositides had an effect suggests the stickiest part of the phospholipid is mediating the effect, and no evidence is provided that the effect is specific, e.g. saturable to a single site on YAP. A specific, saturable and stoichiometric interaction (1:1, 2:1, 2:2, etc) must be demonstrated with KDs determined, otherwise no convincing evidence is provided that the phosphoinositides are specifically regulating the complex.

Although we have not measured binding affinity parameters of YAP with phosphoinositides using quantitative assays such as isothermal calorimetry suggested by Referee #1, we instead monitored impacts of PI4,5P₂ and PI3,4,5P₃ on mediating YAP binding to TEAD with an increasing amount of lipids *in vitro* (Fig. 5D, 5E). TEAD binding to YAP was increased and then saturated with an increasing amount of PI4,5P₂ and PI3,4,5P₃, indicating specific interactions rather than concentration-dependent non-specific interactions.

Rebuttal Fig. 4 IPMK was transiently knocked down in MDA-MB-231 cells by transfecting siRNAs. As a control, non-targeting scrambled siRNAs were transfected. YAP was immunoprecipitated and YAP-associated proteins were analyzed by immunoblotting with the indicated antibodies.

Referee #3:

The Hippo pathway is essential for development, organ size regulation, and tissue homeostasis, and its dysregulation contributes to cancer development. The YAP/TAZ transcriptional co-activators bind with TEA domain (TEAD) family transcription factors to promote gene expression, cell growth, and cancer progression. Previous studies have revealed the critical role of plasma membrane phosphatidylinositol-4-phosphate in hippo regulation through phosphatidylinositol transfer proteins α and β (PITP α/β) (Li, F. L., Fu, V., Liu, G., Tang, T., Konradi, A. W., Peng, X., ... & Guan, K. L. (2022). Hippo pathway regulation by phosphatidylinositol transfer protein and phosphoinositides. *Nature Chemical Biology*, 18(10), 1076-1086.). The current study by Oisun Jung et al. proposed a new mechanistic scheme that nuclear phosphoinositides generated by PIPK1 α and IPMK regulate the spatiotemporal binding of YAP/TAZ to TEADs. PIPK1 α and IPMK inducibly associate with YAP/TAZ via LPXY motifs on PIPK1 α and IPMK and the WW domains on YAP upon agonist activation of membrane receptors. The complex is translocated to the nucleus, where PIPK1 α and IPMK produce PI4,5P₂ and PI3,4,5P₃. PI4,5P₂ and PI3,4,5P₃ are transferred to YAP/TAZ and bind to YAP/TAZ. The binding of PI4,5P₂ and PI3,4,5P₃ to YAP/TAZ facilitates their association with TEADs, resulting in the transcription of YAP/TAZ target genes. The studies are interesting; however, the direct evidence of PI4,5P₂ and PI3,4,5P₃ binding to YAP/TAZ is based on *in vitro* assay with high concentrations of PI4,5P₂ and PI3,4,5P₃ (The results are not very significant though in Figure 4C) or immunofluorescent staining to examine the co-localization, which often results in false positive. It would be critical to examine the existence of the complex of PI4,5P₂, PI3,4,5P₃ and YAP/TAZ in cells by Mass Spectrometry or other techniques. Besides, the manuscript needs to be improved to address several technical and experimental issues before considering published on The EMBO Journal.

As detailed in our responses to question #2 of Referee #2, we utilized a metabolic labelling method with a clickable *myo*-inositol probe to further validate the association of phosphoinositides with the YAP-TEAD complex. In summary, we could clearly detect the *myo*-inositol probe in the YAP or TEAD immunoprecipitation (Fig. 4B and Supplementary Fig. 4A). The association of the *myo*-inositol probe with YAP was largely reduced by knockdown of PIPK1 α and IPMK (Fig. 4I). Along with *in vitro* binding and PLA data (Fig. 4-6), our new data with the click approach now further and consistently validates the association of phosphoinositides with the YAP/TAZ-TEAD complex *in vitro* and *in vivo*.

Additionally, as fully described in question #3 of Referee #2, we monitored the impact of PI4,5P₂ and PI3,4,5P₃ in mediating YAP binding to TEAD with an increasing amount of lipids *in vitro* (Fig. 5D, 5E). In summary, TEAD binding to YAP was increased and then saturated with an increasing amount of PI4,5P₂ and PI3,4,5P₃, indicating specific interactions rather than concentration-dependent non-specific interactions.

1. In Figure 1F, the signal of YAP in the input is relatively weak, it would be beneficial to run the blots of input and pull-down separately and have a better image.

The input blot of Flag-tagged YAP was changed as the reviewer suggested.

2. In Figure 1G, Myc-IPMK has two bands, and both bind to YAP, is the up band the posttranslational modification form of IPMK?

Only one IPMK isoform has been reported (<https://www.ncbi.nlm.nih.gov/gene/253430>) and thus the top band in the immunoblot potentially represents the posttranslational modification. To the best of our knowledge, there is no previous report about posttranslational modifications of IPMK. To this end, IPMK immunoprecipitations were analyzed with phosphorylation specific antibodies. As shown in **Rebuttal Fig. 5B**, phosphorylation on serine residues of IPMK was subtly increased by serum, while tyrosine phosphorylation remained largely unchanged.

3. In Figure 1H, both PIPK1 α and IPMK showed two bands, are the up bands the posttranslational modification forms or isoforms of PIPK1 α and IPMK?

At least 5 isoforms of PIPKI α were reported (<https://www.ncbi.nlm.nih.gov/gene/8394>) but posttranslational modification of PIPKI α has not been fully investigated. Our data in **Rebuttal Fig. 5A** showed that serine phosphorylation of PIPKI α was increased by serum stimulation, while tyrosine phosphorylation was decreased. The double (or multiple) bands in **Supplementary Fig. 2A** (Fig. 1H in the original submission) thus likely represent both posttranslational modification and isoforms.

As described in our responses to question #2 of Referee #3, the multiple bands may represent posttranslational modification of IPMK but we also observed antibody specific band patterning. This is described in detail below at our responses to question #6.

4. In Figure 1I, the IPMK-YAP PLA signal was also significantly increased in the cytosol. What's the explanation given that the interaction happens in the nucleus?

IPMK was predominantly localized in the nucleus (**Rebuttal Fig. 1**) and serum stimulation did not alter its nuclear localization (**Fig. 2D**). Nuclear functions of IPMK are well documented. IPMK regulates key nuclear events such as transcription (Blind *et al.*, 2012; Guha *et al.*, 2019; Xu *et al.*, 2013), DNA damage repair (Wang *et al.*, 2017), mRNA export (Wickramasinghe *et al.*, 2013), and transcription factor stability (Fu *et al.*, 2018). Although largely found in the nucleus, nucleo-cytoplasmic shuttling of IPMK has also been reported. In response to autophagy-inducing stress signals, a subset of IPMK is exported out of the nucleus and inhibits autophagy (Chen *et al.*, 2020). Protein kinase CK2-mediated phosphorylation is reported to promote cytoplasmic retention of IPMK (Meyer *et al.*, 2012). Wnt3a treatment induces plasma membrane localization of IPMK (Wang & Wang, 2012). In line with these reports, we also observed IPMK localization at the perinuclear area and the plasma membrane in serum-stimulated condition (**Rebuttal Fig. 1**).

Several different cytoplasmic YAP localizations have been reported. These include the plasma membrane (Rausch & Hansen, 2020), tight junctions (Goswami *et al.*, 2021), the actin cytoskeleton (Mason *et al.*, 2019), recycling endosomes (Matsudaira *et al.*, 2017), endoplasmic reticulum (Wu *et al.*, 2015), and autophagic vesicles (Jin *et al.*, 2021). Consistent with these reports, we observed that YAP is localized throughout the cytoplasmic area (**Fig. 2C, 2D**). Based on these, IPMK potentially interacts with YAP in the various cytoplasmic locations including the plasma membrane and intracellular vesicles.

5. In Figure 2A, PIPKI α and PIPKI γ showed compensatory effects, the knockdown of PIPKI α increased PIPKI γ and vice versa. It's interesting to see the knockdown of PIPKI γ increased protein level of PIPKI α and TEAD-YAP binding, but it's hard to see the binding of TEAD-TAZ given the blot is a little overexposed, it would be interesting to see how modulation of PIPKI α and PIPKI γ affects TEAD-YAP as well as TEAD-TAZ binding in more details.

We consistently observed that knockdown of one PIPKI isoform increased the expression of other isoforms in MDA-MB-231 cells (Choi *et al.*, 2019; Choi *et al.*, 2016; Choi *et al.*, 2013). Knockdown of PIPKI γ increased PIPKI α expression (**Fig. 3A**) and significantly increased YAP coimmunoprecipitation with TEAD (**Supplementary Fig. 3A**). The TAZ-TEAD interaction was not further increased by knockdown of PIPKI γ (**Supplementary Fig. 3A**). This indicates that endogenous level of PIPKI α is sufficient to maintain the maximal interaction of TAZ with TEAD, while the YAP-TEAD interaction is dependent on PIPKI α expression in MDA-MB-231 cells.

Rebuttal Fig. 5 Serum starved MDA-MB-231 cells were treated with 10% serum for 1 hour. Cells were lysed and endogenous PIPKI α (**A**) or IPMK (**B**) were immunoprecipitated. The PIPKI α and IPMK complex was resolved by SDS-PAGE and further analyzed by immunoblotting with phosphorylation specific antibodies.

6. The authors showed two bands for PIPKI α and IPMK in some blots (Figure 1H, 2A), while in some blots, there is only one band (Figure 2E, 2F), what's the matter in this case?

As discussed in questions #2 and #3 of Referee #3, the multiple bands likely represent isoforms and posttranslational modifications. These multiple immunoblot bands were more clearly seen when separated longer in SDS-PAGE and overexposed (Fig. 1A, 3A, 3B, 3F and Supplementary Fig. 2G). In shorter separation and/or exposure, seemingly single band was observed (Fig. 3E, 3F and Supplementary Fig. 1A, 2B, 2D).

Beyond the separation and exposure issue, the IPMK antibodies we used for immunoblotting showed different staining patterns. We used three anti-IPMK antibodies for immunoblotting in this study: MilliporeSigma (HPA037837), Invitrogen (PA5-21629), and a homemade antibody gifted from Dr. Seyun Kim (KAIST, South Korea) (Beon *et al*, 2022; Min *et al*, 2022). The MilliporeSigma IPMK antibody showed multiple non-specific bands, but a band located between 55 and 40 kDa was decreased upon depletion of IPMK (Fig. 1A, 3A, 3B, 3F). The Invitrogen IPMK antibody showed a relatively specific (one band) immunoblot signal between 55 and 40 kDa (Supplementary Fig. 2G). The IPMK antibody from Dr. Kim's group showed multiple bands between 70 and 40 kDa (Supplementary Fig. 2A, 2B). We have confirmed that these IPMK immunoblot bands were diminished upon depletion of IPMK in MDA-MB-231 cells.

7. In Figure 1F, the blot for input IPMK is not clear.

The input blot of Flag-tagged YAP was changed as Referee #1 and #3 suggested (Fig. 1F).

8. In Figure 3A, it would benefit the readers if the author could include the blots of unbound fractions in each treatment.

We generated the 1Q mutant (question #11 of Referee #3) and repeated the binding assays with PI_{4,5}P₂ and PI_{3,4,5}P₃ beads (Fig. 6B). Both bound (beads) and unbound (supernatant) fractions were immunoblotted, and we found that a similar amount of GST-YAP was detected in the unbound fraction. This indicates that only minor fraction of GST-YAP binds with the phosphoinositide beads. We found that an inaccurate concentration of GST-YAP proteins was inadvertently indicated the figure legend of original submission and the error is corrected (to 0.5 μ M) in the revised text.

9. In Figure 3B, serum stimulation decreased cytosolic PI_{4,5}P₂-YAP PLA, while increasing its nuclear fraction. While in Figure 3B, serum stimulation increased both cytosolic and nuclear PI_{3,4,5}P₃-YAP PLA, does serum stimulation increase total PI_{3,4,5}P₃, not PI_{4,5}P₂? What's the potential explanation?

As shown in Rebuttal Fig. 2, PI_{4,5}P₂ and PI_{3,4,5}P₃ levels were increased by serum stimulation both in the cytoplasm and in the nucleus (please see the 3D reconstructed images). It appears that the increase of PI_{3,4,5}P₃ by serum is more robust than PI_{4,5}P₂. This is consistent with prior reports (Insall & Weiner, 2001; Stephens *et al*, 1993). The reduction of PI_{4,5}P₂-YAP PLA numbers in the cytoplasm by serum stimulation can be explained by the sharp translocation of YAP into the nucleus (Fig. 2C, 2D). The increase of cytoplasmic PI_{3,4,5}P₃-YAP PLA numbers can be explained by the robust accumulation of PI_{3,4,5}P₃ in the cytoplasm by serum (Rebuttal Fig. 2B).

10. The authors indicated, "The impact of natural source PI₅P and PI_{3,4,5}P₃ could not be tested as they are not commercially available.", however, in Figure 4C, natural PI_{3,4,5}P₃ has been tested though without effects on YAP-TEAD1 binding.

The way we labeled the phosphoinositides in the original figure caused the confusion. In the revised figure (Fig. 5B), the labeling of the phosphoinositides was changed to be vertical for clearer indication of the reagents that were used.

11. In Figure 5B, it looks like Lys97 is the most important site for YAP binding to PI_{4,5}P₂ and PI_{3,4,5}P₃; it would be interesting to test how the single mutation affects the binding of phosphoinositides to YAP.

We mutated the lysine 97 residue in the 1Q mutant (Fig. 6A) and tested for phosphoinositide binding using PI4,5P₂ and PI3,4,5P₃ beads. The 1Q mutant showed a partial reduction of PI4,5P₂ and PI3,4,5P₃ binding, while the 3Q mutant completely lost the binding *in vitro* (Fig. 6B). Phosphoinositide interactions with polybasic motifs (PBMs) on phosphoinositide binding proteins are mainly mediated by electrostatic interactions of the negatively charged inositol head group of phosphoinositides with the positively charged amino acids in the PBMs (Bernier *et al.*, 2012; Choi *et al.*, 2015; Di Paolo & De Camilli, 2006; Itoh & Takenawa, 2002). Thus, the PBM-phosphoinositide interactions tend to be multivalent and cooperative. In other words, mutating a single basic residue in the PBM often has only limited impact on phosphoinositide binding, while a complete disruption requires mutating multiple basic residues at a time. These canonical PBM-phosphoinositide interactions were reported in N-WASP (Papayannopoulos *et al.*, 2005), LAPTM4B (Tan *et al.*, 2015a), and IQGAP1 (Choi *et al.*, 2013), and now shown in YAP (Fig. 6A, 6B).

12. Figure 5B and 5C used TEAD1 to examine how the binding of phosphoinositides to YAP affects TEAD1-YAP binding *in vitro*. Why do the authors switch to TEAD4-YAP in the cell-based assay?

All four TEAD isoforms (TEAD1-4) have high sequence homology in their DNA-binding domains (DBD) and their YAP/TAZ-binding domains (Y/TBD) (Supplementary Fig. 5A). Despite the high sequence homology, TEAD paralogs show tissue and development stage specific expression patterns (Currey *et al.*, 2021; Holden & Cunningham, 2018), and it remains to be identified how the TEAD paralogs complement one another in various pathophysiological conditions. We showed that the nuclear phosphoinositide signaling pathway is critical for the pan-TEAD interaction with YAP/TAZ in cells (Fig. 3) and TEAD1 binding to YAP *in vitro* (Fig. 5, 6). To show the generality of our study, we tried a different TEAD paralog (i.e. TEAD4) in the cell-based assay (Fig. 6D). Our data pointed out that YAP interactions with at least two TEAD paralogs are regulated by the phosphoinositide signaling.

Reference cited:

- Adhikari H, Counter CM (2018) Interrogating the protein interactomes of RAS isoforms identifies PIP5K1A as a KRAS-specific vulnerability. *Nat Commun* 9: 3646
- Andrade D, Mehta M, Griffith J, Panneerselvam J, Srivastava A, Kim TD, Janknecht R, Herman T, Ramesh R, Munshi A (2017) YAP1 inhibition radiosensitizes triple negative breast cancer cells by targeting the DNA damage response and cell survival pathways. *Oncotarget* 8: 98495-98508
- Beon J, Han S, Yang H, Park SE, Hyun K, Lee SY, Rhee HW, Seo JK, Kim J, Kim S *et al* (2022) Inositol polyphosphate multikinase physically binds to the SWI/SNF complex and modulates BRG1 occupancy in mouse embryonic stem cells. *Elife* 11
- Bernier LP, Blais D, Boue-Grabot E, Seguela P (2012) A dual polybasic motif determines phosphoinositide binding and regulation in the P2X channel family. *PLoS One* 7: e40595
- Blind RD, Suzawa M, Ingraham HA (2012) Direct modification and activation of a nuclear receptor-PIP(2) complex by the inositol lipid kinase IPMK. *Sci Signal* 5: ra44
- Bura A, Jurak Begonja A (2021) Imaging of Intracellular and Plasma Membrane Pools of PI(4,5)P(2) and PI4P in Human Platelets. *Life (Basel)* 11
- Carrillo ND, Chen M, Cryns VL, Anderson RA (2023) Lipid transfer proteins initiate nuclear phosphoinositide signaling. *bioRxiv*
- Chang CL, Hsieh TS, Yang TT, Rothberg KG, Azizoglu DB, Volk E, Liao JC, Liou J (2013) Feedback regulation of receptor-induced Ca²⁺ signaling mediated by E-Syt1 and Nir2 at endoplasmic reticulum-plasma membrane junctions. *Cell Rep* 5: 813-825
- Chen D, Wang Z, Zhao YG, Zheng H, Zhao H, Liu N, Zhang H (2020) Inositol Polyphosphate Multikinase Inhibits Liquid-Liquid Phase Separation of TFEB to Negatively Regulate Autophagy Activity. *Dev Cell* 55: 588-602 e587
- Chen M, Choi S, Wen T, Chen C, Thapa N, Lee JH, Cryns VL, Anderson RA (2022) A p53-phosphoinositide signalosome regulates nuclear AKT activation. *Nat Cell Biol* 24: 1099-1113
- Chierico L, Joseph AS, Lewis AL, Battaglia G (2014) Live cell imaging of membrane/cytoskeleton interactions and membrane topology. *Sci Rep* 4: 6056
- Choi S, Chen M, Cryns VL, Anderson RA (2019) A nuclear phosphoinositide kinase complex regulates p53. *Nat Cell Biol* 21: 462-475
- Choi S, Hedman AC, Sayedyahosseini S, Thapa N, Sacks DB, Anderson RA (2016) Agonist-stimulated phosphatidylinositol-3,4,5-trisphosphate generation by scaffolded phosphoinositide kinases. *Nat Cell Biol* 18: 1324-1335
- Choi S, Thapa N, Hedman AC, Li Z, Sacks DB, Anderson RA (2013) IQGAP1 is a novel phosphatidylinositol 4,5 bisphosphate effector in regulation of directional cell migration. *EMBO J* 32: 2617-2630
- Choi S, Thapa N, Tan X, Hedman AC, Anderson RA (2015) PIP kinases define PI4,5P(2) signaling specificity by association with effectors. *Biochim Biophys Acta* 1851: 711-723
- Currey L, Thor S, Piper M (2021) TEAD family transcription factors in development and disease. *Development* 148
- Di Paolo G, De Camilli P (2006) Phosphoinositides in cell regulation and membrane dynamics. *Nature* 443: 651-657
- Doughman RL, Firestone AJ, Wojtasiak ML, Bunce MW, Anderson RA (2003) Membrane ruffling requires coordination between type I alpha phosphatidylinositol phosphate kinase and Rac signaling. *J Biol Chem* 278: 23036-23045
- Dupont S, Morsut L, Aragona M, Enzo E, Giulitti S, Cordenonsi M, Zanconato F, Le Digabel J, Forcato M, Bicciato S *et al* (2011) Role of YAP/TAZ in mechanotransduction. *Nature* 474: 179-183
- East MP, Laitinen T, Asquith CRM (2020) PIP5K1A: a potential target for cancers with KRAS or TP53 mutations. *Nat Rev Drug Discov* 19: 436
- Emerling BM, Hurov JB, Poulogiannis G, Tsukazawa KS, Choo-Wing R, Wulf GM, Bell EL, Shim HS, Lamia KA, Rameh LE *et al* (2013) Depletion of a putatively druggable class of phosphatidylinositol kinases inhibits growth of p53-null tumors. *Cell* 155: 844-857

Fu C, Tyagi R, Chin AC, Rojas T, Li RJ, Guha P, Bernstein IA, Rao F, Xu R, Cha JY *et al* (2018) Inositol Polyphosphate Multikinase Inhibits Angiogenesis via Inositol Pentakisphosphate-Induced HIF-1 α Degradation. *Circ Res* 122: 457-472

Fukami K, Endo T, Imamura M, Takenawa T (1994) α -Actinin and vinculin are PIP₂-binding proteins involved in signaling by tyrosine kinase. *J Biol Chem* 269: 1518-1522

Fukami K, Matsuoka K, Nakanishi O, Yamakawa A, Kawai S, Takenawa T (1988) Antibody to phosphatidylinositol 4,5-bisphosphate inhibits oncogene-induced mitogenesis. *Proc Natl Acad Sci U S A* 85: 9057-9061

Gonzales B, de Rocquigny H, Beziau A, Durand S, Burlaud-Gaillard J, Lefebvre A, Krull S, Emond P, Brand D, Piver E (2020) Type I Phosphatidylinositol-4-Phosphate 5-Kinases α and γ Play a Key Role in Targeting HIV-1 Pr55(Gag) to the Plasma Membrane. *J Virol* 94

Goswami S, Balasubramanian I, D'Agostino L, Bandyopadhyay S, Patel R, Avasthi S, Yu S, Goldenring JR, Bonder EM, Gao N (2021) RAB11A-mediated YAP localization to adherens and tight junctions is essential for colonic epithelial integrity. *J Biol Chem* 297: 100848

Guha P, Tyagi R, Chowdhury S, Reilly L, Fu C, Xu R, Resnick AC, Snyder SH (2019) IPMK Mediates Activation of ULK Signaling and Transcriptional Regulation of Autophagy Linked to Liver Inflammation and Regeneration. *Cell Rep* 26: 2692-2703 e2697

Hammond GR, Schiavo G, Irvine RF (2009) Immunocytochemical techniques reveal multiple, distinct cellular pools of PtdIns4P and PtdIns(4,5)P(2). *Biochem J* 422: 23-35

Hasegawa J, Strunk BS, Weisman LS (2017) PI5P and PI(3,5)P(2): Minor, but Essential Phosphoinositides. *Cell Struct Funct* 42: 49-60

Hille B, Dickson EJ, Kruse M, Vivas O, Suh BC (2015) Phosphoinositides regulate ion channels. *Biochim Biophys Acta* 1851: 844-856

Holden JK, Cunningham CN (2018) Targeting the Hippo Pathway and Cancer through the TEAD Family of Transcription Factors. *Cancers (Basel)* 10

Iannotti FA, Silvestri C, Mazzarella E, Martella A, Calvigioni D, Piscitelli F, Ambrosino P, Petrosino S, Czifra G, Biro T *et al* (2014) The endocannabinoid 2-AG controls skeletal muscle cell differentiation via CB1 receptor-dependent inhibition of Kv7 channels. *Proc Natl Acad Sci U S A* 111: E2472-2481

Insall RH, Weiner OD (2001) PIP₃, PIP₂, and cell movement--similar messages, different meanings? *Dev Cell* 1: 743-747

Itoh T, Takenawa T (2002) Phosphoinositide-binding domains: Functional units for temporal and spatial regulation of intracellular signalling. *Cell Signal* 14: 733-743

Jin L, Chen Y, Cheng D, He Z, Shi X, Du B, Xi X, Gao Y, Guo Y (2021) YAP inhibits autophagy and promotes progression of colorectal cancer via upregulating Bcl-2 expression. *Cell Death Dis* 12: 457

Kaan HYK, Chan SW, Tan SKJ, Guo F, Lim CJ, Hong W, Song H (2017) Crystal structure of TAZ-TEAD complex reveals a distinct interaction mode from that of YAP-TEAD complex. *Sci Rep* 7: 2035

Kim OH, Kang GH, Hur J, Lee J, Jung Y, Hong IS, Lee H, Seo SY, Lee DH, Lee CS *et al* (2022) Externalized phosphatidylinositides on apoptotic cells are eat-me signals recognized by CD14. *Cell Death Differ* 29: 1423-1432

King B, Araki J, Palm W, Thompson CB (2020) Yap/Taz promote the scavenging of extracellular nutrients through macropinocytosis. *Genes Dev* 34: 1345-1358

Lakshman B, Messing S, Schmid EM, Clogston JD, Gillette WK, Esposito D, Kessing B, Fletcher DA, Nissley DV, McCormick F *et al* (2019) Quantitative biophysical analysis defines key components modulating recruitment of the GTPase KRAS to the plasma membrane. *J Biol Chem* 294: 2193-2207

Lee B, Park SJ, Hong S, Kim K, Kim S (2021) Inositol Polyphosphate Multikinase Signaling: Multifaceted Functions in Health and Disease. *Mol Cells* 44: 187-194

Li Z, Zhao B, Wang P, Chen F, Dong Z, Yang H, Guan KL, Xu Y (2010) Structural insights into the YAP and TEAD complex. *Genes Dev* 24: 235-240

Ma S, Meng Z, Chen R, Guan KL (2019) The Hippo Pathway: Biology and Pathophysiology. *Annu Rev Biochem* 88: 577-604

Maag D, Maxwell MJ, Hardesty DA, Boucher KL, Choudhari N, Hanno AG, Ma JF, Snowman AS, Pietropaoli JW, Xu R *et al* (2011) Inositol polyphosphate multikinase is a physiologic PI3-kinase that activates Akt/PKB. *Proc Natl Acad Sci U S A* 108: 1391-1396

Mason DE, Collins JM, Dawahare JH, Nguyen TD, Lin Y, Voytik-Harbin SL, Zorlutuna P, Yoder MC, Boerckel JD (2019) YAP and TAZ limit cytoskeletal and focal adhesion maturation to enable persistent cell motility. *J Cell Biol* 218: 1369-1389

Matsudaira T, Mukai K, Noguchi T, Hasegawa J, Hatta T, Iemura SI, Natsume T, Miyamura N, Nishina H, Nakayama J *et al* (2017) Endosomal phosphatidylserine is critical for the YAP signalling pathway in proliferating cells. *Nat Commun* 8: 1246

Meyer R, Nalaskowski MM, Ehm P, Schroder C, Naj X, Brehm MA, Mayr GW (2012) Nucleocytoplasmic shuttling of human inositol phosphate multikinase is influenced by CK2 phosphorylation. *Biol Chem* 393: 149-160

Min H, Kim W, Hong S, Lee S, Jeong J, Kim S, Seong RH (2022) Differentiation and homeostasis of effector Treg cells are regulated by inositol polyphosphates modulating Ca(2+) influx. *Proc Natl Acad Sci U S A* 119: e2121520119

Moroishi T, Hansen CG, Guan KL (2015) The emerging roles of YAP and TAZ in cancer. *Nat Rev Cancer* 15: 73-79

Papayannopoulos V, Co C, Prehoda KE, Snapper S, Taunton J, Lim WA (2005) A polybasic motif allows N-WASP to act as a sensor of PIP(2) density. *Mol Cell* 17: 181-191

Prakash S, Krishna A, Sengupta D (2023) Cofilin-Membrane Interactions: Electrostatic Effects in Phosphoinositide Lipid Binding. *Chemphyschem* 24: e202200509

Rausch V, Hansen CG (2020) The Hippo Pathway, YAP/TAZ, and the Plasma Membrane. *Trends Cell Biol* 30: 32-48

Ricks TJ, Cassilly CD, Carr AJ, Alves DS, Alam S, Tscherch K, Yokley TW, Workman CE, Morrell-Falvey JL, Barrera FN *et al* (2019) Labeling of Phosphatidylinositol Lipid Products in Cells through Metabolic Engineering by Using a Clickable myo-Inositol Probe. *Chembiochem* 20: 172-180

Rong Y, Liu M, Ma L, Du W, Zhang H, Tian Y, Cao Z, Li Y, Ren H, Zhang C *et al* (2012) Clathrin and phosphatidylinositol-4,5-bisphosphate regulate autophagic lysosome reformation. *Nat Cell Biol* 14: 924-934

Saab S, Chang OS, Nagaoka K, Hung MC, Yamaguchi H (2019) The potential role of YAP in Axl-mediated resistance to EGFR tyrosine kinase inhibitors. *Am J Cancer Res* 9: 2719-2729

Sarwar M, Syed Khaja AS, Aleskandarany M, Karlsson R, Althobiti M, Odum N, Mongan NP, Dizeyi N, Johnson H, Green AR *et al* (2019) The role of PIP5K1alpha/pAKT and targeted inhibition of growth of subtypes of breast cancer using PIP5K1alpha inhibitor. *Oncogene* 38: 375-389

Semenas J, Hedblom A, Miftakhova RR, Sarwar M, Larsson R, Shcherbina L, Johansson ME, Harkonen P, Sterner O, Persson JL (2014) The role of PI3K/AKT-related PIP5K1alpha and the discovery of its selective inhibitor for treatment of advanced prostate cancer. *Proc Natl Acad Sci U S A* 111: E3689-3698

Stephens LR, Jackson TR, Hawkins PT (1993) Agonist-stimulated synthesis of phosphatidylinositol(3,4,5)-trisphosphate: a new intracellular signalling system? *Biochim Biophys Acta* 1179: 27-75

Tan X, Sun Y, Thapa N, Liao Y, Hedman AC, Anderson RA (2015a) LAPTM4B is a PtdIns(4,5)P2 effector that regulates EGFR signaling, lysosomal sorting, and degradation. *EMBO J* 34: 475-490

Tan X, Thapa N, Choi S, Anderson RA (2015b) Emerging roles of PtdIns(4,5)P2--beyond the plasma membrane. *J Cell Sci* 128: 4047-4056

Tan X, Thapa N, Liao Y, Choi S, Anderson RA (2016) PtdIns(4,5)P2 signaling regulates ATG14 and autophagy. *Proc Natl Acad Sci U S A* 113: 10896-10901

Thapa N, Anderson RA (2012) PIP2 signaling, an integrator of cell polarity and vesicle trafficking in directionally migrating cells. *Cell Adh Migr* 6: 409-412

Thapa N, Chen M, Horn HT, Choi S, Wen T, Anderson RA (2020) Phosphatidylinositol-3-OH kinase signalling is spatially organized at endosomal compartments by microtubule-associated protein 4. *Nat Cell Biol* 22: 1357-1370

Toska E, Campbell HA, Shandilya J, Goodfellow SJ, Shore P, Medler KF, Roberts SG (2012) Repression of transcription by WT1-BASP1 requires the myristoylation of BASP1 and the PIP2-dependent recruitment of histone deacetylase. *Cell Rep* 2: 462-469

Traynor-Kaplan AE, Thompson BL, Harris AL, Taylor P, Omann GM, Sklar LA (1989) Transient increase in phosphatidylinositol 3,4-bisphosphate and phosphatidylinositol trisphosphate during activation of human neutrophils. *J Biol Chem* 264: 15668-15673

Wang Y, Wang HY (2012) Dvl3 translocates IPMK to the cell membrane in response to Wnt. *Cell Signal* 24: 2389-2395

Wang Y, Xu X, Maglic D, Dill MT, Mojumdar K, Ng PK, Jeong KJ, Tsang YH, Moreno D, Bhavana VH *et al* (2018) Comprehensive Molecular Characterization of the Hippo Signaling Pathway in Cancer. *Cell Rep* 25: 1304-1317 e1305

Wang YH, Hariharan A, Bastianello G, Toyama Y, Shivashankar GV, Foiani M, Sheetz MP (2017) DNA damage causes rapid accumulation of phosphoinositides for ATR signaling. *Nat Commun* 8: 2118

Wang YH, Sheetz MP (2022) When PIP(2) Meets p53: Nuclear Phosphoinositide Signaling in the DNA Damage Response. *Front Cell Dev Biol* 10: 903994

Wickramasinghe VO, Savill JM, Chavali S, Jonsdottir AB, Rajendra E, Gruner T, Laskey RA, Babu MM, Venkitaraman AR (2013) Human inositol polyphosphate multikinase regulates transcript-selective nuclear mRNA export to preserve genome integrity. *Mol Cell* 51: 737-750

Wu H, Wei L, Fan F, Ji S, Zhang S, Geng J, Hong L, Fan X, Chen Q, Tian J *et al* (2015) Integration of Hippo signalling and the unfolded protein response to restrain liver overgrowth and tumorigenesis. *Nat Commun* 6: 6239

Wu PF, Bhore N, Lee YL, Chou JY, Chen YW, Wu PY, Hsu WM, Lee H, Huang YS, Lu PJ *et al* (2020) Phosphatidylinositol-4-phosphate 5-kinase type 1alpha attenuates Aβ production by promoting non-amyloidogenic processing of amyloid precursor protein. *FASEB J* 34: 12127-12146

Xu R, Paul BD, Smith DR, Tyagi R, Rao F, Khan AB, Blech DJ, Vandiver MS, Harraz MM, Guha P *et al* (2013) Inositol polyphosphate multikinase is a transcriptional coactivator required for immediate early gene induction. *Proc Natl Acad Sci U S A* 110: 16181-16186

Yamaguchi H, Taouk GM (2020) A Potential Role of YAP/TAZ in the Interplay Between Metastasis and Metabolic Alterations. *Front Oncol* 10: 928

Yildirim S, Castano E, Sobol M, Philimonenko VV, Dzajak R, Venit T, Hozak P (2013) Involvement of phosphatidylinositol 4,5-bisphosphate in RNA polymerase I transcription. *J Cell Sci* 126: 2730-2739

Yip SC, Eddy RJ, Branch AM, Pang H, Wu H, Yan Y, Drees BE, Neilsen PO, Condeelis J, Backer JM (2008) Quantification of PtdIns(3,4,5)P(3) dynamics in EGF-stimulated carcinoma cells: a comparison of PH-domain-mediated methods with immunological methods. *Biochem J* 411: 441-448

Zanconato F, Battilana G, Forcato M, Filippi L, Azzolin L, Manfrin A, Quaranta E, Di Biagio D, Sigismondo G, Guzzardo V *et al* (2018) Transcriptional addiction in cancer cells is mediated by YAP/TAZ through BRD4. *Nat Med* 24: 1599-1610

Zanconato F, Cordenonsi M, Piccolo S (2019) YAP and TAZ: a signalling hub of the tumour microenvironment. *Nat Rev Cancer* 19: 454-464

Zheng W, Cai R, Hofmann L, Nesin V, Hu Q, Long W, Fatehi M, Liu X, Hussein S, Kong T *et al* (2018) Direct Binding between Pre-S1 and TRP-like Domains in TRPP Channels Mediates Gating and Functional Regulation by PIP2. *Cell reports* 22: 1560-1573

Dear Suyong,

Thank you for submitting a revised version of your manuscript. Your study has now been seen by all original referees. While referee #1 is generally satisfied with the revision, referees #2 and #3 maintain that several of the issues they raised in initial review were not satisfactorily addressed.

Please address the remaining referee issues as follows:

Reviewer 2:

Point 1 - please soften the conclusions to allow for the possibility that another product of IPMK rather than PIP3 mediates Yap/TAZ regulation.

Point 2 - please textually address the issues raised by the reviewer in a point-by-point response and textually soften the conclusions made based on these experiments.

Point 3 - soften the conclusions in the abstract and in the manuscript text to clarify the uncertainty in the involved mechanism.

Point 4 - please address experimentally.

Reviewer 3:

Point 1 - please present a quantification of the data in Fig 4B.

Points 2 and 3 - please address experimentally.

Please also address the following editorial points:

1. Please submit up to five keywords.

2. Please submit a complete author checklist, which you can download from our author guidelines ([https://wol-prod-cdn.literatumonline.com/pb-assets/embo-site/Author Checklist%20-%20EMBO%20J-1561436015657.xlsx](https://wol-prod-cdn.literatumonline.com/pb-assets/embo-site/Author%20Checklist%20-%20EMBO%20J-1561436015657.xlsx)). Please insert information in the checklist that is also reflected in the manuscript. The completed author checklist will also be part of the RPF.

3. Please check that the funding information is correct and identical both in the manuscript and our online system. CA22287, Fred & Pamela Buffett Cancer Center Pilot Projects Program, startup funds from the University of Nebraska Medical Center, Department of Defense Breast Cancer Research Program grants W81XWH-17-1-0259, W81XWH-21-1-0129, Breast Cancer Research Foundation, and National Science Foundation NSF-CHE-2310263 are currently missing in our online system.

4. Main figures should be uploaded as individual, high res figure files in PDF, TIFF or EPS format (no PPT). Up to 5 suppl figures can be made EV figures and also uploaded as individual files. Headings would need renaming to "Figure EV1" etc, and legends should be placed after the main figure legends. The remaining two suppl. figures should be compiled in a PDF and renamed "Appendix Figure S1, etc.". The Appendix PDF needs a ToC with page numbers, and the legends should be included in the appendix file, underneath the figures. Main figures and EV figures have to fit onto one page (correction needed for Fig 7); Appendix figures (e.g. Suppl. Fig 7 if made an Appendix figure) can span more than one page.

5. We require a Data Availability Section at the end of Materials and Methods. As far as I can see, no data deposition in external databases is needed for this paper. If I am correct, then please state in this section: This study includes no data deposited in external repositories. Further information can be found at

<https://www.embopress.org/page/journal/14602075/authorguide#dataavailability>

6. Our data editors have flagged the following issues in figure legends that need correcting:

- Please define the annotated p values **/* in the legend of figure 1c; 2a-b; 3d; 4c-d, f, h; 5b-c; 6c; 7b, c, f, g; supplementary figures 1b; 3a; 4f-g; 5d as appropriate.

- Please indicate the statistical test used for data analysis in the legends of figures 1c; 2a-b; 3d; 4c-d, f, h; 5b-c; 6c; 7a-g; supplementary figures 1b; 3a-c; 4f-g; 5d

- Please note that the box plots need to be defined in terms of minima, maxima, centre, bounds of box and whiskers, and percentile in the legend of figure 7g.

7. CRediT has replaced the traditional author contributions section because it offers a systematic, machine-readable author contributions format that allows for more effective research assessment. Please remove the Authors Contributions from the manuscript and use the free text boxes beneath each contributing author's name in our online submission system to add specific details on the author's contribution. More information is available in our guide to authors.

8. Please add a Disclosure and competing interests statement after Acknowledgments (further info:

<https://www.embopress.org/page/journal/14602075/authorguide#conflictsofinterest>).

9. Please assemble source data into one folder per figure and upload as .zip files. For example, the Source data files for figure 1 need to be saved in a single folder and this needs to be zipped and then uploaded as "SD figure 1.zip" file.

10. Papers published in The EMBO Journal are accompanied online by a 'Synopsis' to enhance discoverability of the manuscript. It consists of A) a short (1-2 sentences) summary of the findings and their significance, B) 3-4 bullet points highlighting key results and C) a synopsis image that is 550x300-600 pixels large (width x height, jpeg or png format). You can either show a model or key data in the synopsis image. Please note that the image size is rather small and that text needs to be readable at the final size. Please send us this information together with the revised manuscript.

With best wishes,

Ieva

We realize that it is difficult to revise to a specific deadline. In the interest of protecting the conceptual advance provided by the work, we recommend a revision within 3 months (15th Feb 2024). Please discuss the revision progress ahead of this time with the editor if you require more time to complete the revisions.

Referee #1:

The manuscript entitled "Nuclear phosphoinositide signaling controls the YAP/TAZ-TEAD pathway in breast cancer" by Jung et al. describes PI4,5P2 and PI3,4,5P3 as YAP/TAZ interactors. The revised manuscript proposes the following mode: PIPKI α and IPMK form a nuclear complex with TEAD, independent of Hippo pathway activators. Upon stimulation of YAP and translocation to the nucleus, the presence of these kinases and phospholipids will contribute to YAP-TEAD complex formation, DNA binding and subsequent activation of YAP/TAZ transcriptional program. The work proposed by Jung. et al supports the hypothesis that PI4,5P2 and PI3,4,5P3 act as modulators of YAP transcriptional activity.

The authors have carried out an extensive revision of the manuscript and they have addressed all my concerns for improving the paper.

However, a note to the authors that the revised manuscript was difficult to navigate because the changes were not highlighted or pointed out in the actual manuscript document. This made reviewing the manuscript very time consuming and it should have not gone to reviewers in this form.

Referee #2:

The manuscript by Jung, et al has not addressed the majority of the concerns raised by this reviewer. In several cases I provided the option to simply change language to more appropriately fit the data, but they have not chosen to do so. In terms of additional experiments, I had one requested experiment that was threaded throughout all my critiques, which the authors did not attempt (3H-inositol metabolic labeling studies to test YAP/TEAD association with phosphoinositides). They did execute a new click-probe labeling experiment, but this was not done with acceptable rigor, the design of the experiment was inherently flawed and the interpretation of the observed results relies on a rather wild assumption. Of my 4 critiques, none were addressed adequately:

Point 1: This experiment is starting to drive at the question at hand, but the experiment as executed is flawed. Inositol phosphates are present in cells between 10-50uM, but the authors tested at only 0.1uM, far below physiological range. However, even if they repeat the experiment at 100uM and see no direct effect on YAP-TEAD, in cells the effect could be indirect, e.g. loss of inositol phosphates indirectly decrease protein phosphorylation or some other mechanism. This is why I suggested the 3H-inositol labeling experiment. If they absolutely need to conclude it's a phosphoinositide and not an inositol phosphate (which I'm not sure why they must), they need to show tritiated glycerol-inositol phosphates (i.e. the product of phosphoinositide deacylation reactions) associated with mammalian YAP-TEAD. I don't see any way around that.

Point 2: Fig 4B and Supplemental 4B there are many problems here, I'll go into six of them point by point. First, the detection method for this new experiment is flawed. Any phospholipids associated with YAP / TEAD would dissociate on SDS-PAGE. The results are presented as westerns blotted against streptavidin to detect the labeled phosphoinositide, and therefore should be blank. The authors interpret the very not-blank western results they actually observe as the phosphoinositides being somehow covalently attached to YAP and TEAD and p53 (Rebutall Fig 3), without much data to support this remarkable interpretation. I will state here that this experiment is the most convincing data I have seen thus far suggesting that phosphoinositide interaction with proteins is covalent. However that is not the focus of this paper, and to reach that conclusion far more work needs to be done, which I would encourage the authors to follow up on. In the present study, its just not an acceptable explanation of the data.

Second, the peracetylated probe used here from Michael Best was demonstrated to be ineffective (i.e. NOT effective) at labeling phosphoinositides in the cited Ricks, et al. 2019 paper (Compound 1b from that paper), in both human and yeast cells. The un-acetylated inositol probe from that paper was shown to be effective at labeling both human and yeast cells. The authors here did not explain why they did not use the simpler inositol probe, they also did not validate the acetylated probe with any rigor, and they did not cite any papers that have effectively used the peraceylated probe.

Third, its unclear why cooper-click was used rather than cooper-free click, either could have been used with this probe. This is a problem because cooper can cause protein denaturation and aggregation, consistent with some of the some of the poor westerns in the paper here (in particular Supplemental 4B) making interpreting the westerns difficult.

Fourth, the authors argue in rebuttal that the PIP2 antibodies are well validated. I disagree. Regardless, the real problem is the stated conclusions derived from the reagents they used. The authors conclude PIP2 levels change when PIP2 antibody reactivity changes, which of course is not how one should interpret a western, PLA or IF experiment. One can only formally conclude that antibody reactivity changes, not that levels of any particular antigen change.

Fifth, its unclear why the authors have not done the requested tritiated inositol metabolic labeling as requested. They state its not in their expertise, but neither is copper-click which they tried. I would suggest, once again, simply changing the language they use in their conclusions to fit the experiments they did, that would satisfy this reviewer.

Sixth - no triplicates, no quantitation, no statistics, poor westerns, but even if that were all fixed, the experiment as executed here was flawed.

Point 3: The authors state they don't have a clear explanation for the data in new Fig 4E and 4G, but they certainly have clear conclusions from that data stated in the abstract, which is why I have a problem with this. The simplest interpretation of new Fig 4E and 4G is the PIP2 and PIP3 antibodies are cross reacting with protein phosphorylation in the PLA, which the authors have not addressed.

Point 4: This is an encouraging start but it has to be finished. It needs to be done in triplicate, non-linear fitted to a one site model (or two site), with r squared reported and EC50 determined. You've got the assay all you need to do is get the EC50 and do some negative control phospholipids, such as identical chain phosphatidyl serine or better yet inactive phosphoinositides. This point is the closest to being addressed.

Minor:

All westerns need MW markers above and below all bands.

The following sentence does not make sense in the methods:

The binding assays were performed in the same lysis buffer that was used for immunoprecipitation by adding 50 nM to 10 M His-tagged proteins and 20 I glutathione beads with the GST-tagged proteins bound to them.

Referee #3:

The study by Oisun Jung et al. proposed a new mechanistic scheme that nuclear phosphoinositides generated by PIPK1 α and IPMK regulate the spatiotemporal binding of YAP/TAZ to TEADs. During the 1st round of review, we raised concerns that the direct evidence of PI4,5P2 and PI3,4,5P3 binding to YAP/TAZ is based on in vitro assay with high concentrations of PI4,5P2 and PI3,4,5P3. The authors attempted to use the metabolic labeling method to detect the binding of YAP/TAZ with these lipids. However, the results may need further verification (see below, Q1). At the same time, some other minor issues are listed for the authors' reference to further enhance the quality and rigor of the manuscript before its publishing in the EMBO Journal.

1, The authors have claimed that they utilized a metabolic labeling method with a clickable myo-inositol probe to further validate the association of phosphoinositides with the YAP/TEAD complex. However, in Figure 4B, the change of click and non-click is

very minor, primarily due to the change of input, especially for TEAD. Besides, the Streptavidin blot is very dirty.

2, In revised Figure 1F, the input of YAP is affected by PIPKI α WT and LPGAA overexpression; the authors should repeat and confirm it.

3, In response to the comments "6. The authors showed two bands for PIPKI α and IPMK in some blots (Figure 1H, 2A), while in some blots, there is only one band (Figure 2E, 2F), The author should clarify the discrepancy. The authors have claimed these IPMK immunoblot bands were diminished upon depletion of IPMK in MDA-MB-231 cells. Could the authors provide some evidence with additional blots?

We would like to thank the referees for their invaluable comments and suggestions. Below we detail the changes to the revised manuscript that address the referees' comments followed by the revisions that we have made.

Reviewer #2:

Point 1: This experiment is starting to drive at the question at hand, but the experiment as executed is flawed. Inositol phosphates are present in cells between 10-50 μM , but the authors tested at only 0.1 μM , far below physiological range. However, even if they repeat the experiment at 100 μM and see no direct effect on YAP-TEAD, in cells the effect could be indirect, e.g. loss of inositol phosphates indirectly decrease protein phosphorylation or some other mechanism. This is why I suggested the 3H-inositol labeling experiment. If they absolutely need to conclude it's a phosphoinositide and not an inositol phosphate (which I'm not sure why they must), they need to show tritiated glycerol-Inositol phosphates (i.e. the product of phosphoinositide de-acylation reactions) associated with mammalian YAP-TEAD. I don't see any way around that.

We softened the conclusions in the revised manuscript to allow for the possibility that other products of IPMK (i.e. inositol phosphates) rather than PI3,4,5P₃ mediates YAP/TAZ regulation.

Point 2: Fig 4B and Supplemental 4B there are many problems here, I'll go into six of them point by point.

First, the detection method for this new experiment is flawed. Any phospholipids associated with YAP/TEAD would dissociate on SDS-PAGE. The results are presented as westerns blotted against streptavidin to detect the labeled phosphoinositide, and therefore should be blank. The authors interpret the very not-blank western results they actually observe as the phosphoinositides being somehow covalently attached to YAP and TEAD and p53 (Rebuttal Fig 3), without much data to support this remarkable interpretation. I will state here that this experiment is the most convincing data I have seen thus far suggesting that phosphoinositide interaction with proteins is covalent. However, that is not the focus of this paper, and to reach that conclusion far more work needs to be done, which I would encourage the authors to follow up on. In the present study, it is just not an acceptable explanation of the data.

We were also surprised by the observations that the labeled phosphoinositides on target proteins are resistant to SDS-PAGE. One of direct interpretations of this result is that phosphoinositides are covalently attached to the targets. The retention of phosphoinositides on targets after SDS-PAGE was first observed in p53 and we have conducted follow up experiments as the reviewer encouraged. We hope the reviewer understands that we have not included the actual data in this rebuttal as a manuscript is under consideration at another journal. The data supporting the covalent association of phosphoinositides on p53 is outlined as follows. First, the covalent association was detected in p53 purified from insect and mammalian cells but not from *E. coli*. Second, we defined regions on p53 that are responsible for the covalent association. Third, we performed RNAi screens to identify enzymes involved in the covalent association. Fourth, we functionally validated a couple of enzymes identified from the RNAi screen.

We have modified the text in our revised manuscript to avoid any conceptual leap in interpretation of the data as the reviewer suggested.

Second, the peracetylated probe used here from the Dr. Michael Best group was demonstrated to be ineffective (i.e. NOT effective) at labeling phosphoinositides in the cited Ricks, et al. 2019 paper (Compound 1b from that paper), in both human and yeast cells. The un-acetylated inositol probe from that paper was shown to be effective at labeling both human and yeast cells. The authors here did not explain why they did not use the simpler inositol probe, they also did not validate the acetylated probe with any rigor, and they did not cite any papers that have effectively used the peracetylated probe.

We used both un-acetylated (2API or compound 1a in Ricks, et al. 2019 paper) and peracetylated (Ac₃2API or compound 1b) and tested for phosphoinositide labeling on p53 and YAP using MDA-MB-231 cells (Rebuttal Fig. 1). p53 was effectively labeled with both 2API and Ac₃2API in MDA-MB-231

cells, while in our initial experiments Ac₃2API behaved better than 2API on YAP. Since then, we decided to use Ac₃2API for probing the association of phosphoinositides with the YAP-TEAD complex. Although 2API worked better than Ac₃2API in T-24 human bladder cancer cells (Ricks *et al*, 2019), our data suggest that the two clickable inositol probes behave differently in a cell type and in a target-specific manner.

Figure for reviewers removed

Third, it is unclear why copper-click was used rather than copper-free click, either could have been used with this probe. This is a problem because copper can cause protein denaturation and aggregation, consistent with some of the poor westerns in the paper here (in particular Supplemental 4B) making interpreting the westerns difficult.

We initially utilized phosphate buffered saline (PBS) for our click reaction. Although PBS is widely used for copper-free and copper-catalyzed click reactions (Baskin *et al*, 2007; Kurra *et al*, 2014; Li *et al*, 2017), we observed that the presence of copper in the reactions induced protein aggregation in PBS-containing reaction conditions. We then optimized the copper-catalyzed click reactions with a lysis buffer containing 1% Brij58, 150 mM NaCl, 20 mM HEPES, pH 7.4, and no noticeable protein aggregation was observed in this condition. As the copper-catalyzed click reactions effectively labeled p53 and YAP (Rebuttal Fig. 1), we have not tried the copper-free reactions for this study.

Fourth, the authors argue in rebuttal that the PIP2 antibodies are well validated. I disagree. Regardless, the real problem is the stated conclusions derived from the reagents they used. The authors conclude PIP2 levels change when PIP2 antibody reactivity changes, which of course is not how one should interpret a western, PLA or IF experiment. One can only formally conclude that antibody reactivity changes, not that levels of any particular antigen change.

We acknowledge Reviewer 2's evaluation of the anti-PI4,5P₂ antibodies. We changed our conclusions from "level change" to "reactivity change" in the revised manuscript.

Fifth, it is unclear why the authors have not done the requested tritiated inositol metabolic labeling as requested. They state it is not in their expertise, but neither is copper-click which they tried. I would suggest, once again, simply changing the language they use in their conclusions to fit the experiments they did, that would satisfy this reviewer.

We discovered the associations of p53 with phosphoinositides and the results were published in 2019 (Choi *et al*, 2019) and 2022 (Chen *et al*, 2022). Since 2020, we started collaborating with the Dr. Michael Best group and found that the clickable inositol probes are working on p53 (Rebuttal Fig. 1). In the first round of revision for our current study, we decided to apply the click approach to a new target YAP-TEAD complex. Instead of relying on ³H-inositol labeling which was totally new to us, we decided to further test our hypothesis with an established click labeling method. We changed the language in the revised manuscript as the Reviewer 2 suggested.

Sixth - no triplicates, no quantitation, no statistics, poor westerns, but even if that were all fixed, the experiment as executed here was flawed.

We updated Fig. 4B with quantification. You can find the quantification results in Fig. EV2F.

Point 3: The authors state they don't have a clear explanation for the data in new Fig 4E and 4G, but they certainly have clear conclusions from that data stated in the abstract, which is why I have a problem with this. The simplest interpretation of new Fig 4E and 4G is the PI4,5P₂ and PI3,4,5P₃ antibodies are cross reacting with protein phosphorylation in the PLA, which the authors have not addressed.

We softened the conclusions in the abstract and in the manuscript text to clarify the uncertainty in the involved mechanism.

Point 4: This is an encouraging start, but it has to be finished. It needs to be done in triplicate, non-linear fitted to a one site model (or two site), with r squared reported and EC50 determined. You've got the assay all you need to do is get the EC50 and do some negative control phospholipids, such as identical chain phosphatidyl serine or better yet inactive phosphoinositides. This point is the closest to being addressed.

We measured binding affinities between YAP and phosphoinositides using microscale thermophoresis (MST). A constant concentration of fluorescently labeled GST or GST-YAP (target, 5 nM) was incubated with increasing concentrations of non-labeled 18:0/20:4 phosphatidylinositol (PI), PI4,5P₂, or PI3,4,5P₃ to calculate the dissociation constants (K_d). Initially, we checked the quality of the interaction by incubating 5 nM GST proteins with 5 μ M lipids in triplicate. Note, a signal-to-noise ratio should be above 5.0 for an interaction to be reproducibly measured in the MST instrument we used. The interaction of GST-YAP with PI4,5P₂ and PI3,4,5P₃ passed the quality check, while other interactions were not significant. Therefore, we further measured the K_d s of GST-YAP binding with PI4,5P₂ and PI3,4,5P₃ in triplicate. The interaction of GST-YAP with PI4,5P₂ and PI3,4,5P₃ showed sigmoidal curves and the calculated K_d s were 165.67 ± 13.50 and 8.13 ± 2.63 nM, respectively. The raw data of the MST experiments were shown in Fig. EV2D and Appendix Fig. S1. Considering the cellular concentrations of PI4,5P₂ and PI3,4,5P₃ are greater than several μ M in stimulated cells (Insall & Weiner, 2001), the binding of YAP to PI4,5P₂ and PI3,4,5P₃ could occur at physiological concentrations. Using the MST approach, we previously reported K_d s of p53 binding to PI4,5P₂ and PI3,4,5P₃ ranging from 80 nM to 1.5 μ M (Chen *et al.*, 2022), pointing out that YAP binding to phosphoinositides is comparable to or tighter than p53. This information is updated in the revised manuscript.

Reviewer #3:

1. The authors have claimed that they utilized a metabolic labeling method with a clickable myo-inositol probe to further validate the association of phosphoinositides with the YAP/TEAD complex. However, in Figure 4B, the change of click and non-click is very minor, primarily due to the change of input, especially for TEAD. Besides, the Streptavidin blot is very dirty. Fig. 4B is updated with quantification (Fig. EV2F) in the revised manuscript.

2. In revised Figure 1F, the input of YAP is affected by PIPKI α WT and LPGAA overexpression; the authors should repeat and confirm it.

We utilized HEK293 cells to overexpress HA-PIPKI α and Flag-YAP. We found that maintaining similar levels of exogenous protein expression was not easy in HEK293 cells even though the same amounts of the DNAs were used for transfection. We provide immunoblot images of all three replicates (including the one in Fig. 1F). The expression levels of exogenous proteins varied, and it doesn't appear that exogenous YAP expression is consistently affected by HA-PIPKI α overexpression (Rebuttal Fig. 2).

3. In response to the comments "6. The authors showed two bands for PIPKI α and IPMK in some blots (Figure 1H, 2A), while in some blots, there is only one band (Figure 2E, 2F), The author should clarify the discrepancy. The authors have claimed these IPMK immunoblot bands were diminished upon depletion of IPMK in MDA-MB-231 cells. Could the authors provide some evidence with additional blots?

Rebuttal Fig. 2 HEK293 cells were transfected with the indicated HA or Flag tagged constructs. Cell lysates were separated by SDS-PAGE and the expression of exogenous proteins and endogenous actin were analyzed by immunoblotting with the indicated antibodies.

Rebuttal Fig. 3 MDA-MB-231 cells were transfected with the indicated siRNAs. The expression of endogenous IPMK or actin was resolved by immunoblotting with different antibodies.

As we detailed in the previous rebuttal, one vs. two (or more) immunoblot bands could result from the use of different antibodies, the degree of separation on SDS-PAGE, and the length of exposure of the immunoblots. We further analyzed these antibody issues. We tested almost all the commercially available IPMK antibodies as well as some home-made antibodies obtained from different research groups via immunoblotting. We found many of those antibodies tested did not work properly in our hands, while the four antibodies shown in **Rebuttal Fig. 3** provided consistent results. The four antibodies showed different banding patterns between molecular weight of 55 and 40 kDa, but importantly the immunoblot bands were consistently diminished by knockdown of IPMK compared to control. We envision that different sizes of bands between molecular weight of 55 and 40 kDa might represent IPMK isoforms or posttranslational modifications. As noted in the previous rebuttal, most of the experiments were done with the Invitrogen antibody.

Reference cited:

- Baskin JM, Prescher JA, Laughlin ST, Agard NJ, Chang PV, Miller IA, Lo A, Codelli JA, Bertozzi CR (2007) Copper-free click chemistry for dynamic in vivo imaging. *Proc Natl Acad Sci U S A* 104: 16793-16797
- Chen M, Choi S, Wen T, Chen C, Thapa N, Lee JH, Cryns VL, Anderson RA (2022) A p53-phosphoinositide signalosome regulates nuclear AKT activation. *Nat Cell Biol* 24: 1099-1113
- Choi S, Chen M, Cryns VL, Anderson RA (2019) A nuclear phosphoinositide kinase complex regulates p53. *Nat Cell Biol* 21: 462-475
- Insall RH, Weiner OD (2001) PIP3, PIP2, and cell movement--similar messages, different meanings? *Dev Cell* 1: 743-747
- Kurra Y, Odoi KA, Lee YJ, Yang Y, Lu T, Wheeler SE, Torres-Kolbus J, Deiters A, Liu WR (2014) Two rapid catalyst-free click reactions for in vivo protein labeling of genetically encoded strained alkene/alkyne functionalities. *Bioconjug Chem* 25: 1730-1738
- Li S, Wang L, Yu F, Zhu Z, Shobaki D, Chen H, Wang M, Wang J, Qin G, Erasquin UJ *et al* (2017) Copper-Catalyzed Click Reaction on/in Live Cells. *Chem Sci* 8: 2107-2114
- Ricks TJ, Cassilly CD, Carr AJ, Alves DS, Alam S, Tscherch K, Yokley TW, Workman CE, Morrell-Falvey JL, Barrera FN *et al* (2019) Labeling of Phosphatidylinositol Lipid Products in Cells through Metabolic Engineering by Using a Clickable myo-Inositol Probe. *Chembiochem* 20: 172-180

Dear Suyong,

Thank you for addressing the remaining issues. I am now pleased to inform you that your manuscript has been accepted for publication in the EMBO Journal.

I will look into the synopsis text in the next couple of days and let you know if any edits to the journal style are needed.

If you have any questions, please do not hesitate to contact the Editorial Office. Thank you for this contribution to The EMBO Journal and congratulations on a very interesting study!

Best wishes,

leva

leva Gailite, PhD
Senior Scientific Editor
The EMBO Journal
Meyerhofstrasse 1
D-69117 Heidelberg
Tel: +4962218891309
i.gailite@embojournal.org
